# Monitoring and modeling seasonally varying anthropogenic and biogenic $CO_2$ over a large tropical metropolitan area

Rafaela Cruz Alves Alberti[1], Thomas Lauvaux[3], Angel Liduvino Vara-Vela[6,7,8], Ricard Segura Barrero[2], Christoffer Karoff[6,7,8], Maria de Fátima Andrade[1], Márcia Talita Amorim Marques[1], Noelia Rojas Benavente[5], Osvaldo Machado Rodrigues Cabral[4], Humberto Ribeiro da Rocha[1], and Rita Yuri Ynoue[1]

[1]Department of Atmospheric Sciences, University of São Paulo, Brazil
[2]Institute of Environmental Sciences and Technology, Universitat Autònoma de Barcelona, Spain.
[3]Universite´ de Reims Champagne-Ardenne, CNRS, GSMA, Reims, France
[4]Brazilian Agricultural Research Corporation, Embrapa Environment, Brazil
[5]Physics Institute, University of São Paulo, Brazil
[6]Department of Geoscience, Aarhus University, Denmark
[7]Department of Physics and Astronomy, Aarhus University, Aarhus, Denmark
[8]iCLIMATE Aarhus University Interdisciplinary Centre for Climate Change, Aarhus, Denmark

**Correspondence:** Rafaela Cruz Alves Alberti (rafaela_alves@usp.br)

**Abstract.** Atmospheric $CO_2$ concentrations in urban areas reflect a combination of fossil fuel emissions and biogenic fluxes, offering a potential approach to assess city climate policies. However, atmospheric models used to simulate urban $CO_2$ plumes face significant uncertainties, particularly in complex urban environments with dense populations and vegetation. This study addresses these challenges by analyzing $CO_2$ dynamics in the Metropolitan Area of São Paulo (MASP) using the Weather Research and Forecasting model with Chemistry (WRF-Chem). Simulations were evaluated against ground-based observations from the METROCLIMA network, the first greenhouse gas monitoring network in South America, and column concentrations ($XCO_2$) from the OCO-2 satellite spanning February to August 2019. To improve biogenic fluxes, we optimized parameters in the Vegetation Photosynthesis and Respiration Model (VPRM) using eddy covariance flux measurements for key vegetation types, including the Atlantic Forest, Cerrado, and sugarcane. Results show that at the urban site (IAG), the model consistently underestimated $CO_2$ concentrations, with a negative mean bias of -9 ppm throughout the simulation period, likely due to the complexity of vehicular emissions and urban dynamics. In contrast, at the vegetated site (PDJ), simulations showed a consistent positive mean bias of 5 ppm and closely matched observations. Seasonal analyses revealed higher $CO_2$ concentrations in winter, driven by greater atmospheric stability and reduced vegetation uptake estimated by VPRM, while summer exhibited lower levels due to increased mixing and higher agricultural productivity. A comparison of biogenic and anthropogenic scenarios highlights the need for integrated emission modeling and improved representation of biogenic fluxes, anthropogenic emissions, and boundary conditions for high-resolution modeling in tropical regions.

# 1 Introduction

Urban areas, although occupying only a small fraction of the Earth's surface, exert an outsized influence on global carbon emissions. Accounting for a staggering 70% of $CO_2$ emissions from fossil fuel burning while covering just 2% of the planet's landmass (Seto et al., 2014; Change et al., 2014), cities have become focal points for climate action. The relentless pace of urbanization has further exacerbated this phenomenon, driving up energy consumption and emissions levels (Seto et al., 2012). Consequently, combating climate change necessitates a targeted approach, with policies increasingly tailored to address urban emissions. In response to the growing need for climate action, initiatives like the International Council for Local Environmental Initiatives (ICLEI), the C40 Cities Climate Leadership Group (C40), and the Covenant of Mayors (CoM) have emerged to coordinate global efforts and share best practices among cities. These initiatives highlight the crucial role cities play in the fight against climate change and the importance of localized mitigation strategies. São Paulo, Brazil's largest municipality (IBGE, 2021), is a member of C40 and focuses on reducing greenhouse gas emissions, with transportation accounting for 58% of its total emissions (SEEG, 2019). The city is working towards carbon neutrality through projects in green infrastructure, urban planning, public transportation improvements, energy efficiency, and waste management (Caetano et al., 2021). These efforts aim to reduce emissions and enhance São Paulo's resilience, fostering a more sustainable urban environment. Central to these efforts is the need for accurate data and robust modeling frameworks to inform policy decisions effectively. Urban atmospheric networks, such as MASP, in Brazil, provide vital insights into greenhouse gas concentrations and emission patterns. By leveraging these datasets alongside sophisticated atmospheric transport models and statistical techniques, policymakers gain tools for designing targeted interventions and monitoring their efficacy. However, the complexity of urban $CO_2$ dynamics presents significant challenges for modeling and analysis. Process-driven biosphere models and inverse modeling techniques offer complementary approaches for capturing the intricate spatio-temporal variabilities inherent in urban environments (Kaiser et al.; Che et al., 2022; Zhang et al., 2023; Wilmot et al., 2024). Despite advancements in modeling capabilities, gaps remain in our understanding of $CO_2$ dynamics, particularly at regional and national scales. South America, in particular, suffers from limited data availability, and research focusing on this region is scarce. Additionally, vegetation models in tropical regions often exhibit poor performance due to inaccuracies in simulating seasonality, oversimplified representations of biodiversity, and errors in carbon and water cycle interactions. These models struggle to capture the complex dynamics of tropical ecosystems, leading to underestimations of productivity and poor predictions of vegetation responses to climate variability (De Pue et al., 2023; He et al., 2024). This study aims to address these gaps by conducting a comprehensive analysis of anthropogenic and biospheric $CO_2$ dynamics near the MASP. To achieve this, we employed the WRF-Chem model, offline coupled with the VPRM model (Mahadevan et al., 2008). Vehicular emissions were incorporated using the Vehicle Emission Inventory model (VEIN) (Ibarra-Espinosa et al., 2018), while emissions from the industrial, energy, residential, and refinery sectors were derived from the EDGAR inventory. This integrated modeling framework enables a detailed assessment of the main drivers of $CO_2$ variability in the region. In addition, we utilized data from the OCO-2 satellite to cover the study domain, comparing WRF-Chem-simulated $XCO_2$ concentrations (considering biogenic and anthropogenic emissions) post-processed using OCO-2 averaging kernels (i.e., smoothed $XCO_2$). Through a combination of model simulations, field observations, and

satellite data analysis, this study seeks to provide an understanding of $CO_2$ dynamics in urban environments. This is the first study in this field conducted in any city in the Global South, making it an innovative effort with significant implications. By setting a precedent, this research paves the way for future studies, contributing to a more comprehensive global picture of $CO_2$ dynamics in urban environments.

## 2  WRF-Chem

### 2.1  Model set-up

A set of high-resolution simulations of atmospheric Greenhouse Gas concentrations were performed with the WRF-Chem model version 4.0. The WRF-Chem was used to simulate the transport of the mole fraction of $CO_2$, and no chemical processes or reactions have been used. The period simulated was from 1 February to 31 August 2019. This period was selected due to available data from monitoring stations from the METROCLIMA network for $CO_2$. The simulations were made for each month. For each run, the simulation was initiated 5 days before and these 5 days were discarded as spin-up time. The single modeling domain was centered at 23.5°S and 46.3°W with a horizontal grid spacing of 3 km as shown in Figure 1, projected on a Lambert plane and consists of 166 grid points in the west-east direction, 106 grid points in the north-south direction, and 34 vertical levels that extend from the surface up to 50 hPa (20 km), as used in previous studies for this same area (Andrade et al., 2015; Vara-Vela et al., 2016; Gavidia-Calderón et al., 2023; Benavente et al., 2023). The meteorological conditions used to drive the simulations were obtained from the European Centre for Medium-Range Weather Forecasts (ECMWF) ERA5 reanalysis dataset, with a horizontal resolution of 0.25º × 0.25º and 6-hourly intervals (Hersbach, 2016). For $CO_2$, initial and boundary conditions were provided by Carbon Tracker, which offers data at a horizontal resolution of 3° in longitude and 2° in latitude, with 25 vertical layers (http://carbontracker.noaa.gov). This global dataset was interpolated to provide lateral boundary conditions for the simulations and ensure consistency with the WRF-Chem. The main physics and chemistry options used in this study are listed in Table 1.

### 2.1.1  Anthropogenic Emissions

In the MASP, the vehicular fleet is the primary source of $CO_2$ emissions (CETESB, 2019). For this study, we employed the VEIN model, a tool designed to estimate emissions from mobile sources. VEIN accounts for both exhaust and evaporative emissions performs speciation, and includes functions to generate and spatially allocate emissions databases (Ibarra-Espinosa et al., 2018). The model enables the use of customized emission factors, which in this study were derived from experimental campaigns conducted in traffic tunnels within São Paulo (Nogueira et al., 2021). VEIN processes vehicle fleet age distributions extrapolates hourly traffic data, and estimates emissions with high temporal and spatial resolution. For consistency with the WRF-Chem model domain, VEIN emissions were aggregated to a 3 km spatial resolution. Additionally, we included Figure B1 in Appendix B, which illustrates the spatial distribution of average daily $CO_2$ emissions for August 2019, the total monthly emissions from February to August, and the diurnal profile of vehicular $CO_2$ emissions as estimated by the VEIN model.

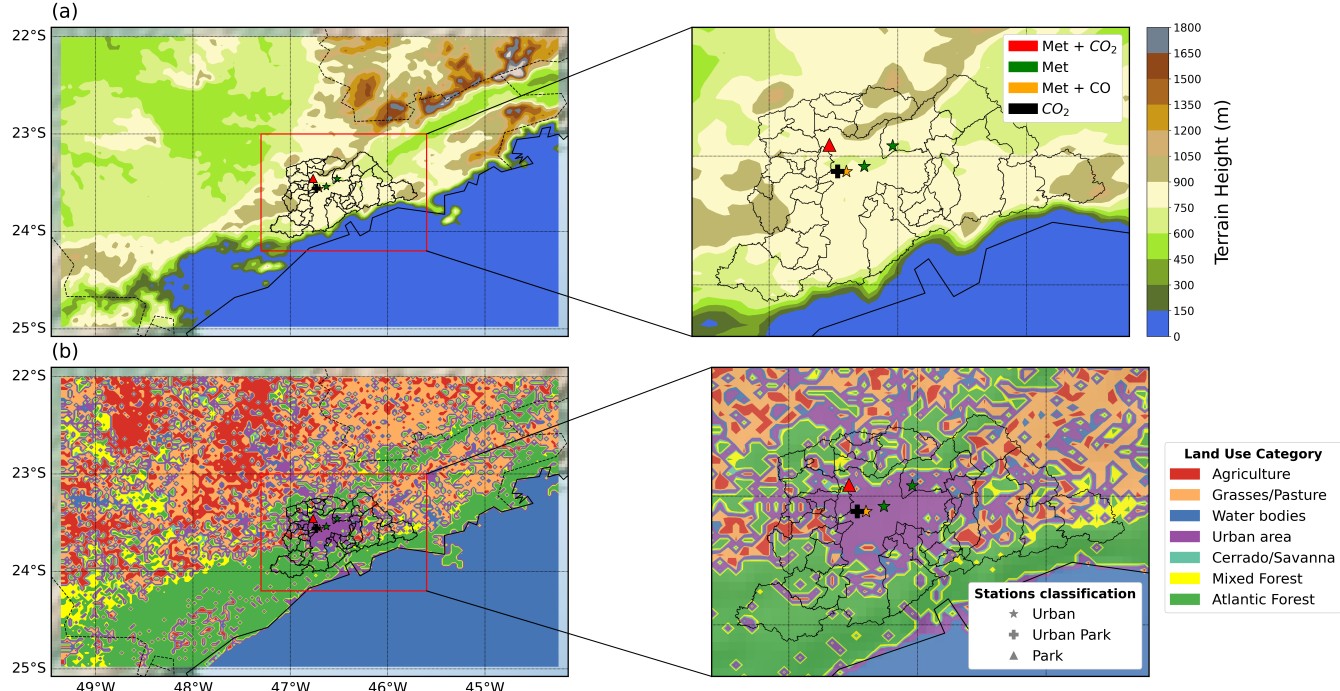

**Figure 1.** Panel (a) shows the terrain height and urban boundaries of the MASP region within the WRF-Chem model domain (D01). Station classifications are indicated using different symbols: Urban (★), Urban Park (✚), and Park (▲). Panel (b) presents the land use category map for the same domain (D01), which was used by the VPRM model to calculate $CO_2$ fluxes. The colors of the station markers represent the type of measurements conducted at each location: red indicates stations measuring both meteorological variables (Met) and $CO_2$ concentrations; green indicates stations measuring only Met; dark yellow denotes stations measuring both Met and CO concentrations; and black indicates stations measuring only $CO_2$ concentrations. The IAG station is marked as (✚), the PDJ station is (▲), Pinheiros station is (★), Guarulhos and Parque D.Pedro II are (★).

Emissions from the industry, refineries, residential, and energy sectors were obtained from the EDGAR v6.0 GHG inventory for 2018 (Crippa et al., 2021). EDGAR provides global annual emissions at $0.1° \times 0.1°$ spatial resolution, which we regridded to 3 km using bilinear interpolation to match the WRF-Chem model domain. EDGAR does not provide hourly temporal profiles, these emissions were assumed constant over the day (Figure B2 in Appendix B). To evaluate the relative contribution of each sector to total emissions in the MASP, Figure B3 (Appendix B) presents the average daily $CO_2$ emissions in August 2019. Transport emissions represented the dominant share, accounting for 76.1%, followed by industry (10.0%), refineries (7.6%), residential (3.8%), and energy (2.5%) sectors.

### 2.1.2 Biogenic Fluxes

Biogenic $CO_2$ fluxes were simulated offline using the VPRM model (Mahadevan et al., 2008) and incorporated as flux input data in the WRF-Chem simulations. This model estimates net ecosystem exchange (NEE) by calculating the difference between

**Table 1.** WRF-Chem Simulation Design.

| Atmosphere Schemes | | |
|---|---|---|
| Scheme | Type | Description/Reference |
| Microphysics | Two-moment | Morrison scheme (Morrison et al., 2009) |
| Longwave radiation | RRTMG | (Iacono et al., 2008) |
| Shortwave radiation | RRTMG | (Iacono et al., 2008) |
| Boundary layer | YSU | (Hong et al., 2006) |
| Land surface | Noah LSM | Unified scheme (Tewari et al., 2007) |
| Initial and Lateral Boundary Conditions | | |
| Meteorological | ERA5 | $0.25°$, 34 pressure levels |
| $CO_2$ | Carbon Tracker | 25 vertical layers |
| Emissions Inventories/Model | | |
| Anthropogenic | EDGAR v6.0 | (Crippa et al., 2021) and VEIN (Ibarra-Espinosa et al., 2018) |
| Biogenic | VPRM | (Mahadevan et al., 2008) |

gross ecosystem exchange (GEE) and ecosystem respiration (R), where negative fluxes indicate $CO_2$ absorption by ecosystems (Equation 1).

$$NEE = GEE - R \tag{1}$$

The meteorological variables 2m air temperature ($T_{2m}$) and downward shortwave radiation (PAR) from WRF model simulations were used to calculate the GEE (Equation 2) and Respiration (Equation 3) fluxes. Additionally, factors such as the light use efficiency ($\lambda$), PAR saturation (PAR0), and the Enhanced Vegetation Index (EVI), which refer to the fraction of shortwave radiation absorbed by leaves were used to calculate GEE. The temperature sensitivity of the photosynthesis parameter (Tscale) and the effects of leaf age on canopy photosynthesis parameter (Pscale) were both calculated as functions of the land surface water index (LSWI) to identify the green-up (leaf expansion) and senescence phases (Mahadevan et al., 2008). These vegetation indices were derived from Moderate Resolution Imaging Spectroradiometer (MODIS) reflectance data from MOD09A1 Version 6 (Vermote, 2021).

$$GEE = \lambda \times T_{\text{scale}} \times P_{\text{scale}} \times W_{\text{scale}} \times EVI \times \frac{1}{1 + \frac{PAR}{PAR_0}} \times PAR \tag{2}$$

Respiratory fluxes (R) were estimated using a linear model based on air temperature and two parameters that represent the linear sensitivity of respiration to air temperature ($\alpha$) and the baseline respiration ($\beta$), as defined in Mahadevan et al. (2008).

$$R = \alpha \times \mathrm{T_{2m}} + \beta \tag{3}$$

The land cover data used by the VPRM were derived from the MapBiomas data (Souza Jr et al., 2020). The VPRM parameters ($\lambda$, PAR0, $\alpha$, $\beta$) were optimized against flux tower NEE for the main land cover type over the study domain described in section 2.2.2.

### 2.1.3 Meteorological data

Meteorological data from the São Paulo State Environmental Protection Agency (CETESB) air quality network were used to evaluate the model's performance in simulating meteorological fields. CETESB manages automatic and manual air quality stations over São Paulo state. These stations provide hourly information on meteorological and pollutant parameters, such as air temperature, wind speed, and wind direction (Table 2), as well as the concentration of air pollutants. Monitoring follows instrumentation standards and directives from the Environmental Protection Agency (US EPA) and the World Health Organization (WHO) respectively for air pollutants, and from the World Meteorological Organization (WMO) for meteorological variables (CETESB, 2019). The air quality and meteorological data are continuously published on the Qualar website (https://qualar.cetesb.sp.gov.br/qualar/). This study used data from four stations located in the MASP (Figure 1): Parque D. Pedro II, PDJ, Guarulhos, and Pinheiros. Table 2 provides the location of the sites, the classification type of the stations, the observed variables, and the data source.

**Table 2.** Location of the sites used for the model evaluation of the meteorological drivers, together with a list of the meteorological variables included in the analysis.

| Sites | Location | Classification | Variables | Source Data |
|---|---|---|---|---|
| Parque D.Pedro II | 23.54S, 46.63W | Urban | $T_{2m}$, WD, WS | CETESB |
| PDJ | 23.45S, 46.76W | Park | $T_{2m}$, WD, WS and $CO_2$ | CETESB/ METROCLIMA |
| Guarulhos | 23.46S, 46.52W | Urban | $T_{2m}$, WD, WS | CETESB |
| Pinheiros | 23.46S, 46.70W | Urban | $T_{2m}$, WD, WS and CO | CETESB |
| IAG | 23.55S, 46.73W | Urban Park | $CO_2$ | METROCLIMA |

Note: Air temperature at 2 m ($T_{2m}$), wind speed (WS), and wind direction (WD).

### 2.2 $CO_2$ observational data

### 2.2.1 Ground-based observations

We assessed near-surface model performance using $CO_2$ observations from the METROCLIMA network in São Paulo (see Table 3 and Figure 1), the first conventional in situ greenhouse gas measurement network established in South America (www.metroclima.iag.usp.br). The network comprises four continuously operating monitoring stations, all located within the

MASP and equipped with cavity ring-down spectroscopy instruments (Picarro) that measure the concentrations of $CO_2$ following the directives from WMO. The monitoring stations are located at various locations within MASP: in a vegetated area at the extreme west (Pico do Jaraguá, PDJ); in a suburban area in the center-west, inside the campus of the University of São Paulo (IAG); at the top of a 100 m building (ICESP); and in an urban area in the east zone characterized by heavy traffic in the neighborhood (UNICID). However, we only used data from the IAG and PDJ sites, which are 13 km apart, as these were the only two stations monitoring $CO_2$ during the selected study period, prior to the Covid-19 pandemic (Souto-Oliveira et al., 2023).

**Table 3.** Description of the METROCLIMA monitoring stations utilized in this study.

| Station | Instrument | Inlet elevation (m) | Altitude (m) |
|---------|------------|---------------------|--------------|
| PDJ | G2301 II | 3 | 1079 |
| IAG | G2301 II | 15 | 731 |

### 2.2.2 $CO_2$ fluxes data and VPRM optimization

In this study, the VPRM model computed the biosphere fluxes for 5 different plant functional types (PFT), representing different vegetation land covers, and for that required a set of four model parameters for each vegetation class, dependent on the region of interest. Ideally, these parameters are optimized using a network of eddy flux towers for each PFT over the domain. The VPRM parameters were optimized for only three PFT corresponding to the three ecosystems observed by eddy-covariance flux towers. However, these three PFT represent almost 60% of land covers over the domain (i.e. sugarcane - 23.86%, Atlantic Forest - 34.86%, and Cerrado - 0.91%). We used a set of parameters optimized by Botía et al. (2022) for the remaining PFT's, such as grasses and mixed forest, based on measurements from sites in the Amazon region in Brazil, deployed in the context of the Large Scale Biosphere-Atmosphere Experiment (LBA-ECO) (Botía et al., 2022). The methodology for optimizing the VPRM parameters for the Atlantic Forest used data from Serra do Mar State Park in São Paulo State, Brazil (23°17'S, 45°03'W at 900 m altitude) for the period from January 2015 to December 2015 (Freitas, 2012). For Cerrado, we used observed data from Pé Gigante, in São Paulo, Brazil (21°36'S, 47°34'W at 660m) from January 2015 to January 2017 (Rocha et al., 2002). For sugarcane we used data from the municipality of Pirassununga, in São Paulo State, Brazil (21°57'S, 47°20'W at 655 m altitude) for the period from November 2016 to August 2017 (Cabral et al., 2020). The VPRM parameters were optimized separately for each PFT using half-hourly observed fluxes from the flux towers over the entire observation periods. We optimized the parameters for the GEE and R simultaneously, and for the default VPRM parameters we used non-linear least squares minimization between the modeled NEE and the flux tower estimation of the observed NEE. In the optimization, the VPRM model is driven by the meteorological measurements of the sites and their specific land covers. The vegetation indices (EVI and LSWI) were derived from the product MOD09A1 of MODIS at 500 m resolution and 8-daily frequency using Google Earth Engine.

### 2.2.3  X$CO_2$ satellite observations

Satellite-based X$CO_2$ observations were utilized in addition to surface $CO_2$ measurements over the study domain. OCO-2, NASA's inaugural Earth remote sensing satellite dedicated to atmospheric $CO_2$ observations, was launched in 2014 (Crisp, 2015). Operating on a solar synchronous orbit, OCO-2 conducts global measurements of $CO_2$ absorption and emission at 13:30 Local Solar Time. The OCO-2 observation data utilized were ACOS L2 Lite Output Filtered with oco2-lite_fle_prefilter_b9, which were converted from Level 1 radiance to Level 2 data using the ACOS retrieval algorithm developed by O'Dell et al. (2012). Data quality assessment for OCO-2 observations can be performed using the xco2_quality_flag and warn_level parameters, as detailed in the OCO-2 Data Product User's Guide (Osterman et al., 2018). In this study, we considered only OCO-2 data with a '0' xco2_quality_flag value that indicates "good" quality. Initially, simulated $CO_2$ concentrations were interpolated to match the latitude, longitude, horizontal resolution, and vertical levels of OCO-2 data. Additionally, to ensure consistency in the comparison, the simulated data were selected to correspond as closely as possible to the OCO-2 overpass time (13:30 Local Solar Time) over the study region. Due to the difference in data types and units between the simulated $CO_2$ concentrations and observed X$CO_2$ from satellites, a conversion was necessary prior to comparison. Consequently, $CO_2$ concentrations simulated at each pressure level in the WRF-Chem were transformed into X$CO_2$ concentrations following the methods by Connor et al. (2008) and O'Dell et al. (2012), as follows:

$$XCO_2^{\text{model}} = XCO_{2\text{a}} + \sum_i w_i^T A_i (CO_2^{\text{interp}} - CO_{2\text{a}})_i \tag{4}$$

where X$CO_{2\text{a}}$ is a priori X$CO_2$, $w_i^T$ is the pressure weighting function, $A_i$ is the column averaging kernel, $CO_2^{\text{interp}}$ is the interpolated simulated $CO_2$ concentrations of WRF-Chem, and $CO_{2\text{a}}$ is a priori $CO_2$.

### 2.3  Evaluation metrics

Several statistical metrics are available for assessing the effectiveness of atmospheric models. These include mean bias error (bias, Equation A1), indicating the average difference between the simulation and the observation; root-mean-square error (RMSE, Equation A2), which quantifies the square root of the average squared deviation between simulation and observation; and the correlation coefficient ($R^2$, Equation A3), representing the degree and direction of the linear connection between the simulation and the observation. To evaluate the model performance, we calculated the bias, RMSE, and $R^2$, with the corresponding equations provided in Appendix A.

## 3  Results

Hourly simulations were conducted from 1 February to 31 August 2019, with each month simulation including a five-day spin-up period. In the following sections, the performance of meteorological drivers will be first presented, followed by the terrestrial surface $CO_2$ fluxes and atmospheric $CO_2$ concentrations from the IAG and PDJ stations. These measurements were

used to evaluate the model performances and to assess the local impacts of the main $CO_2$ sources and sinks on atmospheric $CO_2$ concentrations.

## 3.1 Model performance for meteorological drivers

The assessment of the meteorological model performances is essential for accurately simulating greenhouse gas concentrations. In this study, the model represented the temporal variability and trends of 2-meter temperature ($T_{2\mathrm{m}}$), 10-meter wind speed (WS), and direction (WD) throughout the simulation period, as illustrated in Figure 2 and the supplementary material. The WRF-Chem model effectively captured significant changes in the observed variables, although it failed to accurately represent the maximum and minimum peaks, particularly for wind speed. The simulated 2-meter temperature tended to overestimate values at specific sites, such as Parque D. Pedro II (bias = 0.5°C), Guarulhos (bias = 0.1°C) (see figure B4a and B5a in Appendix B), and PDJ (bias = 0.7°C) (see Figure 2a). However, at the Pinheiros station, the simulated surface temperature was underestimated (bias = -0.7°C) (Figure B6a in Appendix B).

In terms of biases, the model overestimated the wind speed at all sites (bias < 1.5 m$s^{-}$1), with PDJ exhibiting the highest mean bias (1.4 m$s^{-}$1). This overestimation could be attributed to the model's misrepresentation of land use, leading to elevated wind speeds in areas classified as urban rather than vegetated. Notably, numerical models tend to lack sensitivity in simulating very low-velocity speeds due to imperfections in land surface processes and the model's ability to accurately resolve topographical features (Shimada et al., 2011; Zhang et al., 2009; Vara-Vela et al., 2018, 2021). The model's wind directions showed sufficient sensitivity, aligning accurately with observed values. Both the model and observations indicated that prevailing winds were predominantly from the southeast. In summary, the WRF model showed proficiency in reproducing atmospheric conditions in the study area, particularly concerning air temperature and wind direction, with similar performances as previous studies (Feng et al., 2016; Deng et al., 2017).

## 3.2 The VPRM Model: Evaluation with Flux Tower Data

The optimization results are shown in Table 4. Substituting alpha and beta back into the respiration equation led to a better model representation of NEE compared to NEE values simulated with default parameters (Mahadevan et al., 2008) for the main PFT across the domain.

The optimized VPRM parameters for the Atlantic Forest exhibited the greatest discrepancies compared to other vegetation classes. The geomorphological characteristics of the Atlantic forest differ from those of the evergreen forest studied by (Mahadevan et al., 2008), where the default parameters (VPRM_default, represented by the red curve in Figure 3) were used. The optimized VPRM parameters (VPRM_optimized, shown as the green curve in Figure 3) more accurately captured the seasonal cycle in the daily average NEE for the three PFTs optimized in this study. The model was particularly successful in capturing the seasonal profile for the agricultural ecosystem, which can be attributed to the more pronounced seasonal transitions of sugarcane (as indicated by the EVI), even though the low-resolution satellite indices do not fully capture the onset of the growing season. However, this allowed the model to better represent the GEE equation for this ecosystem. For the Cerrado, the model smoothed the NEE peaks, and the GEE and respiration equations were also smoothed with the optimization. Optimizing the

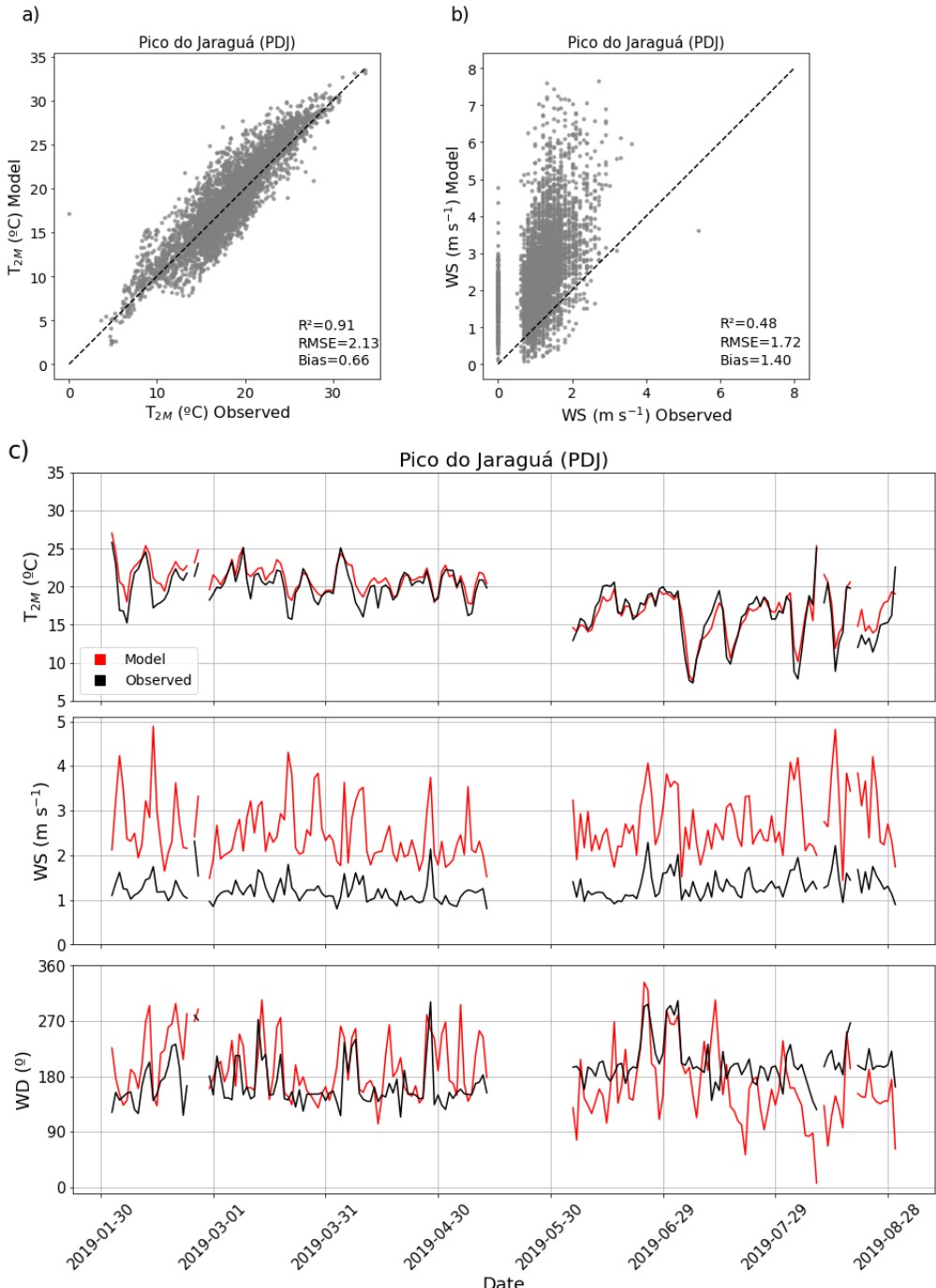

**Figure 2.** Panels (a) and (b) show scatter plots comparing model outputs and observations at the PDJ station for hourly values of 2m air temperature ($T_{2m}$) and 10 m wind speed (WS), respectively. Panel (c) presents the daily averages from February to August 2019 for 2m air temperature ($T_{2m}$), 10 m wind speed (WS), and wind direction (WD). The black line represents observational data, while the red line indicates model simulations.

**Table 4.** Default (Mahadevan et al., 2008) and Optimized VPRM parameters (highlighted) for Atlantic Forest, Cerrado and sugarcane, and for mixed forest and grasses from Botía et al. (2022).

| Type of Vegetation (PFTs) | Default | | | | Optimized & Botía et al. (2022) | | | |
|---|---|---|---|---|---|---|---|---|
| | PARo | $\lambda$ | $\alpha$ | $\beta$ | PARo | $\lambda$ | $\alpha$ | $\beta$ |
| **Atlantic Forest** | **570** | **0.127** | **0.271** | **0.250** | **178615** | **0.008** | **-0.211** | **4.715** |
| Mixed forest | 629 | 0.123 | 0.244 | 0.240 | 206 | 0.255 | 0.342 | 0.000 |
| Grasses | 321 | 0.122 | 0.028 | 0.480 | 15475 | 0.056 | 0.312 | 7.337 |
| **Cerrado** | **3241** | **0.057** | **0.012** | **0.580** | **2300** | **0.616** | **0.070** | **1.665** |
| **sugarcane** | **2051** | **0.200** | **0.209** | **0.802** | **14550** | **0.049** | **-0.339** | **10.052** |
| Urban area | 0.0 | 0.0 | 0.0 | 0.0 | 0.0 | 0.0 | 0.0 | 0.0 |

VPRM parameters improved the representation of the growing season, especially for the Atlantic Forest and sugarcane, while
using optimized or default parameters for the Cerrado resulted in similar NEE simulation.

The first panel in Figure 4 shows the monthly net $CO_2$ flux simulated by the VPRM model for 2019. February represents a
summer month, while August represents a winter month. The second panel shows the monthly hourly net $CO_2$ flux simulated
at the three flux tower sites used to optimize the VPRM model parameters. In February, negative NEE values are found in the
northern part of the MASP, while the southern part exhibits positive NEE fluxes in the coastal region. During the summer,
ecosystem productivity is expected to peak across all land cover classes, typically resulting in negative NEE. This behavior
was clearly observed in February (Figure 4a) for Cerrado, sugarcane, and pasture areas. In contrast, the Atlantic Forest in the
southwestern portion of the domain exhibited positive NEE values, an unexpected pattern for a summer month. This may be
linked to a combination of structural and anthropogenic factors, as well as limitations of the model itself. The Atlantic Forest is
marked by structural heterogeneity, extreme biodiversity, and high fragmentation, which can lead to significant local variation
in $CO_2$ fluxes. In addition, the SEEG (2021) report highlights a progressive decline in the biome's carbon sink function. Model
limitations also likely contribute to these discrepancies, particularly simplifications in VPRM's equations of respiration and
phenology, which may not fully capture the complex dynamics of ecosystems like the Atlantic Forest (Rezende et al., 2018;
Segura-Barrero et al., 2025).

In August, the cold and dry conditions, due to reduced solar radiation and a lower leaf area index, resulted in positive fluxes
across most of the domain and low negative fluxes in only a few areas (Figure 4b). The highest positive NEE values are found in
the southern coastal region. Generally, larger areas with negative $CO_2$ fluxes are observed in February compared to August for
the same dominant land cover classes. This indicates greater $CO_2$ absorption by agriculture in February compared to forested
regions. Conversely, in August, $CO_2$ fluxes are predominantly lower and negative across most of the domain, with higher
positive values in the coastal area, especially in the south. Overall, the domain acts as a net $CO_2$ sink during summer, while
vegetation becomes a $CO_2$ source in winter, except for the Atlantic Forest in the southern part of the study area. The second
panel also shows simulated fluxes for the same flux tower sites, with negative net fluxes in February, particularly in the Atlantic
Forest, sugarcane, and Cerrado. This underscores the reduction in negative fluxes during winter, as seen in the August data for

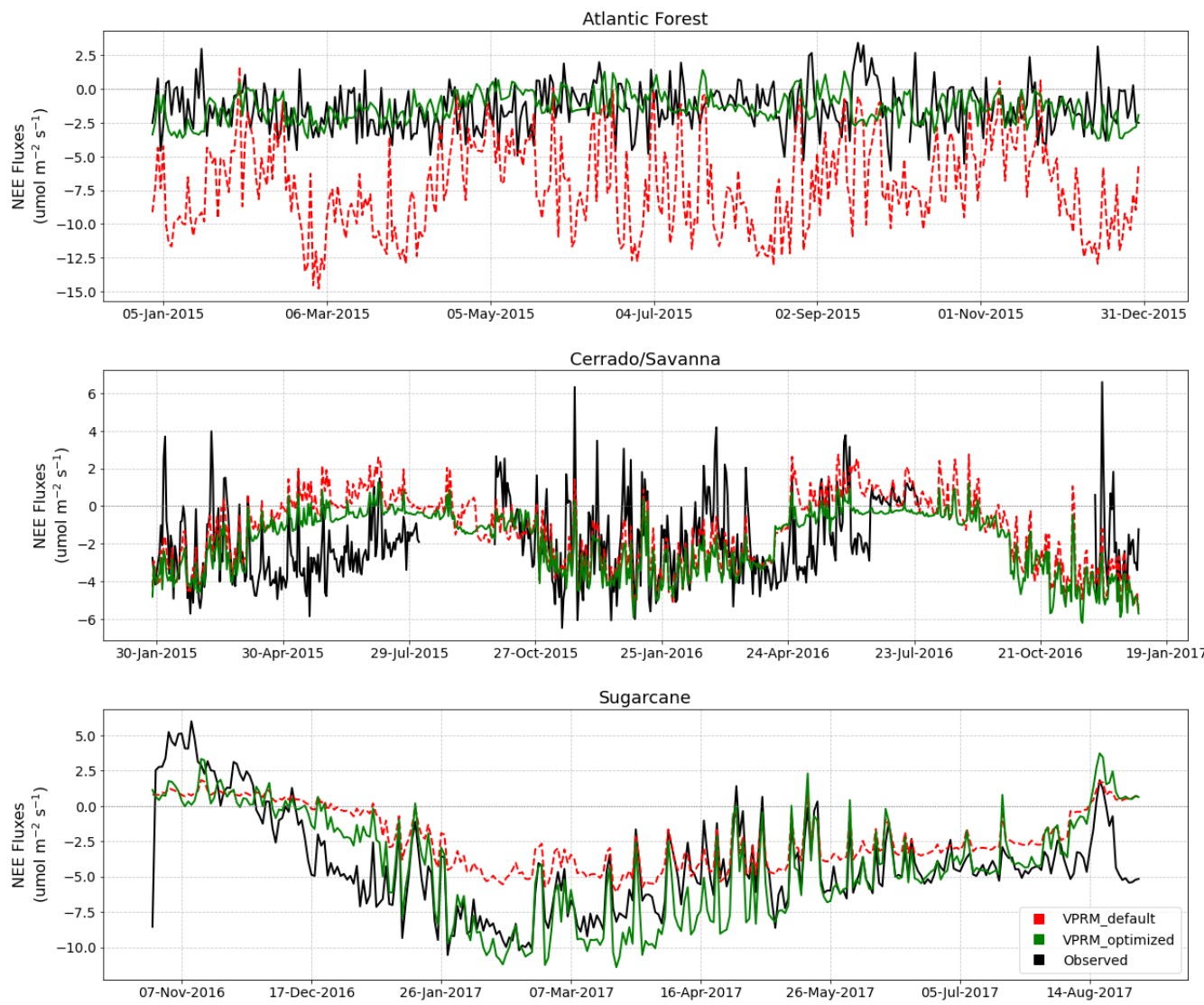

**Figure 3.** Daily variability of NEE fluxes ($\mu mol\, m^{-2}\, s^{-1}$) from the flux tower (black line), compared with NEE fluxes simulated by the VPRM model using default (red line) and optimized (green line) parameters for the Atlantic Forest, Cerrado/Savanna, and sugarcane.

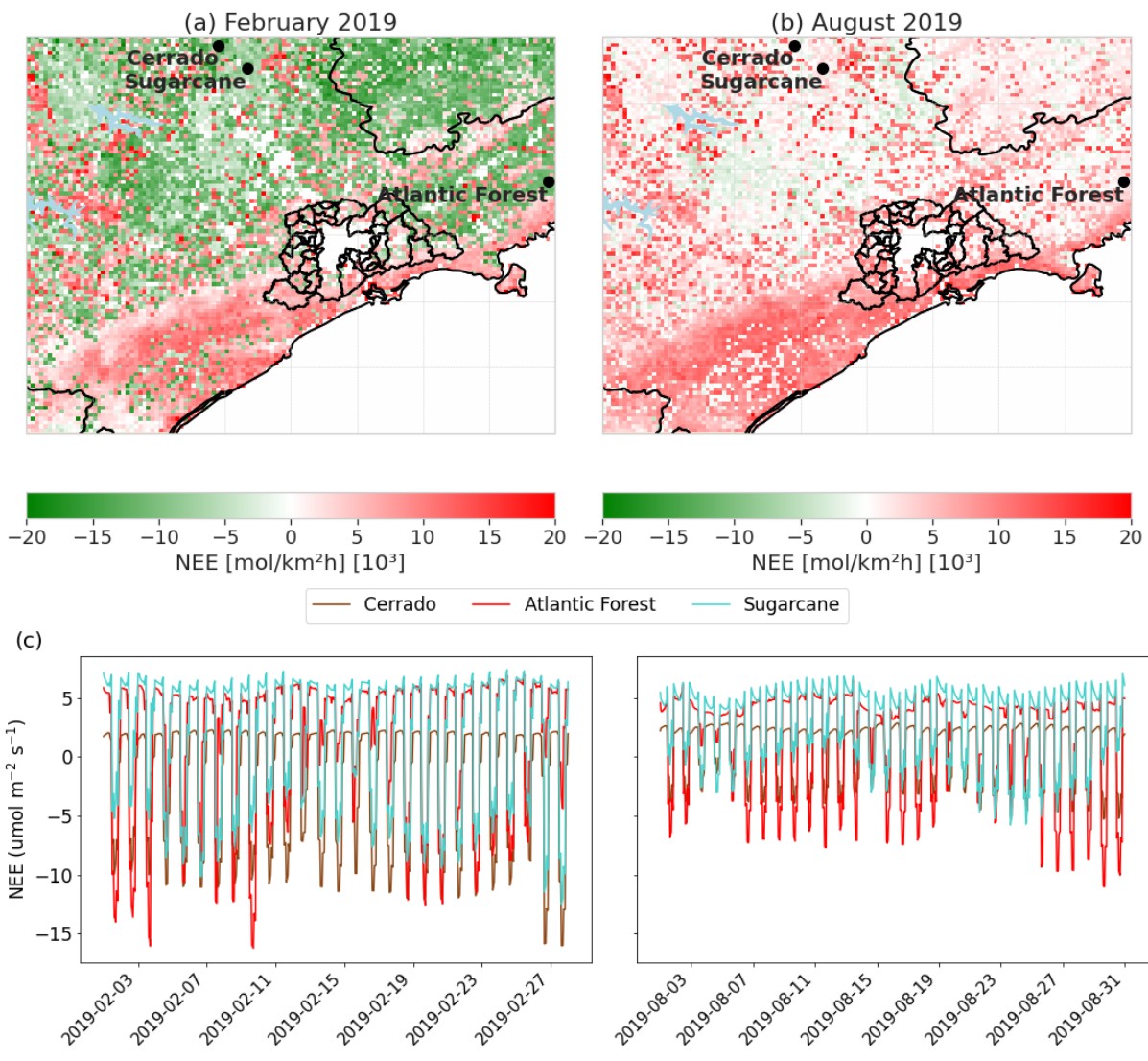

**Figure 4.** The first panel shows the monthly mean of net ecosystem exchange (NEE) ($mol\,km^{-2}\,h^{-1}$) for February (a) and August (b) 2019. The second panel (c) presents the hourly variability of NEE ($\mu mol\,m^{-2}\,s^{-1}$) for the same months (February and August) at three different PFTs: Atlantic Forest, Cerrado/Savanna, and sugarcane.

all three vegetation types. Unfortunately, observed data from these flux towers for this period were not available for statistical model evaluation. However, Figure 4 illustrates the significant influence of climatic drivers on reduced flux trends, consistent with findings by Raju et al. (2023) for a tropical region. Note that the respiration equation in Mahadevan et al. (2008) is a simple linear function of temperature and does not account for seasonal or spatial variability in biomass and litter inputs to soil carbon pools Gourdji et al. (2022), which is particularly relevant for forest ecosystems like the Atlantic Forest.

## 3.3 Seasonal variations in observed and modeled $CO_2$ mixing ratios

Figure 5 and Table 5 depict the monthly mean, standard deviation, bias and RMSE of $CO_2$ concentrations at two sites in the MASP. In 2019, the IAG station recorded $CO_2$ values ranging from 406 to 464 ppm. The seasonal variation peaked during winter (June to August, 437.3 $\pm$ 32.2 ppm), followed by autumn (March to May, 433.0 $\pm$ 26.0 ppm), with the lowest values observed in summer (February, 432.7 $\pm$ 24.6 ppm). This variation in $CO_2$ levels is primarily influenced by the geographical location of the observation site, as well as meteorological conditions such as wind speed and atmospheric stability, and seasonal patterns of photosynthesis and vehicular traffic (see Figure B1 in Appendix B). The maximum and minimum monthly $CO_2$ concentrations at IAG were recorded in June (442.5 $\pm$ 32.8 ppm), during the winter season, and March (430.2 $\pm$ 24.5 ppm), during the autumn season, respectively. During this month, the MASP experiences changes in synoptic circulation and atmospheric moisture that typically reduce atmospheric stability and increase the dispersion of various gases and particles (Chiquetto et al., 2024). Meanwhile, at the PDJ station, $CO_2$ levels ranged from 414 ppm to 417 ppm. The seasonal variation peaked during autumn (416.8 $\pm$ 9.5 ppm), closely followed by summer (416.0 $\pm$ 10.3 ppm), with the lowest values observed in winter (414.6 $\pm$ 7.4 ppm). The maximum monthly $CO_2$ mean at PDJ was identified in May (417.3 $\pm$ 9.1 ppm), corresponding to the autumn season, while the minimum was recorded in July (414.0 $\pm$ 6.3 ppm), during the winter season. Monthly values at PDJ exhibited less variability and a smaller standard deviation compared to the IAG site. This result was expected, considering that the IAG site is significantly impacted by vehicular traffic in its vicinity. In contrast, PDJ is located at a higher elevation in a more vegetated area, with less influence from local anthropogenic sources. Additionally, lower $CO_2$ concentrations were expected at PDJ during the summer due to the stronger vegetation signal compared to the IAG site. However, PDJ actually shows peak $CO_2$ levels in summer and the lowest values in winter, indicating that additional ecological and ecosystem variables need to be considered for a better understanding of this location.

The simulated $CO_2$ concentrations for the IAG station ranged from 410 ppm to 437 ppm, with a seasonal variation peaking in winter (429.4 $\pm$ 19.2 ppm), followed by autumn (425.2 $\pm$ 15.1 ppm), and the lowest values occurring in summer (422.3 $\pm$ 12.3 ppm), mirroring the observed data. Notably, the highest and lowest monthly $CO_2$ concentrations at IAG were identified in June (438.7 $\pm$ 22.5 ppm) and February (418.1 $\pm$ 10.0 ppm), respectively. Although the maximum monthly value from the model coincided with the observed data, the month with the minimum concentration was February, which may be attributed to gaps in measurement, which were not considered when calculating the mean, thereby influencing the observed monthly mean. The $CO_2$ concentrations at PDJ ranged from 415 ppm to 426 ppm, with seasonal variation peaking in winter (421.8 $\pm$ 11.8 ppm), followed by autumn (420.4 $\pm$ 10.1 ppm), and the lowest values occurring in summer (419.0 $\pm$ 8.8 ppm). The model data profile for PDJ more closely resembles the simulated IAG profile than the PDJ station's observed profile, which

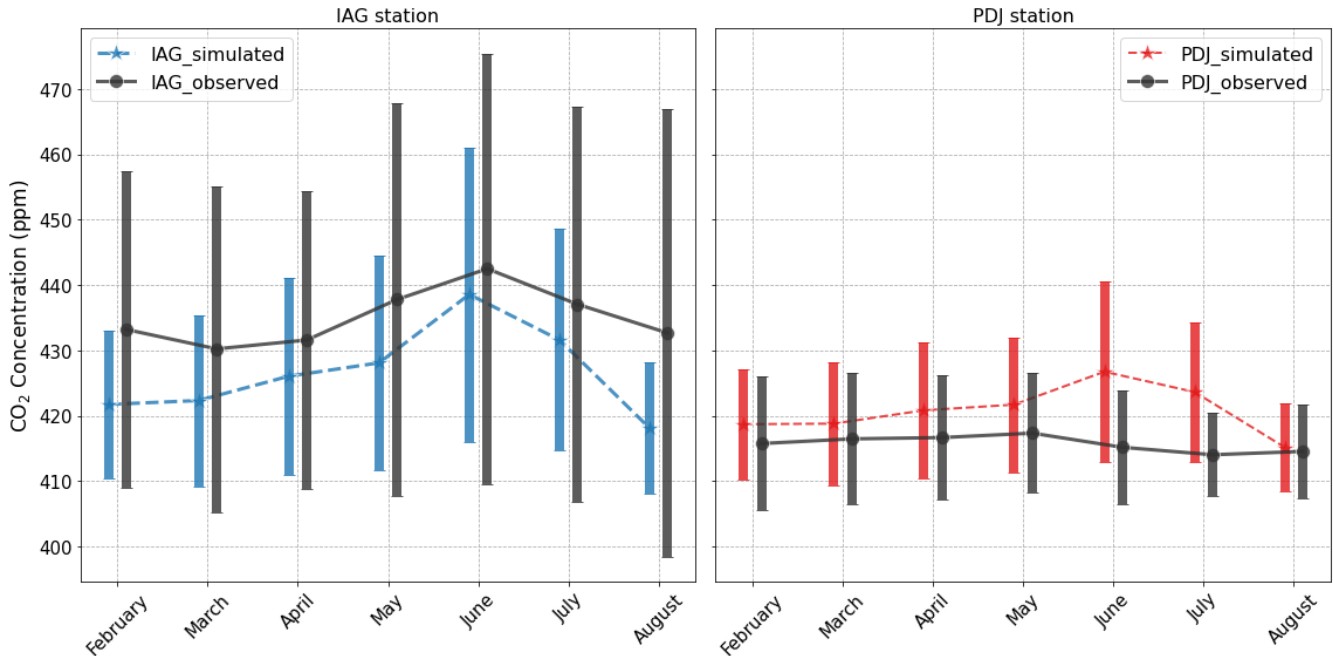

**Figure 5.** $CO_2$ concentration seasonality observed and simulated at IAG and PDJ stations in 2019. Error bars represent the monthly standard deviation.

likely stems from model limitations, including grid resolution and insufficient representation of localized characteristics at
different sites. However, negative biases were observed for all seasonal periods at IAG, indicating an underestimation of $CO_2$
concentrations and higher RMSE compared to the statistics for the PDJ station. The PDJ station exhibited low positive biases
and smaller standard deviations between the model and observations. Its higher elevation and dense vegetation cover simplify
the representation of seasonal trends, reducing the influence of urban emissions and resulting in lower $CO_2$ concentrations at
this site (see Figure B7 in Appendix B).

**Table 5.** Seasonality means and standard deviation of $CO_2$ concentrations for IAG and Pico do Jaraguá (PDJ) stations.

| Station | Season | Observed (ppm) | Simulated (ppm) | Bias (ppm) | RMSE (ppm) |
|---------|--------|----------------|-----------------|------------|------------|
|         | Summer (February) | $432.7 \pm 24.6$ | $422.3 \pm 12.3$ | -12.1 | 25.2 |
| IAG     | Autumn (MAM) | $433.0 \pm 26.0$ | $425.2 \pm 15.1$ | -7.5 | 24.8 |
|         | Winter (JJA) | $437.3 \pm 32.2$ | $429.4 \pm 19.2$ | -7.2 | 31.1 |
|         | Summer (February) | $416.0 \pm 10.3$ | $419.0 \pm 8.8$ | 3.6 | 11.1 |
| PDJ     | Autumn (MAM) | $416.8 \pm 9.5$ | $420.4 \pm 10.1$ | 3.6 | 12.0 |
|         | Winter (JJA) | $414.6 \pm 7.4$ | $421.8 \pm 11.8$ | 7.3 | 13.8 |

### 3.3.1 Distribution of surface $CO_2$ concentrations

In addition to the simulations conducted for the period from February to August 2019, using the same configurations and input data, we performed simulations involving variable emission scenarios for the summer (February) and winter (August) seasons. The aim was to comprehensively understand the dynamics of $CO_2$ concentration in the metropolitan region and surrounding areas during these distinct seasonal periods. Figure 6 shows the monthly average spatial distributions of simulated $CO_2$ concentrations under four conditions: a) Background without emissions, considering only boundary and initial conditions (BCK); b) considering both anthropogenic emissions and biogenic fluxes (see Table 1) (ALL); c) considering biogenic fluxes only (BIO); and d) considering anthropogenic emissions (energy, industry, residential, refinery, and vehicular sectors) only (ANT).

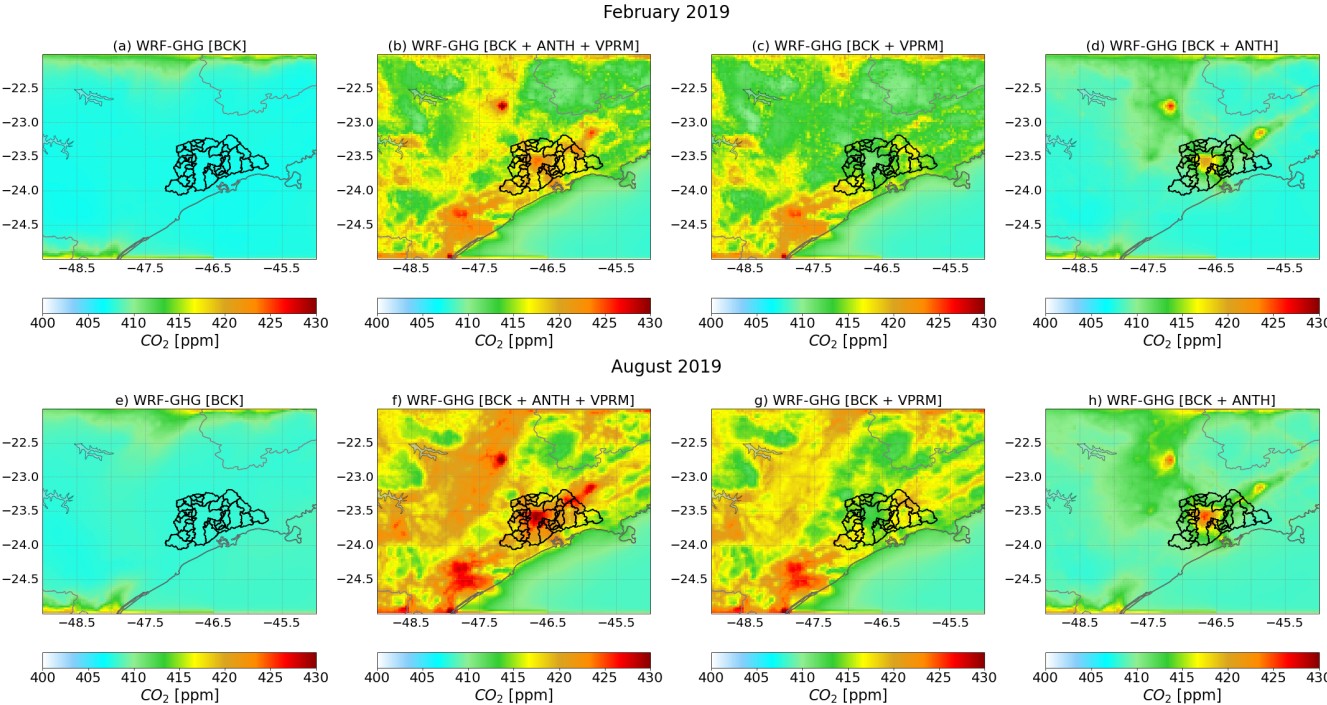

**Figure 6.** Atmospheric $CO_2$ concentrations under different emission scenarios (refer to the text). The panels in the first row represent the monthly mean concentration for February (a, b, c, d), while the panels in the second row represent the monthly mean concentration for the August period (e, f, g, h). Panels a) and e) represent the background scenario. Panels b) and f) represent simulation of total (background, anthropogenic and biogenic) emissions scenario, panels c) and g) represent simulation of only background and biogenic scenario, and d) and h) represent simulation of only background and anthropogenic scenario.

Figure 6a shows that the simulated background $CO_2$ concentration in February ranged around 408 ppm across most of the domain. For biogenic simulations (Figure 6c), we observed an average increase of 14 ppm across the domain compared to the previous simulation. The increase, however, was only 6 ppm in downtown MASP. Although the VPRM model did

not explicitly calculate CO2 fluxes in urban areas due to limited vegetation coverage, the transport of biogenic signals from the surrounding vegetated regions into the urban area is evident. The southwest region of the domain, characterized by the Atlantic Forest, exhibits the highest $CO_2$ concentrations in this scenario, ranging from 420 to 424 ppm. This dense vegetation region and higher ecosystem respiration contribute to elevated $CO_2$ levels, underscoring the influence of biogenic sources on regional concentration patterns. This region has altitudes lower than 200 m and the $CO_2$ released to the atmosphere by the vegetation is trapped due to the Serra do Mar, with altitudes higher than 500 m. The Atlantic Forest present on the northern coast, on the other hand, is concentrated on the plateau of Serra do Mar, and thus, the $CO_2$ released is better dispersed to other areas. The simulation with anthropogenic emissions (Figure 6d) stands out elevated $CO_2$ concentrations over the center of the city of São Paulo, characterized by high vehicle emissions, as well as over other two urban areas in the north and northeast of MASP. The monthly mean $CO_2$ concentration in these two urban areas was roughly 420 ppm, attributed to emissions from refineries represented by the EDGAR datasets as well as vehicles. Figure 6b shows the simulated $CO_2$ concentration considering both vegetation fluxes and anthropogenic emissions. As expected, this simulation combines both contributions, resulting in high $CO_2$ concentrations over urban areas and along the coastal region. For August, it can be observed that the background concentrations (Figure 6e) were slightly higher around MASP. Additionally, the monthly mean $CO_2$ concentration for the scenario in August with only biogenic sources was 8 ppm higher than that in February, which can be explained by the lower photosynthetic rates in this period, as observed in Figure 4. The Atlantic Forest in the coastal region exhibits more positive $CO_2$ fluxes and lower photosynthetic activities, characterized by lower amounts of rainfall in the region that contribute to this reduced photosynthetic production by vegetation. The simulation with only anthropogenic emissions (Figure 6h) shows higher $CO_2$ concentrations compared to those in February. This increase in $CO_2$ levels in August is attributed to a lower planetary boundary layer (PBL) height. However, it is important to point out that the EDGAR anthropogenic emission inventory generally overestimates the emissions around local anthropogenic sources (e.g., urban areas) (Seo et al., 2024). The higher simulated $CO_2$ concentration for August compared to February, in the scenario with both biogenic and anthropogenic sources, is largely dependent on factors such as atmospheric stability and meteorological conditions. Atmospheric stability, along with meteorological variables such as humidity, solar radiation, and temperature, plays a crucial role in determining biogenic $CO_2$ concentrations. In addition, under stable atmospheric conditions, such as those often observed during winter periods, $CO_2$ concentrations tend to accumulate near the surface, resulting in higher concentrations, especially in urban areas. Therefore, the comparative analysis between the simulations of $CO_2$ concentrations during summer and winter periods highlights the importance of accurately representing not only anthropogenic emissions, but also biogenic fluxes from vegetation.

### 3.3.2 Evaluation of sources contribution

In Figure 7, we applied a data selection scheme to all-time series to minimize the effects of local contributions and increase the spatial representativeness of each record, it consists of retaining daytime (09–17 h local) data, when the air is well-mixed, providing a large spatial representativeness with minimum influence from local sources (Gerbig et al., 2008; Ramonet et al., 2020). Figure 7, shows the comparison of the daily daytime average $CO_2$ concentrations simulated by the model for February and August 2019, considering both biogenic and anthropogenic sources (see Figures 6b and 6f), at both IAG and PDJ sites.

The left panels (Figures 7a, 7c, 7e, and 7g) depict the simulated $CO_2$ concentration considering both anthropogenic and biogenic sources (all_sources, in gray), alongside observed concentrations (observed, in purple) for both sites. Conversely, the right panels (Figures 7b, 7d, 7f, and 7h) display the different simulations considering anthropogenic and biogenic sources separately to the daily concentration. In Figure 7a, which represents only one summer month with available observational data (February 2019), the model generally underestimated $CO_2$ concentrations. The observed average was 424.0 ppm, while the simulated average was 416.0 ppm an underestimation of approximately 8 ppm. This difference may be partially attributed to the presence of data gaps in the observational data for this site, as only available values were considered when calculating the monthly mean. For the anthropogenic sources the simulation is aligned with the expectations that the emission is dominated by vehicular emissions around this vicinity (Figure 7b). However, on February 23rd, 24th and 25th, there was a distinct peak in the observed $CO_2$ concentrations. This spike is absent in both the all-source and anthropogenic simulations, suggesting that other localized or transient activities, not accounted for in the emissions inventory, may have contributed. This discrepancy likely arises because the inventories assume identical emissions for all days with only hourly variations. As a result, specific events or activities that occur on these particular days are not captured in the simulations. Furthermore, on February 2nd and 22nd, observed $CO_2$ peaks were captured by the model with similar magnitude only when both anthropogenic and biogenic emissions were included.

At the PDJ site, the mean observed and simulated $CO_2$ concentration in February was 414 ppm. The model captures the overall trend and major peaks of $CO_2$ variability during this period, with biogenic contributions more pronounced at PDJ compared to the IAG site (Figure 7d). This higher biogenic influence at PDJ is attributed to its location in a vegetated area and localized in higher altitude than IAG, relatively isolated from vehicular emissions and other anthropogenic sources typical of urban environments, as previously discussed.

In August, characterized by a drier, more stable boundary layer and lower wind speed, observed data for IAG showed an average of 426 ppm (Figure 7e), while with the model showed a monthly average of 413 ppm, resulting in a discrepancy of 13 ppm, i.e. a higher difference compared to February. In terms of the contributions of the sources (Figure 7f), simulations showed similar daily patterns, with a few days where $CO_2$ contributions from biogenic fluxes exceeded those from anthropogenic source. In contrast, for PDJ (Figure 7g), both the observed and simulated monthly average concentrations were 412 ppm. While the model slightly underestimated some days in the month and overestimated others, it generally captured the observed variability. Regarding the source contributions, the model simulation aligned with the observed temporal profile, displaying a more pronounced biogenic signal than at the IAG site, which further emphasizes the significant role of vegetation as a source of $CO_2$ emissions at this location (Figure 7h). Before late August, observed values tended to be higher than the simulations, whereas in the final days of the month, the model overestimated $CO_2$ concentrations. This overestimation is associated with an increase in background concentrations, a pattern also observed at the IAG site during the same period.

The bias and RMSE for each simulation at the IAG and PDJ sites for February and August 2019 are illustrated (see Figure B8 in Appendix B). At IAG, the average bias ranged from -14.31 to -9.17 ppm, while at PDJ it ranged from -3.54 to -0.96 ppm. RMSE values were consistently higher at IAG, exceeding 20 ppm in most scenarios, while PDJ showed lower errors, generally below 12 ppm.

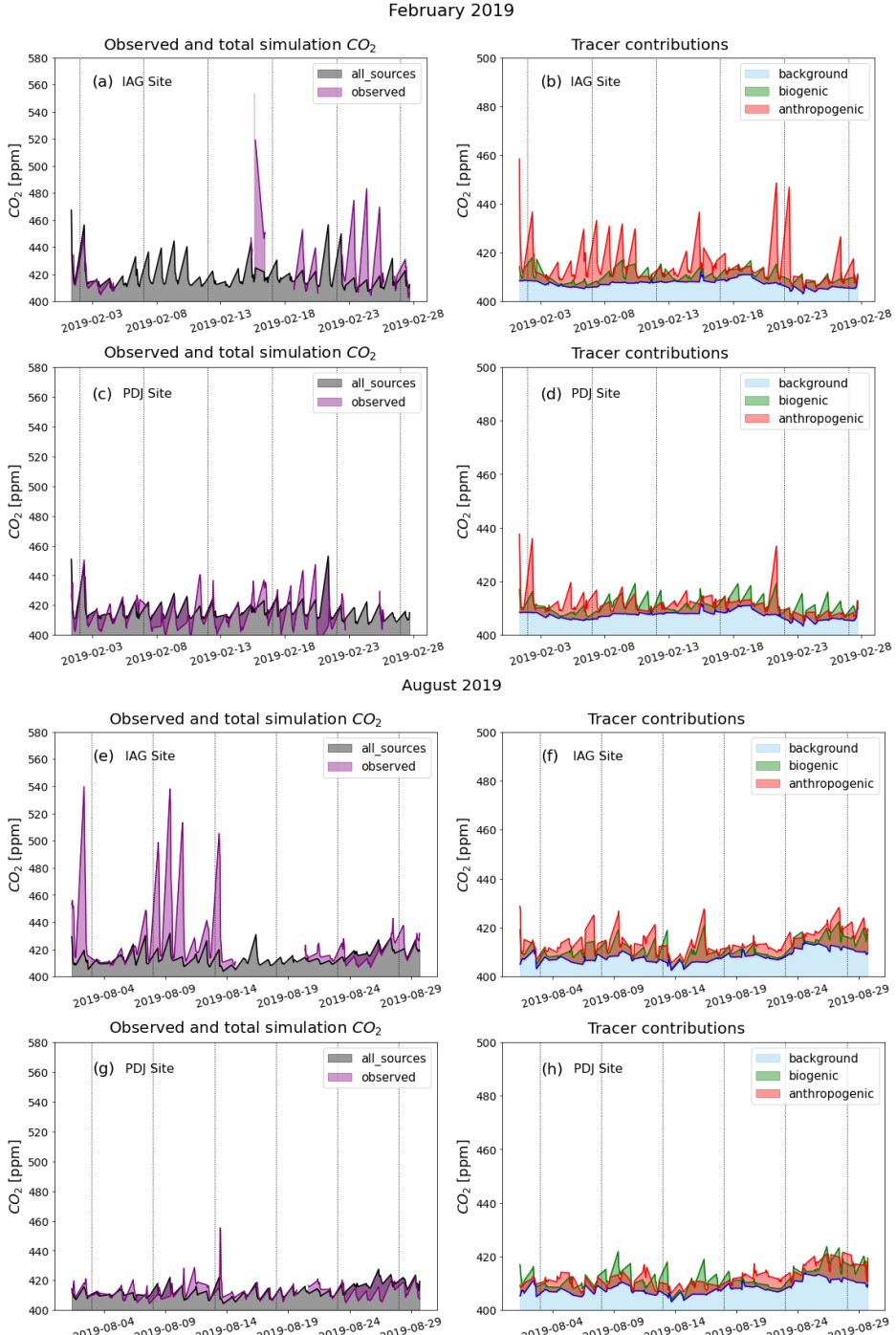

**Figure 7.** Daily mean $CO_2$ concentrations simulated and observed for the IAG site in February 2019 (a), for the PDJ site in February (c), for the IAG site in August (e), and for the PDJ site in August (g). And the daily simulated at the BCK (background), VPRM (biogenic), and ANTH (anthropogenic) scenarios for the IAG site during February (b), for the PDJ site in February (d), for the IAG site during August (f), and for the PDJ site in August (h).

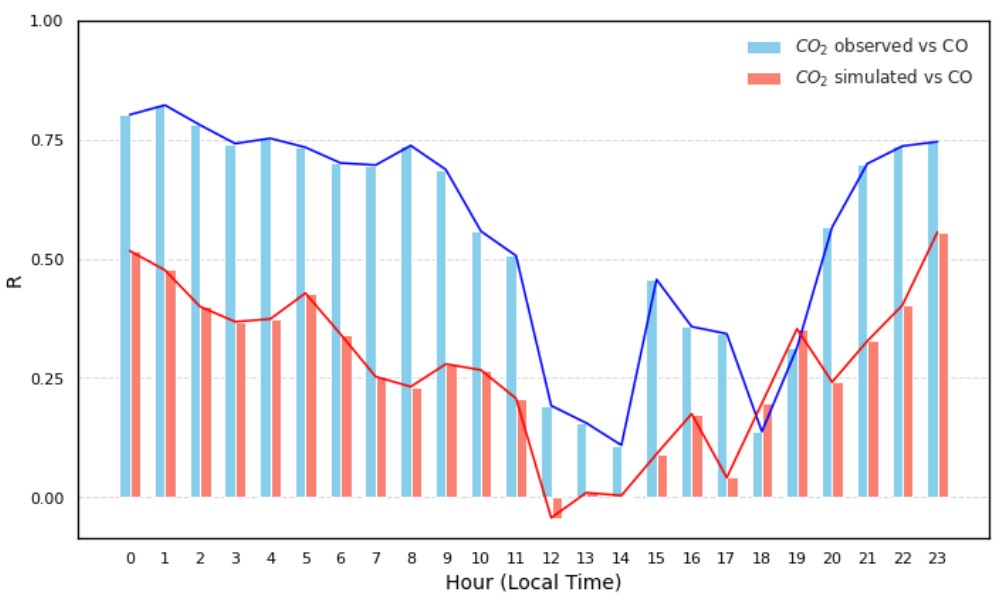

**Figure 8.** Hourly correlation between $CO_2$ concentrations observed at the IAG site and CO concentrations observed at the Pinheiros site (blue bars), and between simulated $CO_2$ concentrations at the IAG site and observed CO concentrations at the Pinheiros site (orange bars) for the period from February to August 2019

Considering that CO serves as a vehicular tracer, we analyzed CO concentrations at the Pinheiros site using data from the

CETESB network (see Figure 1 and Table 1) to compare with $CO_2$ concentration profiles at the IAG site for February to August

2019, located less than 3 kilometers away from the Pinheiros site. The hourly correlation between observed $CO_2$ concentrations

at the IAG site and observed CO concentrations at Pinheiros was determined, along with the correlation between simulated

$CO_2$ concentrations for IAG and observed CO concentrations. In Figure 8, both bar graphs of the hourly correlation between

$CO_2$ and CO concentrations show values above 0.5 for observed $CO_2$ and above 0.25 for simulated $CO_2$ during the early

366    hours of the day (until 10h) and again in the evening (after 19h). Midday, this correlation decreases and even turns negative for

the simulated $CO_2$ vs. CO graph, suggesting the influence of the photosynthesis process on $CO_2$ concentrations, which is also

evident in the observed data. The similarity between the trend lines of the hourly correlation profiles for observed $CO_2$ vs. CO

and simulated $CO_2$ vs. CO is evident.

In addition to the correlation between gases, Figure 9 indicates that both the modeled and observed $CO_2$ profiles suggest

that a significant portion of the $CO_2$ concentrations at the IAG site originates from vehicular sources, as carbon monoxide is

a trace gas associated with traffic emissions (Nogueira et al., 2021). Peaks in the $CO_2$ time series at IAG are observed at the

beginning, where the model fails to capture the magnitude of these concentrations. These peaks also appear in the observed CO

profile at the begin of the month, confirming that a large part of the $CO_2$ concentrations at IAG comes from vehicular sources,

particularly on days with high concentrations, which are also reflected in the CO profile. However, the model struggles to

simulate this high $CO_2$ concentrations since it assumes that emissions follow the same diurnal variation every day of the

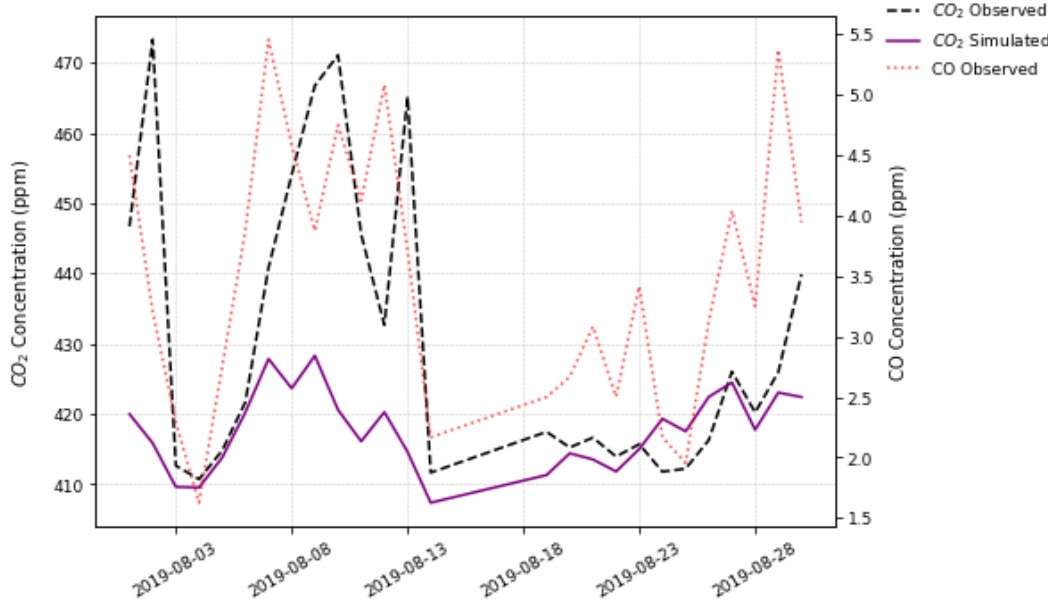

**Figure 9.** Daily mean concentrations of $CO_2$, both observed (black dashed line) and simulated (purple line), at the IAG site, along with observed CO concentrations (red dotted line) at the Pinheiros site during August 2019.

month. Additionally, a distinct increase in CO concentrations without a corresponding rise in $CO_2$ was observed between August 18 and 21 and August 27 and 28, which coincided with the long-range transport of smoke plumes from Amazon forest fires to São Paulo (Bencherif et al., 2020). While biomass burning emits both CO and $CO_2$, their atmospheric transport and dispersion differ significantly. CO is more prevalent in incomplete combustion and tends to be transported at altitudes that favor long-range dispersion, whereas $CO_2$ concentrations are more influenced by local emissions and atmospheric mixing (Gatti et al., 2010). These transport dynamics, combined with the long distance of the event's origin, likely explain why the CO peak was detected at Pinheiros but not accompanied by a significant $CO_2$ enhancement at the IAG site.

### 3.3.3 Model evaluation against OCO-2 and X$CO_2$ observations

Figure 10a shows the monthly boxplots of observed and all_sources simulated X$CO_2$ concentrations for the period from 1 April 2019 to 31 August 2019. However, due to insufficient OCO-2 data over MASP during this period, the analysis covers all simulated domains rather than solely the metropolitan area. Regarding temporal variability, a clear seasonal cycle of X$CO_2$ is evident from its smooth month-to-month variation (green boxes in Figure 10a). The simulated X$CO_2$ concentrations, i.e., the simulated profiles with smoothing, generally captured this cycle, although with a less dispersion (length of the box) compared to the observed X$CO_2$ concentrations. Notably, model-observation discrepancies are most pronounced during the winter months, with differences in median concentrations ranging from 0.8 to 1.5 ppm, while they are minimized during the autumn season,

with differences in median concentrations between 0.5 and 0.6 ppm. The simulated $XCO_2$ concentrations demonstrate similar trends within the same range but tend to slightly underestimate values on most days.

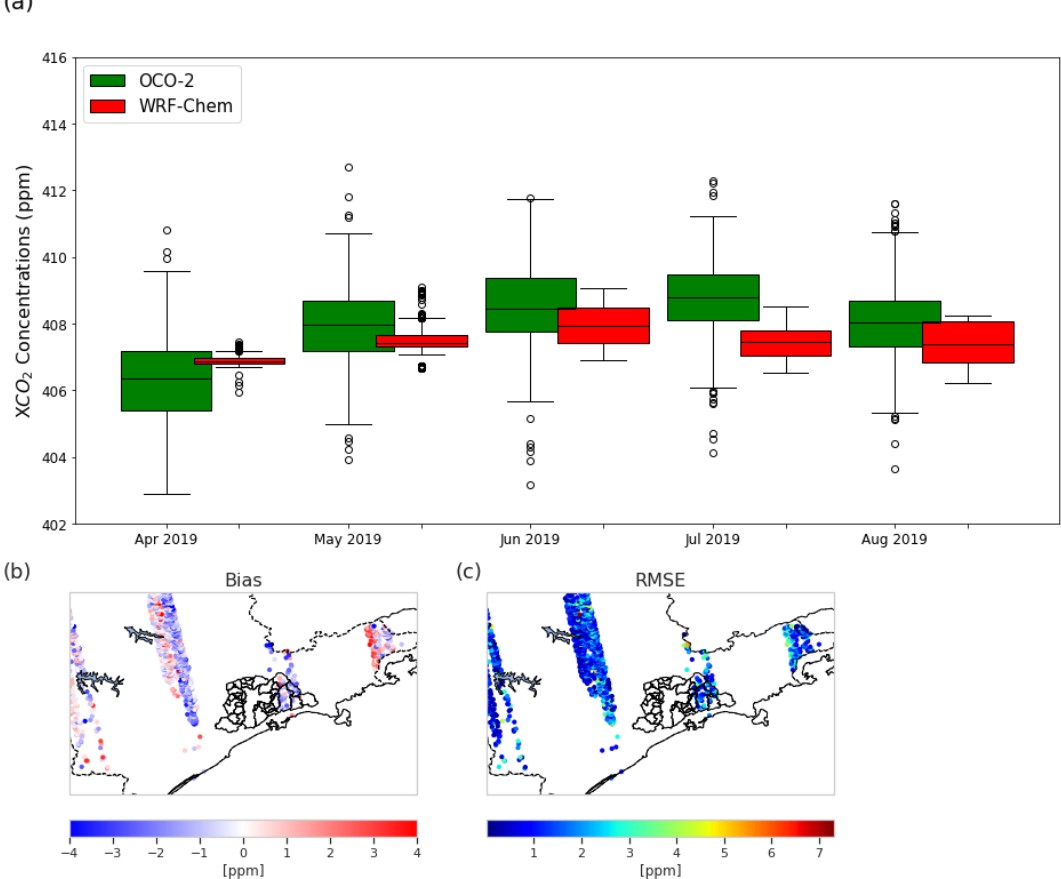

**Figure 10.** a) Monthly boxplots of observed and simulated $XCO_2$ concentrations for the period from 1 April 2019 to 31 August 2019, b) Bias and c) RMSE calculated by pixel over the study domain.

When generating time-averaged modeled values, we only take into account the measurement period as previously mentioned. Regarding $XCO_2$, the smoothed column concentrations (depicted by red dotted lines in Figure B9 in Appendix B) consistently fall below the observed values on a global scale. Figure 10b and 10c depicts the bias and RMSE, respectively, calculated across the pixel-by-pixel domain. Higher RMSE values are evident in the eastern region of MASP and along the border of São Paulo and Rio de Janeiro states. In these areas, characterized by heavy vehicular traffic, the model tends to overestimate $XCO_2$ concentrations. Conversely, for the central region of the domain, we observe slightly negative bias values accompanied by higher RMSE values, indicating an underestimation of $XCO_2$ concentrations.The uncertainties surrounding $XCO_2$ simulation stem from various factors, including potential biases in the model's wind representation, particularly in urban areas, consideration of emissions solely at the surface rather than at different pressure levels, as well as errors in the initial and boundary conditions of

concentration provided by the Carbon Tracker, which has also been seen in other studies (Chen et al., 2019; Lian et al., 2021; Peiro et al., 2022).

## 4   Conclusions

A comprehensive assessment of atmospheric $CO_2$ concentrations in the MASP and its surroundings was conducted, utilizing the WRF-Chem model using the greenhouse gas module. Given the burgeoning demand for research in this domain, particularly in South America, where urban areas are marked by significant emission sources, this study aimed to furnish a broad understanding of the key characteristics of $CO_2$ concentrations. To ensure an accurate estimation of $CO_2$ levels in MASP, the initial focus of the evaluation was on the model's capability to simulate meteorological variables. Biogenic fluxes were derived from the VPRM model, which was fine-tuned with flux tower data. Our results show that using this local data significantly improved simulated biogenic $CO_2$ fluxes, highlighted the model's capacity to represent key seasonal dynamics, with negative NEE values predominating in February (summer) and positive values in August (winter). However, we recommend the deployment of additional flux towers and targeted measurement campaigns to improve the characterization other ecosystems. A more comprehensive representation of PFTs is essential, as vegetation processes play a fundamental role in shaping $CO_2$ patterns in tropical regions. The availability of additional flux tower data would enable a more refined optimization approach, enhancing the characterization of parameters for each vegetation type. Anthropogenic emissions were curated from vehicular model and global inventory to provide a comprehensive representation of urban emissions, incorporating spatial and temporal resolution for key sources such as vehicular traffic for our domain. Boundary and initial conditions were scrutinized using global products. The WRF-Chem model demonstrated skill in simulating meteorological variables, particularly temperature; however, discrepancies in local wind speed and direction persisted. These differences are attributed to the region's complex topography and the model's resolution (3 km), which limits its ability to capture fine-scale dynamical processes.

Simulated $CO_2$ concentrations exhibited distinct diurnal cycles influenced by local emissions, boundary layer dynamics, and vegetation fluxes. The model's performance varied between monitoring stations, highlighting the interplay between urban and vegetative environments. At the IAG site, $CO_2$ concentrations were consistently underestimated, with negative biases of -9.17 ppm in February and -12.83 ppm in August. This underestimation was closely linked to the model's difficulty in capturing the impact of high vehicular emission densities, as indicated by the correlation with CO concentrations. Conversely, at the vegetated and elevated PDJ site, the model closely matched observational data, with minimal biases of 0.73 ppm in February and -0.61 ppm in August. In suburban locations such as the PDJ site, distant from urban sources, anthropogenic emissions diminish, and the vertical gradient of $CO_2$ concentration generated by city emissions attenuates through atmospheric convection and diffusion processes. However, during the growing season, the contribution of biogenic flux to $CO_2$ concentration warrants attention, especially concerning the simulation of nocturnal $CO_2$ concentrations and ecosystem respiration. Improvements in the respiration equation of the VPRM model (Gourdji et al., 2022) could enhance the accuracy of these simulations. Importantly, the modeled $CO_2$ concentrations exhibited high sensitivity not only to atmospheric vertical mixing near the surface but also to the prescribed temporal profiles of anthropogenic and biogenic emissions, highlighting the underestimation of ve-

hicular emissions. These sources of error, particularly pronounced in winter, present challenges in accurately quantifying city emissions.

In general, the WRF-Chem model demonstrated proficiency in simulating seasonal variations, including X$CO_2$, with profiles akin to OCO-2 data. This study underscores the imperative for further investigations and applications of the WRF-Chem model in uncharted regions such as the MASP, showcasing its prowess in simulating meteorological fields and $CO_2$ observations.

*Code availability.* The WRF-Chem model code version 4.0 is freely distributed by NCAR at https://www2.mmm.ucar.edu/wrf/users/download/ (Skamarock et al., 2019). The VPRM code adapted from https://github.com/Georgy-Nerobelov/VPRM-code (Nerobelov et al., 2021). VEIN can be installed from CRAN, and it is also available on Zenodo https://doi.org/10.5281/zenodo.3714187 (Ibarra-Espinosa et al., 2018). Run control files, preprocessing and postprocessing scripts, and relevant primary input/output data sets needed to replicate the modelling results are available upon request from the corresponding author.

*Data availability.* All datasets and model results corresponding to this study are available upon request from the corresponding author.

*Author contributions.* RA performed the simulations and prepared the manuscript with the support of all co-authors. RA and RY design the experiment. TL and RB provided support to set up and run VPRM parameters optimization. OC, MM, and HR provided the observed data used in this work. RA, RY, TL, RB, AV, MA, NR, and CK contributed to the analysis and interpretation of the results.

*Competing interests.* The authors declare that they have no conflict of interest.

*Acknowledgements.* This work was supported by the National Council for Scientific and Technological Development (CNPq) fellowship process number 141962/2019-4, the FAPESP (process number 2016/18438-0 and 2021/11762-5), the French Ministry of Research (Junior Chair professor CASAL) and the Innovation Fund Denmark through the INNO-CCUS project MONICA, the National Institute of Science and Technology – INCT Klimapolis, which is funded by the Brazilian Ministry of Science, Technology, and Innovation (MCTI), and the National Council for Scientific and Technological Development (CNPq) under project number 406728/2022-4.

## Appendix A: Metrics evaluation

$$Bias = \frac{\sum_{i=1}^{N}(pred_i - obs_i)}{N} \tag{A1}$$

$$458 \quad RMSE = \sqrt{\frac{\sum_{i=1}^{N}(pred_i - obs_i)^2}{N}} \tag{A2}$$

$$459 \quad R^2 = \frac{\sum_{i=1}^{N}(pred_i - \overline{pred_i})(obs_i - \overline{obs_i})}{\sqrt{\sum_{i=1}^{N}(pred_i - \overline{pred_i})^2 \sum_{i=1}^{N}(obs_i - \overline{obs_i})^2}} \tag{A3}$$

where $pred_i$ is the model simulation value, $obs_i$ is the observed value, and N is the number of observations.

## 461 Appendix B: Supplementary figures

This appendix contains figures that give some additional insight to the conclusions given in the sections above and are refer-
enced in the text.

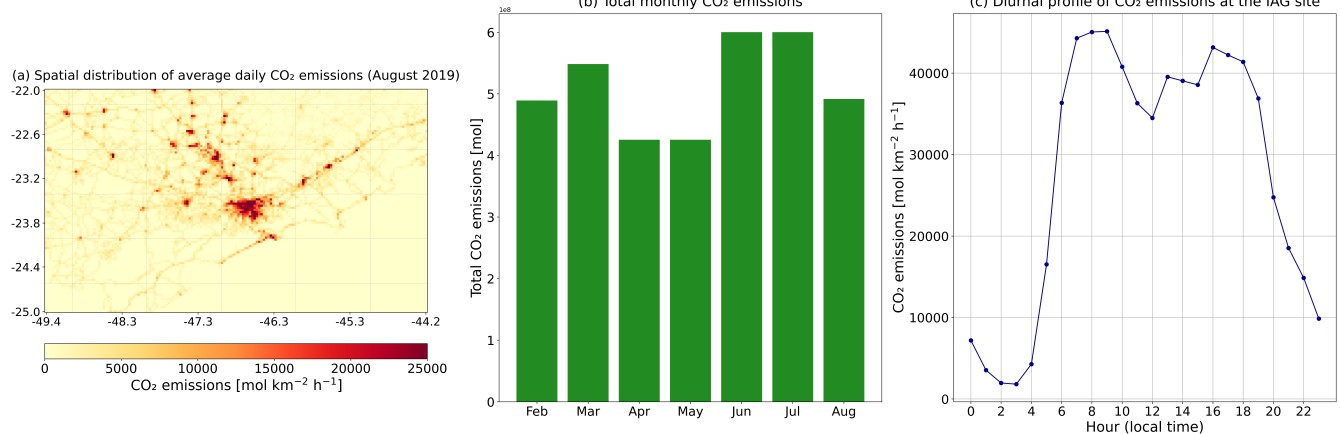

**Figure B1.** Vehicular $CO_2$ emissions as estimated by the VEIN model over the study domain (D01). The panel (a) represents the spatial distribution of average daily $CO_2$ emissions for August 2019 over D01. Panel (b) represents the total monthly $CO_2$ emissions from February to August 2019 over the D01. Panel (c) shows the diurnal profile of CO2 emissions at the IAG site during August 2019.

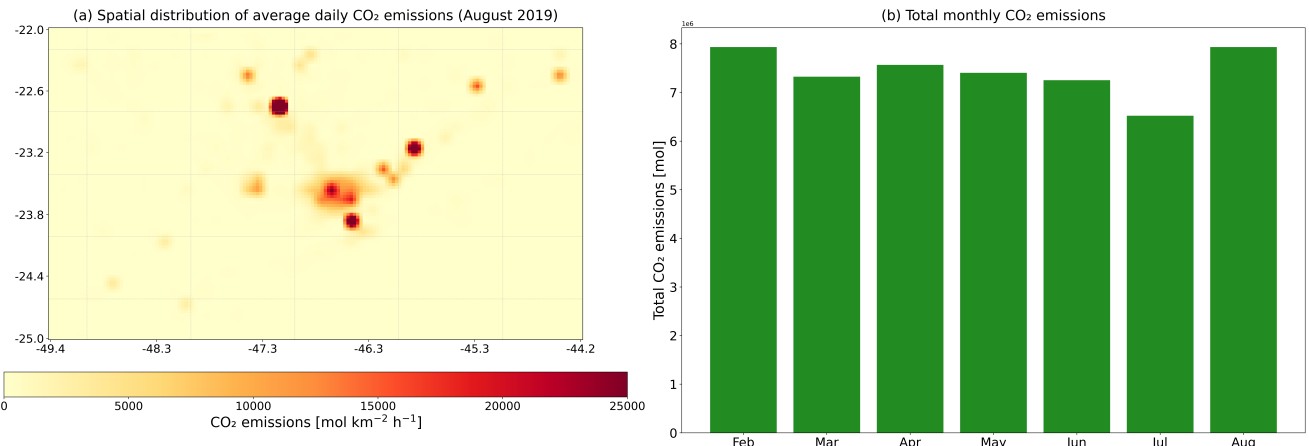

**Figure B2.** $CO_2$ emissions from energy, residential, refineries, and industry sectors by the EDGAR inventory over the study domain (D01). Panel (a) shows the spatial distribution of average daily $CO_2$ emissions for August 2019 over D01. The panel (b) represents the monthly total $CO_2$ emissions from February to August 2019 over the domain.

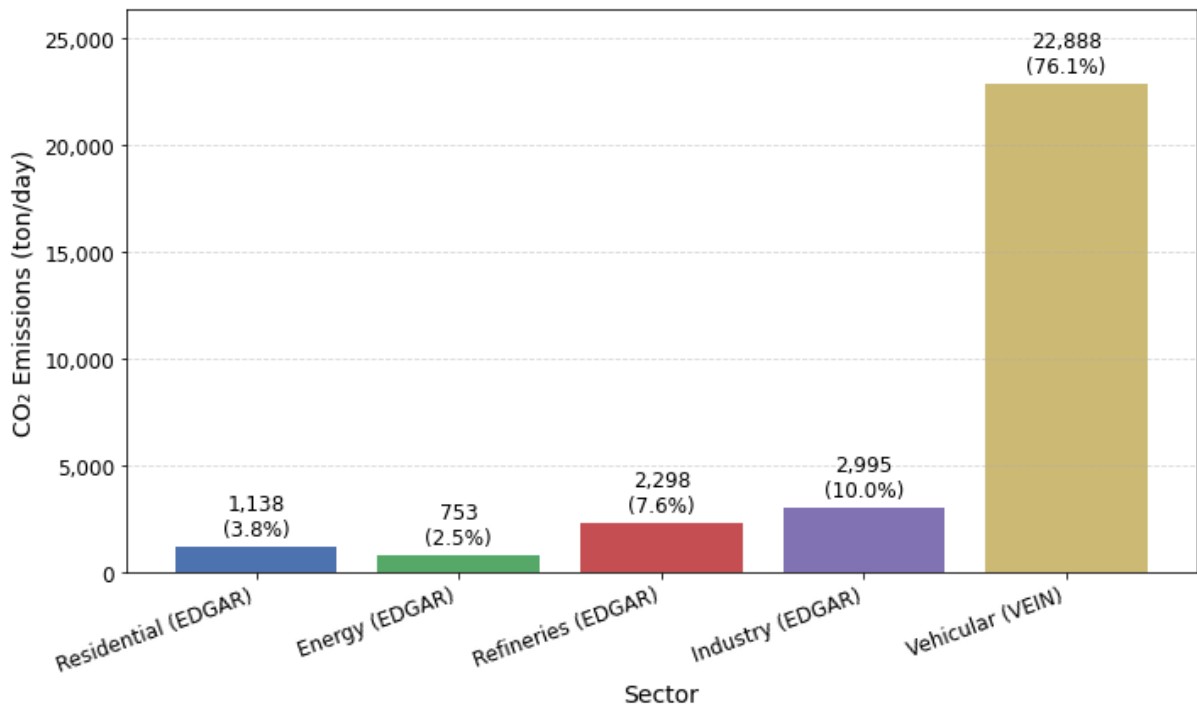

**Figure B3.** Average daily anthropogenic $CO_2$ emissions (in tons) for August 2019 within the simulated domain, disaggregated by sector. Bars represent the mean daily emissions per sector, while percentages indicate each sector's relative contribution to total anthropogenic emissions

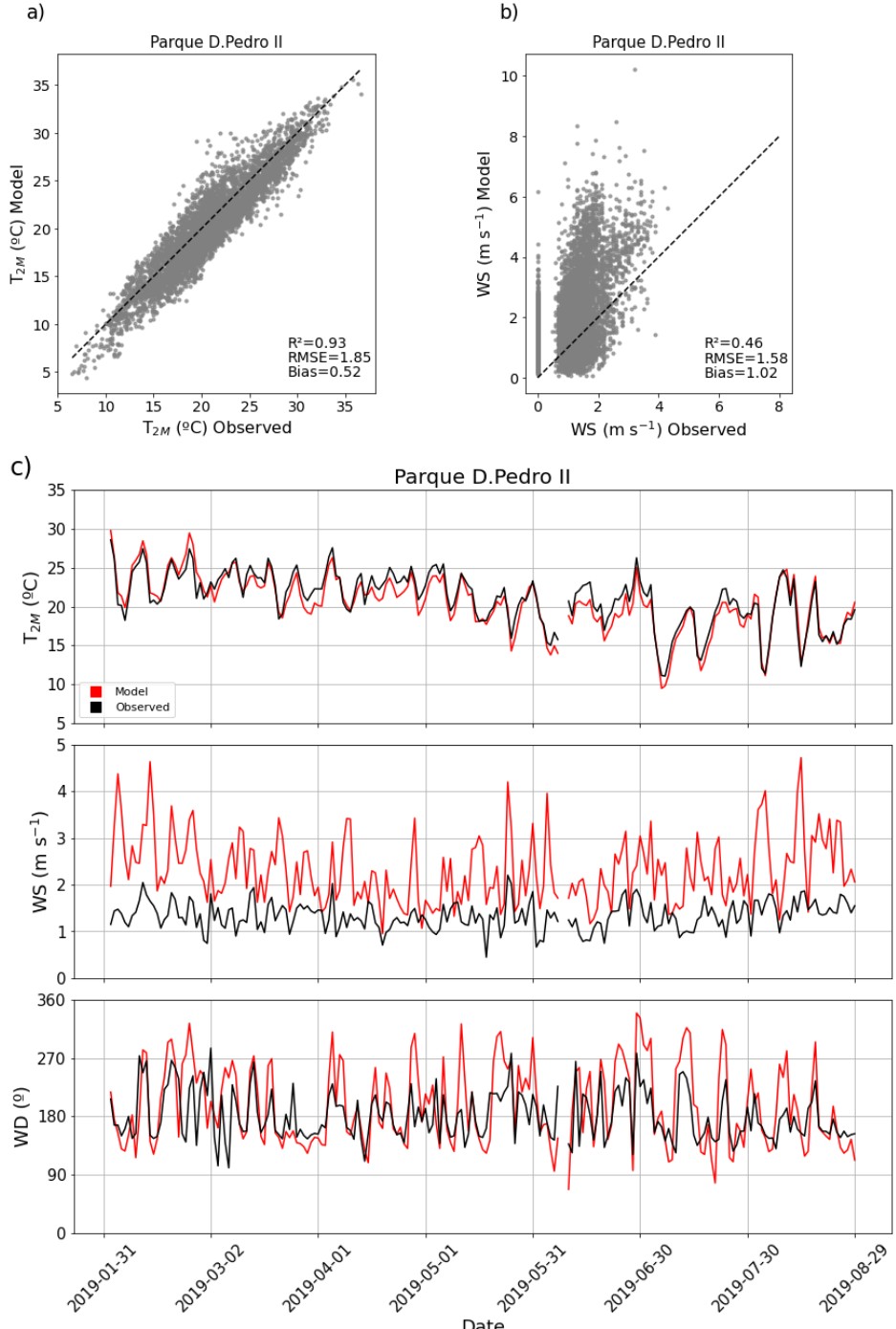

**Figure B4.** Panels (a) and (b) show scatter plots comparing model outputs and observations at the Parque D. Pedro II station for hourly values of 2m air temperature ($T_{2m}$) and 10 m wind speed (WS), respectively. Panel (c) presents the daily averages from February to August 2019 for 2m air temperature ($T_{2m}$), 10 m wind speed (WS), and wind direction (WD). The black line represents observational data, while the red line indicates model simulations.

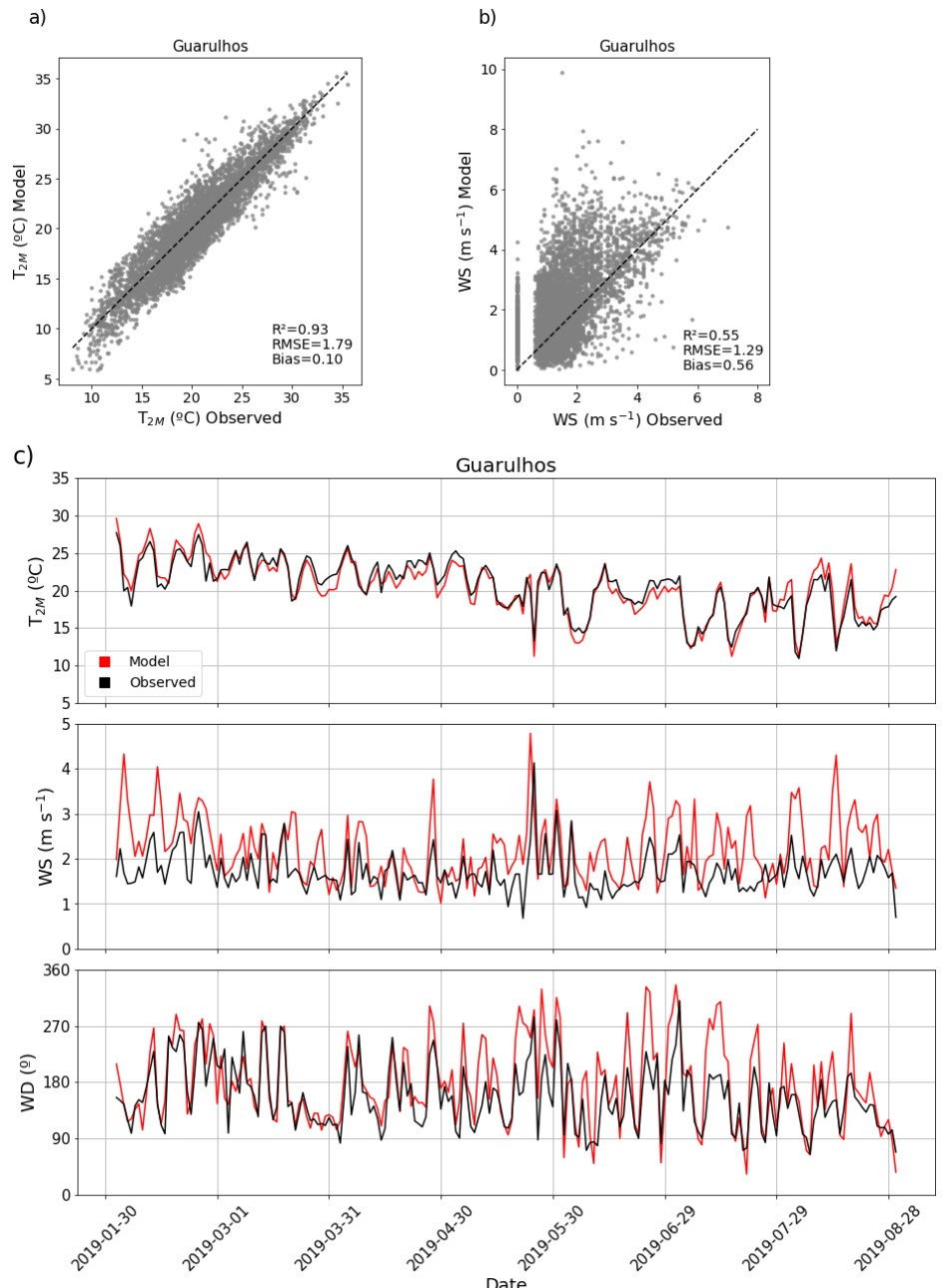

**Figure B5.** Panels (a) and (b) show scatter plots comparing model outputs and observations at the Guarulhos station for hourly values of 2m air temperature ($T_{2m}$) and 10 m wind speed (WS), respectively. Panel (c) presents the daily averages from February to August 2019 for 2m air temperature ($T_{2m}$), 10 m wind speed (WS), and wind direction (WD). The black line represents observational data, while the red line indicates model simulations.

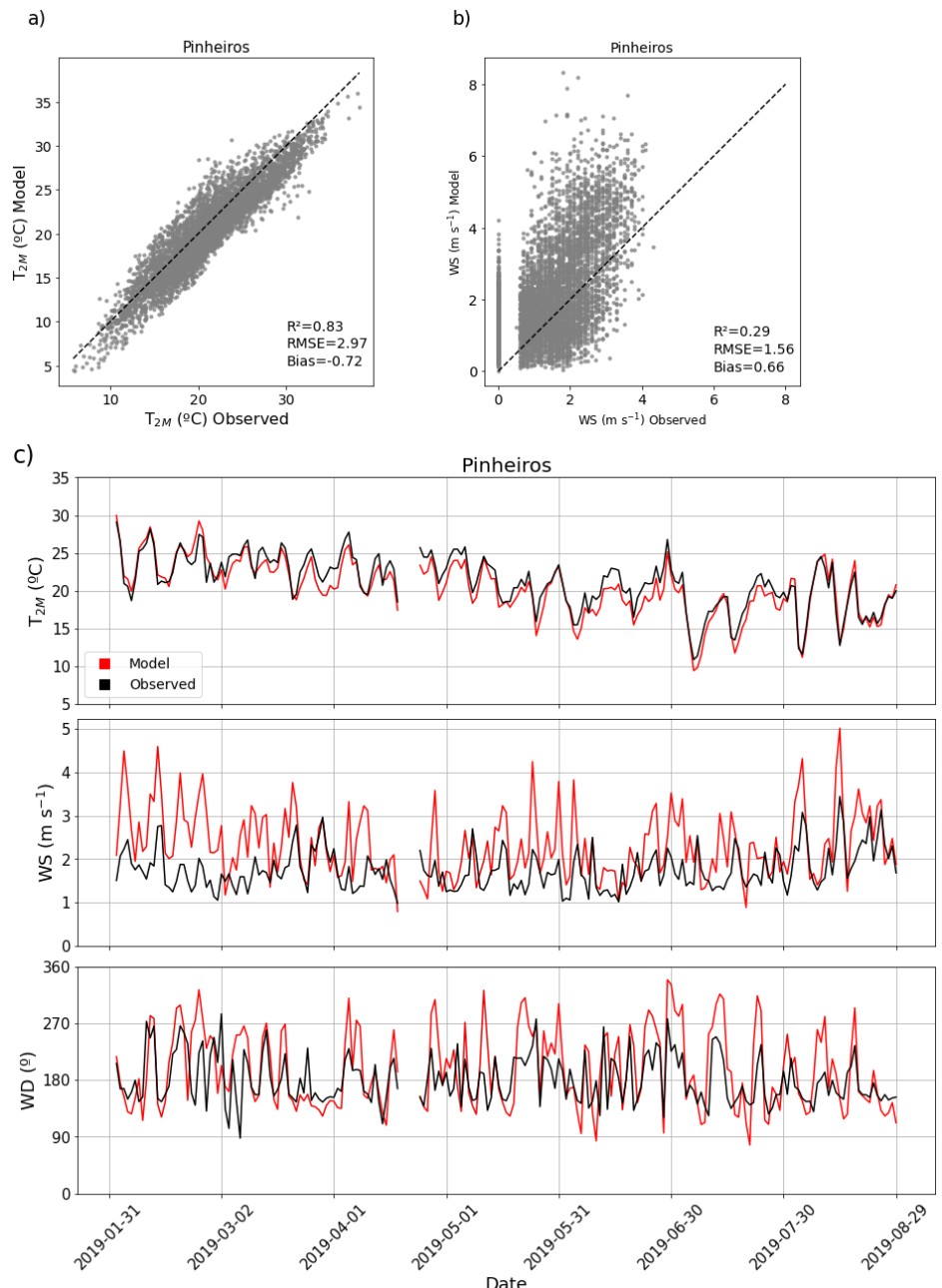

**Figure B6.** Panels (a) and (b) show scatter plots comparing model outputs and observations at the Pinheiros station for hourly values of 2m air temperature ($T_{2m}$) and 10 m wind speed (WS), respectively. Panel (c) presents the daily averages from February to August 2019 for 2m air temperature ($T_{2m}$), 10 m wind speed (WS), and wind direction (WD). The black line represents observational data, while the red line indicates model simulations.

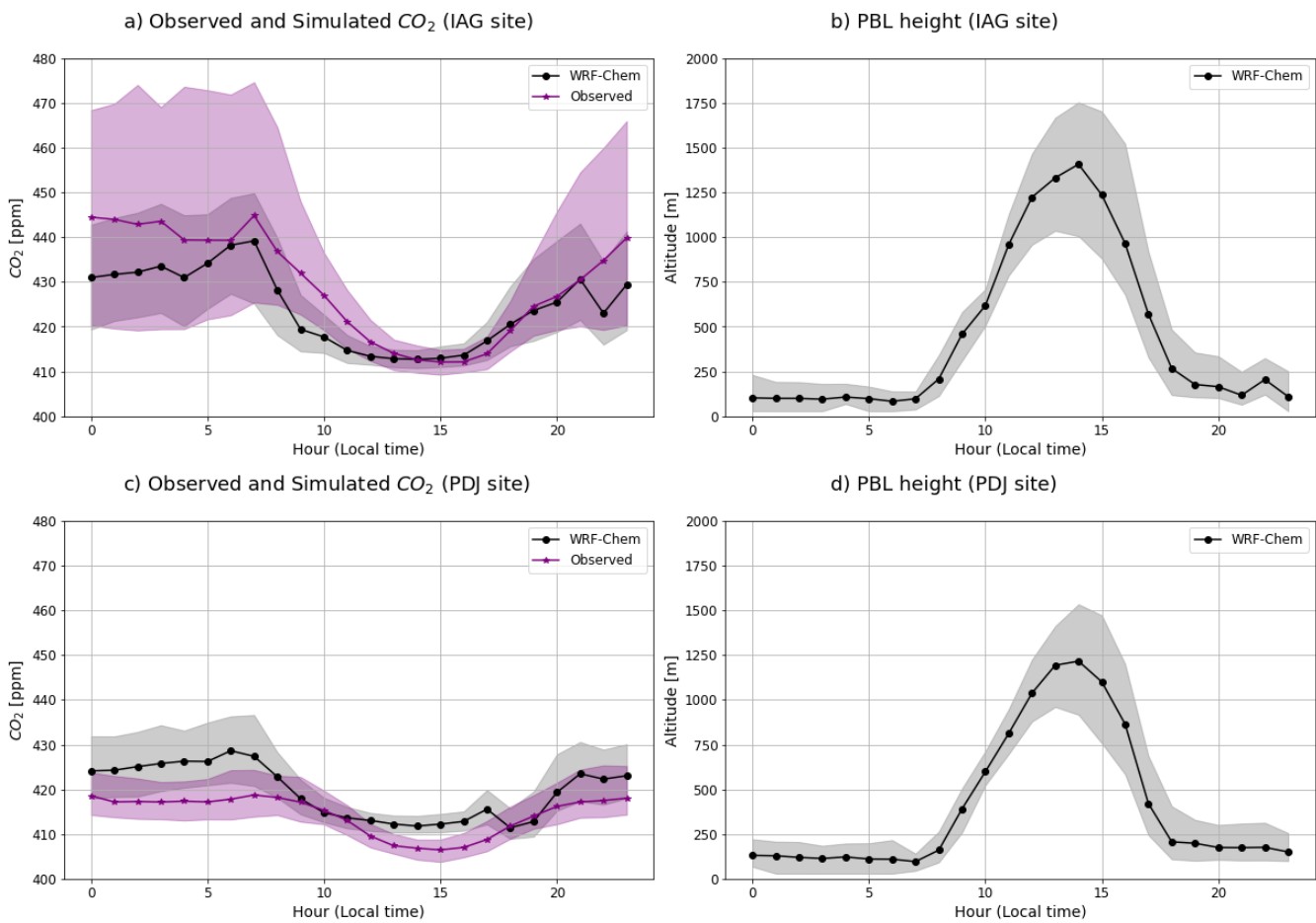

**Figure B7.** Diurnal cycle of in situ $CO_2$ concentration and planetary boundary layer (PBL) height for the entire simulated period. The black line represents the median hourly concentrations from WRF-Chem, while the purple line corresponds to the observed values. The shaded areas indicate the interquartile ranges. Panel a) shows the observed and simulated surface $CO_2$ concentration at the IAG site; b) the simulated PBL height at the IAG site; c) the observed and simulated surface $CO_2$ concentration at the PDJ site; and d) the simulated PBL height at the PDJ site.

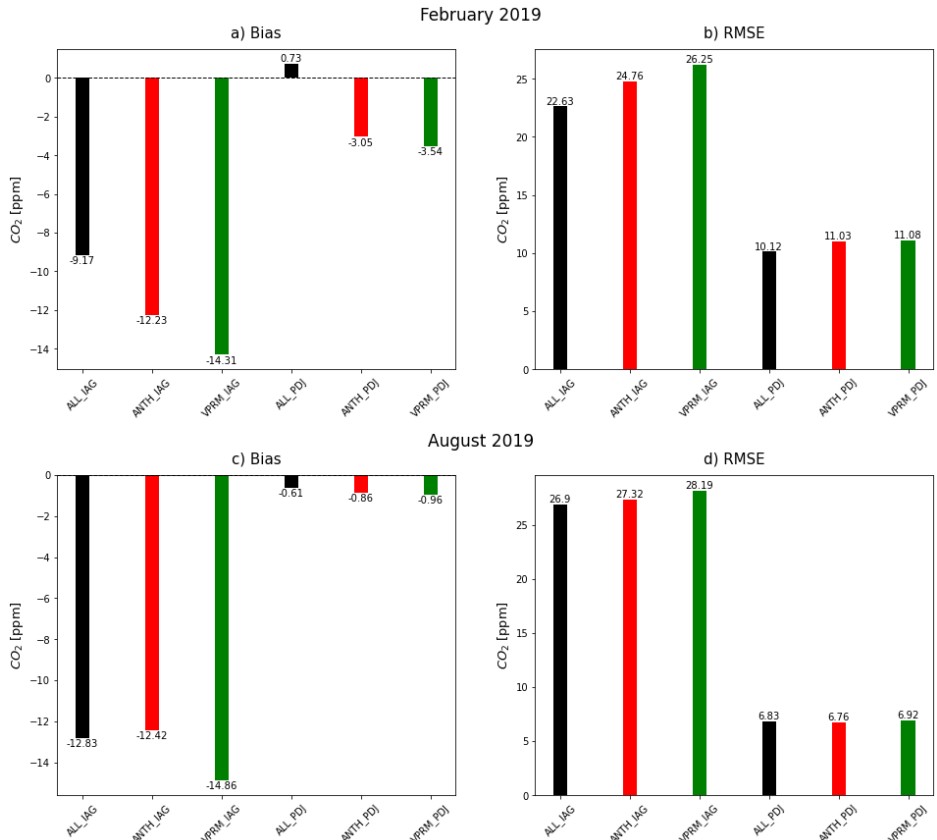

**Figure B8.** Bias (ppm) and RMSE (ppm) for each simulation at the surface $CO_2$ observation sites. Panels (a) and (b) represent the simulations for February, while panels (c) and (d) represent the simulations for August (ALL_*: black, ANTH_*: red, VPRM_*: green) *Represents the observation sites, e.g. IAG and PDJ.

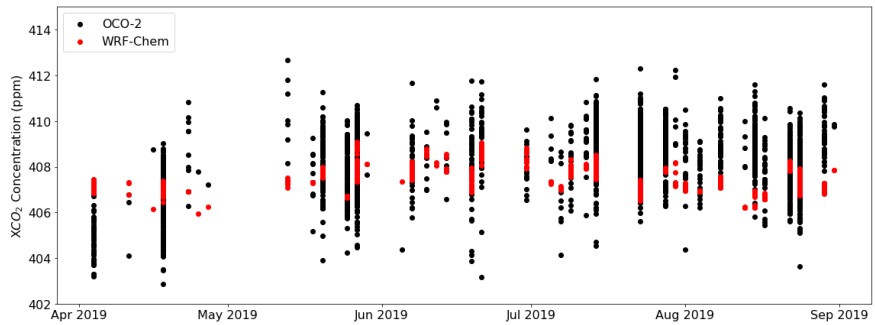

**Figure B9.** Time series of smoothed column concentrations observed (black) and modeled (red) for the period from 1 April 2019 to 31 August 2019.

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
