# Peer review of "Monitoring and modeling seasonally varying anthropogenic and biogenic $CO_2$ over a large tropical metropolitan area"

_EGUsphere, 2024_

## Referee Comment (RC2)

Alberti et al. in this manuscript titled "Monitoring and modeling seasonally varying anthropogenic and biogenic $CO_2$ over a large tropical metropolitan area" attempted to investigate the atmospheric $CO_2$ dynamics in the Metropolitan Area of São Paulo, Brazil with using WRF-GHG/WRF-Chem model output after performing model validation with measurements of two sites (IAG and PDJ) located in the study domain. Although this is an important study for this region, there are a lot of concerns that appeared in the present manuscript and failed to demonstrate the scientific novelty with the model output. For instance, there is a methodological issue in the model setup (nested domain, chem IC/BCs, emissions, etc.) and I guess large uncertainties in the model output compared to the measured concentration are associated with this methodological issue (authors also acknowledged this issue but didn't put any attempt to overcome this issue). Also model output is not well analyzed to support the major conclusions made in this study. Therefore, it needs more work before accepting this manuscript for the ACP.

**Major issues:**
  1) **Model setup**
Since authors in this study attempted to perform WRF-Chem/GHG simulation in a small domain particularly in urban settings, they didn't configure 3km resolution from a nested domain. If you think configuring nested domains is computationally expensive, you might perform a shorter period (Feb and August only). Nested domain configuration will allow you to perform model evaluation in two different resolutions. In Line#259: How did you establish this statement that model resolution is actually causing the mismatch between simulated and observed concentration at the PDJ site? Chemical IC/BCs should be well representative. If you think chem Carbon Tracker based IC/BCs are good enough for your model setup it needs well justification, but as you mentioned in line# 381-382 why you didn't give a try to see what benefit you will achieve if you use other chem IC/BCs such as CAMS which has more finer resolution than CT? It needs more detailed information for the VEIN and EDGAR products while creating anthropogenic emissions. What is the spatial and temporal resolution and year of representation? Did you combine and sum both VEIN and EDGAR products to get total anthropogenic emissions? At this moment it's not clear how these two inventories are configured in the model. You can also provide a map showing the differences of VEIN and EDGAR products. Are there any emission scaling factors being implemented in the inventory for diurnal cycle? It's not clear how diurnal variation is imposed in the model. You might need to impose a temporal anthropogenic emission variation following Nassar et al. (2013). Also look into the following paper where they imposed temporal emission factor

Callewaert, S., Brioude, J., Langerock, B., Duflot, V., Fonteyn, D., Müller, J.-F., Metzger, J.-M., Hermans, C., Kumps, N., Ramonet, M., Lopez, M., Mahieu, E., and De Mazière, M.: Analysis of $CO_2$, $CH_4$, and CO surface and column concentrations observed at Réunion Island by assessing WRF-Chem simulations, Atmos. Chem. Phys., 22, 7763–7792, https://doi.org/10.5194/acp-22-7763-2022, 2022. I saw you referring to Gourdji et al. (2022) in a couple of areas but I'm surprised why you didn't user their approach to optimized your VPRM model. In section 2.2.2

($CO_2$ fluxes data) you should provide more detailed information about the Eddy flux sites used in the VPRM optimization process (perhaps in the supplementary or appendix).

**2) Analyzing model outputs**

Meteorological output such as wind speed and direction should be presented with a wind rose figure. Wind rose figures will help to interpret the sources of $CO_2$ in a site. In many places you are talking about atmospheric stability but there is no analysis to support atmospheric stability-based conclusions (such as section 3.3.1). You can perform PBL height related $CO_2$ profiles at two sites (IAG and PDJ). I'm not convinced with the conclusion made in this paragraph (line#345 to 354 – source contribution), because: Not sure the hour format in Fig. 8. Is this UTC or local hour? If midday has less vehicles how do you expect late night to have high vehicle emissions and cause high correlation between $CO_2$ and CO? Again, it's important to provide some evidence of the diurnal cycle in the VEIN or EDGAR product. In line#236: Do you have any information on the seasonal variation of traffic emission? Seasonal wind rose plot will help you to establish the wind speed related statement written in this line. Abstract is not well written and there is only one conclusion in the abstract without any quantitative evidence.

**3) Figures**

Most of the figures in the current manuscript are not publication standard. Figure captions are not well described. For instance, four panels in Fig. 1 but not clear which one is for what; no panel number for WS and WD panels in Fig. 2. Please elaborate captions in all figures and make clear for all legends.

**Minor comments:**

I'm confused with line#86 and #139 because in #86 it is indicated meteorological variables will come from wrf outputs but in line#139 it is written VPRM model is driven by the meteorological measurements of the sites. Please make it clear.

How did you decide 5 days would be good enough for spin up? Please give a justification.

Table 2, better to write full name of the variables (T2, Wd, Ws etc.) in the table footnote. I think it's more relevant to classify PDJ as Sub-urban park and IAG as University campus/urban park.

Section 2.2.2 ($CO_2$ fluxes data) should be merged with section 2.1.2 or vice versa.

NEE should have a standard unit system (PgC/yr or TgC/yr) and be consistent throughout the manuscript. Currently, two different units are introduced for NEE in Fig. 3 and Fig. 4.

Line#221: Is this statement correct? I see barely green colored in the August map (Fig. 4b).

Line:#223: only vegetation? Line#227 to line#230: Please use quantitative comparison between your study and Raju et al. (2023). Therefore, I suggested following the methodology of Gourdji et al. (2022) while using Eddy flux data for optimizing VPRM.

Be consistent with Pico do Jaragua and PDJ.

Spatial map: Fig 6: Adding a third row with the differences between first and 2nd row would be great.

Line 276: coastal region southwest of MASP -> southeast??

Line 312: how do you confirm this is related to vehicular emission?

Line 317-318: what do you mean by profile? Did you perform any altitude related comparison? Also, I don't see the simulated concentrations are consistent with the observed data at Fig. 7c.

Line# 149: Osterman et al. -> year??
Line# 151: I think satellite measurement time was also matched with the model data during interpolation. Please mention this information. But, if measurement time was not matched please do that.

Line#177:178: Acronym should be consistent throughout the manuscript. For instance, TM, WS and WD in these lines are different than Table 2.

---

## Author Comment (AC1)

First, we would like to thank the anonymous referees for their valuable comments in the interactive comment on "Monitoring and modeling seasonally varying anthropogenic and biogenic *CO2* over a large tropical metropolitan area" by Rafaela Cruz Alves Alberti et al. Their valuable feedback has helped us identify areas for improvement and refine the manuscript accordingly. The manuscript has been revised after the reviewer's comments in order to correct errors and to introduce the reviewers' suggestions for improving the quality of the paper. The comments are in black. Our answers are in blue. Modifications to the manuscript are written in italic.

**Anonymous Referee #1**

Alberti et al. used the WRF-GHG model to simulate CO2 concentration variations in the São Paulo region from February to August 2019. The simulated values were compared with observational data for 2-meter temperature, 10-meter wind, surface CO2, and XCO2. Overall, the study's concept, analytical content, and main conclusions are not novel. The only unique aspect might be the lack of similar studies in the São Paulo region in South America. However, the manuscript currently has many critical missing pieces of information, and the model setup appears to have unreasonable aspects. In the results analysis section, some textual descriptions seem to be based more on prior knowledge rather than being clearly supported by the figures in this study. Most figures in the manuscript are of poor quality, contain several errors, and the text includes several typos. Therefore, I do not suggest publication, unless the authors could convincingly address various concerns I have mentioned below through substantial major revisions.

**Major comments**

1. Model set-up

The authors ran WRF-GHG with only a single domain at a 3 km spatial resolution. First, considering the coarse resolution of input data such as meteorological data and the CarbonTracker CO2 background and initial conditions (e.g.,  $3^{\circ} \times 2^{\circ}$ ), interpolating/downscaling such data directly down to 3 km using WRF, a regional-scale model, might lead to various instabilities or poor performance. For example, the official recommendations suggest using nesting when the input data resolution is coarser than the model resolution by more than a factor of 5–10. Boundary conditions (BCs) for external sources are typically provided at 3–6-hour intervals and lack tendencies for all predicted fields.

Most importantly, based on the distance between the two CO2 observation stations in São Paulo used in this study and the rather similar model performance at both sites shown in Fig. 7 (I will mention it in detail below), the 3 km resolution is insufficient. Given the locations of the CO2 and meteorological observation stations shown in Fig. 1, why didn't the authors set up nesting to achieve a finer resolution, such as 1 km?

The decision to perform the simulation at a spatial resolution of 3 km came from several studies using WRF-Chem over São Paulo. Regarding the absence of nesting while using ERA-5 at the boundaries, we first note that we follow the recommendations (factor of 7-8) for meteorological nudging. We acknowledge that typical simulations using WRF include a large parent domain, but in a similar way as Lauvaux et al. (2012) using a single grid at 10-km resolution, we tested and validated a single-grid configuration by comparing the model performances to previous studies.

The studies by Vara-Vela et al. (2018), Gavidia-Calderón (2021), and Ibarra-Espinosa et al. (2022) configured simulations with two domains to model aerosols and gas pollutants across the MASP. These studies applied finer resolutions in attempts to improve model performance for meteorological and chemical variables. However, they did not show significant statistical differences compared to our

results, even when employing configurations with greater spatial and temporal detail. Our simulated variables were within the thresholds proposed by Monk et al. (2019), except for wind direction. Gavidia-Calderón (2021) reached the same conclusion, using three nesting domains with the finest resolution of 1 km. This behavior was also observed in the aforementioned studies, suggesting that this limitation may be related to modeling meteorological conditions in the region, regardless of the resolution or specific model configuration used. We think that unless additional meteorological measurements are assimilated, the WRF-Chem performances will remain similar even at higher resolutions.

| References                 | Domain                                                                                                                                                          | Objective   | Results                                                                                                                           |
|----------------------------|-----------------------------------------------------------------------------------------------------------------------------------------------------------------|-------------|-----------------------------------------------------------------------------------------------------------------------------------|
| Gavidia-Calderón
(2021) | Triple-nested domains
are set up centered in
MASP. The mother
domain has a spatial
resolution of 25 km,
the second 5 km, and
the finest 1 km. | Air quality | T2m (MB= 0.28)
ws10 (m/s) (MB = 0.79)
wd10 (degree) (MB= -16.24)
Mean results of 16 air quality
networks from CETESB. |

Although nesting techniques to achieve finer resolutions, such as 1 km, could provide additional details, implementing such techniques was beyond our scope. When additional measurement sites will be available, we intend to assimilate the surface meteorological measurements. High-resolution simulations over long periods require significant computational resources, which were not available during this study. As we reached similar model performances as previous studies, we decided to focus on 3-km resolution simulations.

Regarding the CO2 boundary conditions, unfortunately, global inversion models show significant biases at monthly timescales, affected by large uncertainties in tropical net fluxes (Peiro et al., 2022), primarily due to errors from deep convection vertical transport. Unless additional measurements are being made available, large-scale conditions will remain highly uncertain in that region of the globe. Therefore, we focused on the local and regional fluxes, assuming that boundary conditions will remain problematic, even if we propagate our boundary conditions through multiple domains, or at higher frequencies.

1. Model input: Anthropogenic Emissions

In Section 2.1.1, regarding the anthropogenic emission input data used in the model, many critical details are missing:

Lines 73–78: The authors mention that vehicular emissions are the primary emission source in the region. What is the spatial resolution of the VEIN inventory? In the analysis section in the manuscript, poor CO2 simulation performance at the observation sites is repeatedly attributed to the underestimation or overestimation of vehicular emissions. In addition, what is the spatial distribution of vehicular emissions? The authors should include a figure showing the anthropogenic emission distribution for the region.

Line 79: The authors briefly state that other emission sources are from EDGAR. What is the total anthropogenic emission for the region? What are the proportions of emissions from different anthropogenic sources? EDGAR provides emission data at a coarse resolution—how was this processed to fit the 3 km resolution of the model? Which year and version of the EDGAR data were applied? Was a temporal profile used?

In addition, Line 289-290, the authors cite "the EDGAR anthropogenic emission inventory generally overestimates the emissions around local anthropogenic sources (e.g., urban areas)". Did the authors check the total emissions and spatial distribution of EDGAR data in this region? As mentioned above, these infos are not provided in the manuscript.

Based on the authors' description, the WRF-GHG simulation only includes vehicular, energy, and industrial emissions as model inputs. Are these the only 3 sources accounted for, or there are also other sources like residential emissions? All of this essential information is not clearly provided in the manuscript.

Thank you for the detailed comments and valuable suggestions for improving our manuscript. Below, we address the issues raised and explain how we have incorporated the requested information into the revised manuscript.

We have added a detailed explanation regarding the spatial resolution of the VEIN model in the manuscript and also clarified that the VEIN model produces spatially and temporally resolved emissions based on high-resolution inputs, such as traffic flow, urban morphology, and emission factors. For consistency with the WRF-Chem model, the VEIN outputs were aggregated to a resolution of 3 km, which corresponds to spatial grid resolution. To complement this discussion, we have also included a new figure (Fig. B4) in Appendix B, showing the spatial distribution of vehicular emissions in the region. This figure complements the discussions about CO2 simulation performance and the influence of vehicular emissions.

"For consistency with the WRF-Chem model domain, VEIN emissions were aggregated to a 3 km spatial resolution. Additionally, we included a figure (Fig. B4) in Appendix B illustrating the daily mean and hourly temporal variation of vehicular emissions for all months in the study period." Text has updated in lines 77-80.

We also specified in the manuscript that we used EDGAR v6.0 GHG (Crippa et al., 2021) for the year 2018, which provides a spatial resolution of  $0.1^{\circ} \times 0.1^{\circ}$ . We detailed the data processing steps, including conversion to NetCDF format, horizontal resizing to the 3 km resolution, and temporal adaptation for the simulation period. These details have been added to the methodology section: "These emissions were processed to match the 3 km spatial resolution of the WRF-Chem model using interpolation techniques. However, EDGAR emissions lack temporal variability and were assumed constant throughout the day, as the inventory does not provide hourly profiles (Fig. B5 Appendix B)." (Lines 82–85).

Additionally, while the VEIN data include hourly temporal profiles derived from local traffic data, the EDGAR emissions are assumed to be temporally invariant due to the original dataset's lack of hourly resolution. The revised manuscript acknowledges this limitation and also includes a spatial distribution of EDGAR emissions in the study region in Appendix B (Fig. B5).

Line 390-391: In Section 4 (Conclusion), the authors state that "Anthropogenic emissions were curated from diverse models and products to accurately reflect real urban conditions." However, where is the evidence to support the claim of "accurately"? This is particularly questionable given the significant bias and RMSE observed in the simulated CO2 concentrations (Figure 4A).

Thank you for pointing out this inconsistency in our conclusion. We acknowledge that the term "accurately" may not fully align with the observed biases and RMSE in the simulated CO2 concentrations. To address this, we have revised the sentence in the conclusion section to more better reflect the scope and limitations of the study. The updated text you can see in Lines 421-423 mentioned the "anthropogenic emissions were curated from vehicular model and global inventory to provide a comprehensive representation of urban emissions, incorporating spatial and temporal resolution for key sources such as vehicular traffic for our domain".

Line 399-400: The authors provide conclusions regarding the temporal profile of simulated CO2 emissions, but they do not introduce or present the "prescribed temporal profiles of anthropogenic emissions" in the manuscript. While this conclusion is correct based on prior knowledge, it is not well-supported by the analysis presented in this study.

To address this, we have included Fig. B4 -h) in Appendix B, which illustrates the hourly temporal profiles of vehicular emissions as modeled by the VEIN model, using real traffic data.

This figure demonstrates how vehicular emissions vary throughout the day and across different months, providing critical insights into the temporal dynamics of one of the primary sources of CO2

in the MASP. Incorporating these temporal profiles into the WRF-Chem model ensures a more realistic representation of emissions, aligning with the observed diurnal patterns of CO2 concentrations in urban areas.

**1. Model input: Biogenic Emissions**

Line 44, Line 81, Line 86: The authors mention in Line 44 that VPRM is coupled to WRF-GHG, which indicates that it could be either online or offline coupling. However, in Line 81, they say it is offline but implemented as a module, while in Line 86, they mention that VPRM's temperature and shortwave radiation inputs come from the WRF model. This raises several questions: Did the authors first run WRF to obtain these meteorological inputs, then run VPRM to calculate the biogenic fluxes, and finally use these fluxes as a tracer in a subsequent WRF-GHG simulation? If so, this is inconsistent with Line 82, because it is just a model flux input, same as anthropogenic flux input, rather than a coupled module within WRF-GHG. Additionally, how was the "default" VPRM in Fig. 3 calculated? Was it based on the online-coupled VPRM in WRF-Chem, or was it handled differently? These infos are unclear.

Thank you for pointing out the inconsistencies in our descriptions regarding the coupling and implementation of VPRM within the WRF-Chem framework.

The VPRM model was used offline for this study. Specifically, WRF-Chem was first run to generate meteorological outputs, including temperature and shortwave radiation. It has been corrected in Line 44: "*offline coupled*". These outputs were then used as inputs to VPRM to calculate the biogenic CO2 fluxes. The resulting biogenic fluxes were subsequently incorporated into the WRF-Chem model combining with anthropogenic emissions. We agree that this workflow aligns more closely with the concept of using VPRM as a flux input, rather than as an online-coupled module and we have corrected this in Line 92.

Regarding the "default" VPRM simulations shown in Fig. 3, these simulations were conducted similarly to the optimization simulations but using the parameters defined in the Mahadevan et al. (2008) study. The VPRM was run at an hourly resolution for the flux tower sites during periods with available observational data. Site-specific simulations incorporated local meteorological inputs and vegetation indices to adapt the parameters to the conditions at each site. This approach enabled a comparative evaluation of the model's performance when using its standard parameter set versus parameters optimized for the specific flux tower conditions.

In Section 2.1.2, the authors dedicate a large portion of the text to introduce the VPRM model. This information could be simply cited from the VPRM paper (Mahadevan et al., 2008) or moved to the supplementary materials. Similarly, in Section 2.3, there is no need to include basic explanations of metrics such as bias, RMSE, and correlation in the main text. These are well-known concepts and can either be briefly mentioned.

**We agree to move the information in Section 2.3 to the Appendix A and the text information in Lines 167-172.**

In Section 3.2, I question the validity of the comparison between the optimized VPRM, default VPRM, and observed biogenic fluxes, which concludes that the optimized flux is closer to the observed data. This approach is problematic because the authors used the observed data to optimize the VPRM parameters (Lines 137-138) and then compared the optimized flux against the same set of observed data. This is inappropriate, thus the main conclusion here is also questionable whether it is credible. The authors need to validate the model using independent observational data rather than the same dataset used for optimization. For example, the author could use half of the observational time series to optimize the model parameters and the other half to validate.

We acknowledge that independent validation would be ideal to assess the robustness of the model under conditions not included in the optimization. However, due to the limited number of flux measurement sites (only three towers covering different ecosystems) and the need for a full seasonal cycle over a year to constrain the VPRM model, we were unable to set aside an independent subset for evaluation.

To address this, we followed the recommendations of previous studies (Mahadevan et al., 2008; Ahmadov et al., 2007; Xiao et al., 2014), adapting them to our specific context by using available observational datasets and evaluating model performance with non-independent measurements.

While this approach is not ideal, we optimized parameters over the entire year, using MODIS EVI as a proxy for photosynthesis—an important simplification for evergreen ecosystems like the Atlantic Forest and incorporating soil moisture to represent water availability, a key limitation for tropical ecosystems. As shown in Figure 3, certain periods exhibit poor performance, indicating that despite optimization, the VPRM model does not always fully capture seasonal dynamics.

To clarify this point, and to balance our conclusions, we have recommended, in the conclusion section, that additional of flux towers should be deployed, and possibly some dedicated measurement campaigns to measure the other ecosystems should be carried out.

"However, we recommend the deployment of additional flux towers and targeted measurement campaigns to better characterize other ecosystems and a more comprehensive representation of PFTs is essential, as vegetation processes play a major role in shaping CO2 patterns in tropical regions." (Lines 417-421).

My second concern is the poor quality and errors in many figures.

Figure 1: There are inconsistencies between the data types shown in Figure 1 and the site information in Table 1. For instance, Figure 1 indicates two sites observing CO, while Table 1 lists only one CO observation site. Additional issues with Figure 1 include:

Legend clarity: The legend is not intuitive. The caption should include a statement explaining that different symbols represent site types and different colors indicate the types of observational data.

Scale: A scale bar should be added to the figure. For example, it is difficult to discern the distance between the two CO2 observation sites. The authors state that the CO and CO2 sites are less than 3 km apart, therefore are the two CO2 sites only 1 km apart?

Why were the observation sites placed so close together? If the two CO2 sites are only 1 km apart, this raises the earlier question of why the model resolution was set at 3 km. With such close proximity, it is highly likely that these two sites fall within the same model grid cell, which could explain why their simulation results appear so similar, as observed in Figure 7.

In panel (b), is the land use map from WRF, or is it another map? The grid cells do not appear to follow a "regular" grid—did the authors use interpolation or smoothing? Note that in WRF NetCDF files, the land use map is provided as the dominant type for each grid cell. Additionally, it is recommended to change the colormap used in the land use map. The current color scheme and shapefile make it very difficult to distinguish between different land use types.

**• Inconsistencies between Figure 1 and Table 2:**

We have reviewed the data types and site information. The discrepancy regarding the CO observation sites has been corrected. Figure 1 now accurately represents the observational data and aligns with the information provided in Table 2.

• Legend Clarity:

The legend has been updated to enhance clarity. We added a statement in the caption explaining that different symbols represent site types and that different colors correspond to the observational data types. This ensures a better interpretation of Figure 1.

• Scale:

The error in the original figure contributed to this misunderstanding. The IAG site, represented by the black cross (CO2 measurements), and the Pinheiros site, represented by the yellow star symbol (CO measurements), are located less than 3 km apart. This proximity was deliberate to facilitate correlation analysis between CO2 and CO concentrations, as shown in Section 3.3.2 (Figures 8 and 9). The distance between the CO2 observation sites, IAG and PDJ, is approximately 13 km. Regarding the question of proximity to the model resolution: While the IAG and PDJ sites fall within separate grid cells in the 3 km resolution of the WRF-Chem model, the chosen resolution is a balance between computational feasibility and the need to represent regional-scale processes.

Additionally, the image below, taken from Google Earth, illustrates the actual distances and spatial relationships between the sites at a scale of 10 km.

---

## Author Comment (AC2)

First, we would like to thank the anonymous referees for their valuable comments in the interactive comment on "**Monitoring and modeling seasonally varying anthropogenic and biogenic *CO2* over a large tropical metropolitan area**" by Rafaela Cruz Alves Alberti et al. Their valuable feedback has helped us identify areas for improvement and refine the manuscript accordingly. The manuscript has been revised after the reviewer's comments in order to correct errors and to introduce the reviewers' suggestions for improving the quality of the paper. The comments are in black. Our answers are in blue. Modifications to the manuscript are written in italic.

**Anonymous Referee #1**

Alberti et al. used the WRF-GHG model to simulate CO2 concentration variations in the São Paulo region from February to August 2019. The simulated values were compared with observational data for 2-meter temperature, 10-meter wind, surface CO2, and XCO2. Overall, the study's concept, analytical content, and main conclusions are not novel. The only unique aspect might be the lack of similar studies in the São Paulo region in South America. However, the manuscript currently has many critical missing pieces of information, and the model setup appears to have unreasonable aspects. In the results analysis section, some textual descriptions seem to be based more on prior knowledge rather than being clearly supported by the figures in this study. Most figures in the manuscript are of poor quality, contain several errors, and the text includes several typos. Therefore, I do not suggest publication, unless the authors could convincingly address various concerns I have mentioned below through substantial major revisions.

**Major comments**

1. Model set-up

The authors ran WRF-GHG with only a single domain at a 3 km spatial resolution. First, considering the coarse resolution of input data such as meteorological data and the CarbonTracker CO2 background and initial conditions (e.g., 3° × 2°), interpolating/downscaling such data directly down to 3 km using WRF, a regional-scale model, might lead to various instabilities or poor performance. For example, the official recommendations suggest using nesting when the input data resolution is coarser than the model resolution by more than a factor of 5–10. Boundary conditions (BCs) for external sources are typically provided at 3–6-hour intervals and lack tendencies for all predicted fields.

Most importantly, based on the distance between the two CO2 observation stations in São Paulo used in this study and the rather similar model performance at both sites shown in Fig. 7 (I will mention it in detail below), the 3 km resolution is insufficient. Given the locations of the CO2 and meteorological observation stations shown in Fig. 1, why didn't the authors set up nesting to achieve a finer resolution, such as 1 km?

The decision to perform the simulation at a spatial resolution of 3 km came from several studies using WRF-Chem over São Paulo. Regarding the absence of nesting while using ERA-5 at the boundaries, we first note that we follow the recommendations (factor of 7-8) for meteorological nudging. We acknowledge that typical simulations using WRF include a large parent domain, but in a similar way as Lauvaux et al. (2012) using a single grid at 10-km resolution, we tested and validated a single-grid configuration by comparing the model performances to previous studies.

The studies by Vara-Vela et al. (2018), Gavidia-Calderón (2021), and Ibarra-Espinosa et al. (2022) configured simulations with two domains to model aerosols and gas pollutants across the MASP. These studies applied finer resolutions in attempts to improve model performance for meteorological and chemical variables. However, they did not show significant statistical differences compared to our

results, even when employing configurations with greater spatial and temporal detail. Our simulated variables were within the thresholds proposed by Monk et al. (2019), except for wind direction. Gavidia-Calderón (2021) reached the same conclusion, using three nesting domains with the finest resolution of 1 km. This behavior was also observed in the aforementioned studies, suggesting that this limitation may be related to modeling meteorological conditions in the region, regardless of the resolution or specific model configuration used. We think that unless additional meteorological measurements are assimilated, the WRF-Chem performances will remain similar even at higher resolutions.

| References | Domain | Objective | Results |
|---|---|---|---|
| Gavidia-Calderón (2021) | Triple-nested domains are set up centered in MASP. The mother domain has a spatial resolution of 25 km, the second 5 km, and the finest 1 km. | Air quality | T2m (MB= 0.28) ws10 (m/s) (MB = 0.79) wd10 (degree) (MB= -16.24)  Mean results of 16 air quality networks from CETESB. |

Although nesting techniques to achieve finer resolutions, such as 1 km, could provide additional details, implementing such techniques was beyond our scope. When additional measurement sites will be available, we intend to assimilate the surface meteorological measurements. High-resolution simulations over long periods require significant computational resources, which were not available during this study. As we reached similar model performances as previous studies, we decided to focus on 3-km resolution simulations.

Regarding the $CO_2$ boundary conditions, unfortunately, global inversion models show significant biases at monthly timescales, affected by large uncertainties in tropical net fluxes (Peiro et al., 2022), primarily due to errors from deep convection vertical transport. Unless additional measurements are being made available, large-scale conditions will remain highly uncertain in that region of the globe. Therefore, we focused on the local and regional fluxes, assuming that boundary conditions will remain problematic, even if we propagate our boundary conditions through multiple domains, or at higher frequencies.

1. Model input: Anthropogenic Emissions

In Section 2.1.1, regarding the anthropogenic emission input data used in the model, many critical details are missing:

Lines 73–78: The authors mention that vehicular emissions are the primary emission source in the region. What is the spatial resolution of the VEIN inventory? In the analysis section in the manuscript, poor $CO_2$ simulation performance at the observation sites is repeatedly attributed to the underestimation or overestimation of vehicular emissions. In addition, what is the spatial distribution of vehicular emissions? The authors should include a figure showing the anthropogenic emission distribution for the region.

Line 79: The authors briefly state that other emission sources are from EDGAR. What is the total anthropogenic emission for the region? What are the proportions of emissions from different anthropogenic sources? EDGAR provides emission data at a coarse resolution—how was this processed to fit the 3 km resolution of the model? Which year and version of the EDGAR data were applied? Was a temporal profile used?

In addition, Line 289-290, the authors cite "the EDGAR anthropogenic emission inventory generally overestimates the emissions around local anthropogenic sources (e.g., urban areas)". Did the authors check the total emissions and spatial distribution of EDGAR data in this region? As mentioned above, these infos are not provided in the manuscript.

Based on the authors' description, the WRF-GHG simulation only includes vehicular, energy, and industrial emissions as model inputs. Are these the only 3 sources accounted for, or there are also other sources like residential emissions? All of this essential information is not clearly provided in the manuscript.

Thank you for the detailed comments and valuable suggestions for improving our manuscript. Below, we address the issues raised and explain how we have incorporated the requested information into the revised manuscript.

We have added a detailed explanation regarding the spatial resolution of the VEIN model in the manuscript and also clarified that the VEIN model produces spatially and temporally resolved emissions based on high-resolution inputs, such as traffic flow, urban morphology, and emission factors. For consistency with the WRF-Chem model, the VEIN outputs were aggregated to a resolution of 3 km, which corresponds to spatial grid resolution. To complement this discussion, we have also included a new figure (Fig. B4) in Appendix B, showing the spatial distribution of vehicular emissions in the region. This figure complements the discussions about CO2 simulation performance and the influence of vehicular emissions.

"*For consistency with the WRF-Chem model domain, VEIN emissions were aggregated to a 3 km spatial resolution. Additionally, we included a figure (Fig. B4) in Appendix B illustrating the daily mean and hourly temporal variation of vehicular emissions for all months in the study period.*" Text has updated in lines 77-80.

We also specified in the manuscript that we used EDGAR v6.0 GHG (Crippa et al., 2021) for the year 2018, which provides a spatial resolution of $0.1° \times 0.1°$. We detailed the data processing steps, including conversion to NetCDF format, horizontal resizing to the 3 km resolution, and temporal adaptation for the simulation period. These details have been added to the methodology section: "*These emissions were processed to match the 3 km spatial resolution of the WRF-Chem model using interpolation techniques. However, EDGAR emissions lack temporal variability and were assumed constant throughout the day, as the inventory does not provide hourly profiles (Fig. B5 Appendix B).*" (Lines 82–85).

Additionally, while the VEIN data include hourly temporal profiles derived from local traffic data, the EDGAR emissions are assumed to be temporally invariant due to the original dataset's lack of hourly resolution. The revised manuscript acknowledges this limitation and also includes a spatial distribution of EDGAR emissions in the study region in Appendix B (Fig. B5).

Line 390-391: In Section 4 (Conclusion), the authors state that "Anthropogenic emissions were curated from diverse models and products to accurately reflect real urban conditions." However, where is the evidence to support the claim of "accurately"? This is particularly questionable given the significant bias and RMSE observed in the simulated CO2 concentrations (Figure 4A).

Thank you for pointing out this inconsistency in our conclusion. We acknowledge that the term "accurately" may not fully align with the observed biases and RMSE in the simulated CO2 concentrations. To address this, we have revised the sentence in the conclusion section to more better reflect the scope and limitations of the study. The updated text you can see in Lines 421-423 mentioned the "*anthropogenic emissions were curated from vehicular model and global inventory to provide a comprehensive representation of urban emissions, incorporating spatial and temporal resolution for key sources such as vehicular traffic for our domain*".

Line 399-400: The authors provide conclusions regarding the temporal profile of simulated CO2 emissions, but they do not introduce or present the "prescribed temporal profiles of anthropogenic emissions" in the manuscript. While this conclusion is correct based on prior knowledge, it is not well-supported by the analysis presented in this study.

To address this, we have included Fig. B4 -h) in Appendix B, which illustrates the hourly temporal profiles of vehicular emissions as modeled by the VEIN model, using real traffic data. This figure demonstrates how vehicular emissions vary throughout the day and across different months, providing critical insights into the temporal dynamics of one of the primary sources of CO2

in the MASP. Incorporating these temporal profiles into the WRF-Chem model ensures a more realistic representation of emissions, aligning with the observed diurnal patterns of CO2 concentrations in urban areas.

1. Model input: Biogenic Emissions

Line 44, Line 81, Line 86: The authors mention in Line 44 that VPRM is coupled to WRF-GHG, which indicates that it could be either online or offline coupling. However, in Line 81, they say it is offline but implemented as a module, while in Line 86, they mention that VPRM's temperature and shortwave radiation inputs come from the WRF model. This raises several questions: Did the authors first run WRF to obtain these meteorological inputs, then run VPRM to calculate the biogenic fluxes, and finally use these fluxes as a tracer in a subsequent WRF-GHG simulation? If so, this is inconsistent with Line 82, because it is just a model flux input, same as anthropogenic flux input, rather than a coupled module within WRF-GHG. Additionally, how was the "default" VPRM in Fig. 3 calculated? Was it based on the online-coupled VPRM in WRF-Chem, or was it handled differently? These infos are unclear.

Thank you for pointing out the inconsistencies in our descriptions regarding the coupling and implementation of VPRM within the WRF-Chem framework.
The VPRM model was used offline for this study. Specifically, WRF-Chem was first run to generate meteorological outputs, including temperature and shortwave radiation. It has been corrected in Line 44: "*offline coupled*". These outputs were then used as inputs to VPRM to calculate the biogenic CO2 fluxes. The resulting biogenic fluxes were subsequently incorporated into the WRF-Chem model combining with anthropogenic emissions. We agree that this workflow aligns more closely with the concept of using VPRM as a flux input, rather than as an online-coupled module and we have corrected this in Line 92.
Regarding the "default" VPRM simulations shown in Fig. 3, these simulations were conducted similarly to the optimization simulations but using the parameters defined in the Mahadevan et al. (2008) study. The VPRM was run at an hourly resolution for the flux tower sites during periods with available observational data. Site-specific simulations incorporated local meteorological inputs and vegetation indices to adapt the parameters to the conditions at each site. This approach enabled a comparative evaluation of the model's performance when using its standard parameter set versus parameters optimized for the specific flux tower conditions.

In Section 2.1.2, the authors dedicate a large portion of the text to introduce the VPRM model. This information could be simply cited from the VPRM paper (Mahadevan et al., 2008) or moved to the supplementary materials. Similarly, in Section 2.3, there is no need to include basic explanations of metrics such as bias, RMSE, and correlation in the main text. These are well-known concepts and can either be briefly mentioned.

We agree to move the information in Section 2.3 to the Appendix A and the text information in Lines 167-172.

In Section 3.2, I question the validity of the comparison between the optimized VPRM, default VPRM, and observed biogenic fluxes, which concludes that the optimized flux is closer to the observed data. This approach is problematic because the authors used the observed data to optimize the VPRM parameters (Lines 137-138) and then compared the optimized flux against the same set of observed data. This is inappropriate, thus the main conclusion here is also questionable whether it is credible. The authors need to validate the model using independent observational data rather than the same dataset used for optimization. For example, the author could use half of the observational time series to optimize the model parameters and the other half to validate.

We acknowledge that independent validation would be ideal to assess the robustness of the model under conditions not included in the optimization. However, due to the limited number of flux measurement sites (only three towers covering different ecosystems) and the need for a full seasonal

cycle over a year to constrain the VPRM model, we were unable to set aside an independent subset for evaluation.

To address this, we followed the recommendations of previous studies (Mahadevan et al., 2008; Ahmadov et al., 2007; Xiao et al., 2014), adapting them to our specific context by using available observational datasets and evaluating model performance with non-independent measurements.

While this approach is not ideal, we optimized parameters over the entire year, using MODIS EVI as a proxy for photosynthesis—an important simplification for evergreen ecosystems like the Atlantic Forest and incorporating soil moisture to represent water availability, a key limitation for tropical ecosystems. As shown in Figure 3, certain periods exhibit poor performance, indicating that despite optimization, the VPRM model does not always fully capture seasonal dynamics.

To clarify this point, and to balance our conclusions, we have recommended, in the conclusion section, that additional of flux towers should be deployed, and possibly some dedicated measurement campaigns to measure the other ecosystems should be carried out.

*"However, we recommend the deployment of additional flux towers and targeted measurement campaigns to better characterize other ecosystems and a more comprehensive representation of PFTs is essential, as vegetation processes play a major role in shaping CO2 patterns in tropical regions."* (Lines 417-421).

My second concern is the poor quality and errors in many figures.

Figure 1: There are inconsistencies between the data types shown in Figure 1 and the site information in Table 1. For instance, Figure 1 indicates two sites observing CO, while Table 1 lists only one CO observation site. Additional issues with Figure 1 include:

Legend clarity: The legend is not intuitive. The caption should include a statement explaining that different symbols represent site types and different colors indicate the types of observational data.

Scale: A scale bar should be added to the figure. For example, it is difficult to discern the distance between the two CO2 observation sites. The authors state that the CO and CO2 sites are less than 3 km apart, therefore are the two CO2 sites only 1 km apart?

Why were the observation sites placed so close together? If the two CO2 sites are only 1 km apart, this raises the earlier question of why the model resolution was set at 3 km. With such close proximity, it is highly likely that these two sites fall within the same model grid cell, which could explain why their simulation results appear so similar, as observed in Figure 7.

In panel (b), is the land use map from WRF, or is it another map? The grid cells do not appear to follow a "regular" grid—did the authors use interpolation or smoothing? Note that in WRF NetCDF files, the land use map is provided as the dominant type for each grid cell. Additionally, it is recommended to change the colormap used in the land use map. The current color scheme and shapefile make it very difficult to distinguish between different land use types.

- Inconsistencies between Figure 1 and Table 2:

We have reviewed the data types and site information. The discrepancy regarding the CO observation sites has been corrected. Figure 1 now accurately represents the observational data and aligns with the information provided in Table 2.

- Legend Clarity:

The legend has been updated to enhance clarity. We added a statement in the caption explaining that different symbols represent site types and that different colors correspond to the observational data types. This ensures a better interpretation of Figure 1.

- Scale:

The error in the original figure contributed to this misunderstanding. The IAG site, represented by the black cross (CO2 measurements), and the Pinheiros site, represented by the yellow star symbol (CO measurements), are located less than 3 km apart. This proximity was deliberate to facilitate correlation analysis between CO2 and CO concentrations, as shown in Section 3.3.2 (Figures 8 and 9). The distance between the CO2 observation sites, IAG and PDJ, is approximately 13 km. Regarding the question of proximity to the model resolution: While the IAG and PDJ sites fall within separate grid cells in the 3 km resolution of the WRF-Chem model, the chosen resolution is a balance between computational feasibility and the need to represent regional-scale processes.

Additionally, the image below, taken from Google Earth, illustrates the actual distances and spatial relationships between the sites at a scale of 10 km.

[Figure]

Source: Google Earth image.

- Panel (b) Land Use Map

The land use map in panel (b) represents the land use file used in the VPRM model to calculate CO2 fluxes. The original file had a resolution of 50 meters, but it was interpolated to match the 3 km resolution of the D01 domain used in this study. This interpolation ensures consistency between the spatial resolution of the input data and the model domain.

To improve interpretability, we have updated the colormap to ensure that different land use types are easily distinguishable. The caption of Figure 1 has also been revised to clearly describe the methodology applied in creating the land use map, including the interpolation process and its purpose within the VPRM framework.

Figure 2b:

In the comparison of the hourly model and observation wind speed, why didn't the authors plot a 1:1 ratio line instead of a regression line? This choice seems unusual.

It is also strange that there are two types of symbols in the scatter plot—some are circles, while others are crosses.

I assume due to the precision decimal issue in the observational data, it has vertical patterns in the scatter. However, why does this only occur for wind speeds below 3 m/s, and not above?

Why does the WRF-simulated 10 m wind speed show many values of 0 m/s?

I suggest making this plot square rather than rectangular for better visual clarity.

We agree that plotting a 1:1 ratio line is a more standard and effective approach to evaluating the scatter plots. Accordingly, as suggested we have updated the panels Figure 2a) and b) for PDJ site and the figures in appendix B (B1, B2 and B3 ) for the other sites.
We have corrected the issue with repeating axes to enhance clarity and understanding. Additionally, we have revised the figures to incorporate one symbol and square rather. The updated Figure 2a) and b) for PDJ site in the paper are shown below.

[Figure]

Figure. The panels in a) show the scatter plots of hourly measurements of 2 m air temperature (T2m) and b) show 10 m wind speed (WS) compared to observed data from the PDJ station. The figure illustrates the relationship between modeled and observed data.

The vertical patterns below 3 m/s could be addressed by the rounding precision of observational instruments, which is more noticeable at low speeds. At higher speeds, natural variability masks this effect. Additionally, high-altitude locations often experience weaker forcing, making low wind speeds more frequent and emphasizing these patterns.
The 0 m/s values in the WRF-simulated wind speeds are likely due to calm conditions at high altitudes, such as stable layers or weak forcing, where turbulence and mixing are minimal.

Table 4: "Summer (February to March), Autumn (March to June), Winter (June to August)". Why do March and June appear in different seasons simultaneously? How did the authors calculate the seasonal mean with this? Could this be the reason why the maximum and minimum seasonal values mentioned in the opening paragraph of Section 3.3 appear unusual (which I will mention below)?

Thank you for your observation. Initially, the seasonal calculations were based on the standard dates for the beginning and end of each season: summer (December 22 to March 20), autumn (March 21 to June 20), and winter (June 21 to September 22). However, we recognize the confusion caused by overlapping months between seasons, particularly March and June, which appeared in multiple seasons simultaneously in our earlier analysis.
Additionally, during the review of the paper, we identified an error in the model output file that we used to calculate the averages for February to July with that we did all the plots related to them (during the discussing it will be indicate which plot has changed because of this issue).
To ensure clarity and consistency, we adopted the conventional meteorological seasonal grouping: DJF (December-January-February), MAM (March-April-May), and JJA (June-July-August).
As we only have February data representative of the summer season, February will be the sole month considered for the DJF group in our updated analysis. This revised approach eliminates the overlap issue and ensures alignment with standard practices in seasonal analysis.
The following lines have been updated to reflect these changes and corrections: 236–237, 240, 244–246, 254–256, 259–260, and 266–267, as well as Table 4.

Figure 5: The standard deviation in this figure is very difficult to discern. It is recommended to revise the figure, for example, by splitting it into two separate plots or offsetting the data slightly for better clarity.

Thank you for your suggestion. We have revised Figure 5 according to your recommendations and the corrections mentioned above. The figure was splitted into two separate plots, and we changed the colors, legend, and bar widths to enhance the clarity of the results.

Figure 6: The colorbar uses discrete colors. In the figure, do the colors represent the WRF grid values directly, or are they smoothed and interpolated? Currently, the colorbar visually is confusing—for example, the light green–dark green–light green again makes it hard to distinguish values. It is recommended to change the colormap.

Figure 6 represents the WRF grid directly, showing the monthly average concentrations for February and August. We redid Figure 6 by reducing the number of bins in the color map, making it easier to distinguish the concentration values on the map.

Figure 7:

The titles of (g) and (h) are wrong, should be PDG site.

As I mentioned earlier, the IAG and PDJ sites are very close to each other, and the model resolution is coarse. The emission resolution has also not been provided, which makes the simulation values for these two sites nearly identical, which is problematic or less interesting to compare with the two observations.

Why are the biogenic concentrations positive in both summer and winter? This implies that the vegetation acts as an emission source rather than a sink. Given that the authors are using only daytime data (09-17h local as written in Line 301), this seems strange. Could the authors clarify the reasoning behind this? Note that in Line 222, the authors find "the domain acts as a net $CO_2$ sink during summer"!

As we have previously clarified, the distance between the IAG and PDJ sites is 13 km, meaning they are not located on the same model grid. Additionally, we have made a correction for the month of February related outputs from the model. However, we observed distinct daily profiles for each site, primarily due to the contribution of vehicular emissions at IAG and biogenic emissions at PDJ. Regarding the biogenic $CO_2$ concentrations, it is important to clarify that panels (b), (d), (f), and (h) of Figure 7 represent $CO_2$ concentrations under different simulation scenarios: only biogenic fluxes calculated by the VPRM (green line) and only anthropogenic emissions (red line), both added to the

background concentrations (BCK), as described in lines 316-319. The CO2 concentrations simulated solely with the VPRM biogenic fluxes (represented by the green curve) are more pronounced at the PDJ site, both in February and August. However, it is important to note that the CO2 fluxes generated by the VPRM are null for urban areas. Thus, the biogenic CO2 concentrations observed at IAG and PDJ are, in reality, transported from other regions.
As illustrated in Figure 4, the VPRM model calculates a positive NEE for some regions of the domain, even during summer, indicating that these areas act as a net source of CO2 to the atmosphere. This characteristic is evident around the city of São Paulo, while the fluxes remain null within the urban area of São Paulo itself. Additionally, modifications have been made in the manuscript to improve clarity and accuracy. Specifically, the average concentration for February was updated in line 322.
Furthermore, corrections and clarifications were made in lines 324–331 to enhance the text's overall clarity. Also, the titles of (g) and (h) in Figure 7 were corrected and the legend was updated.
We have updated Figure B7 in Appendix B and lines 359-360 to incorporate the correction that we have already discussed previously related to the bias and RMSE to February.

Figure 9: in the caption, it is written "hourly", but the plot is daily.

The legend of Figure 9 was corrected.

My third concern is the analysis and results of the model and observation data comparison. In summary, contrary to the conclusions given by the authors, I do not think that the model's current performance of wind and CO2 is satisfactory. Moreover, many aspects lack reasonable explanations, and some interpretations and analyses seem to be based more on empirical knowledge rather than being clearly supported by the figures provided. Specifically, the issues are as follows:

Section 2.2.1: Please include the distances between observation sites, as well as the inlet heights of the observations at each site above the ground. The lack of these important information will affect the analysis of the comparison between the model and the observation results.

We have added Table 3 in Section 2.2.1, which provides detailed information about the CO2 monitoring network used in this study, including the inlet heights of the observations at each site above the ground. Additionally, we included the distances between observation sites in the text on line 126 for clarity.

Section 3.1: Line 179-180, the authors say that "the GHG model effectively captured significant changes in the observed variables". Also in Line 190, "In summary, the WRF model showed proficiency in reproducing atmospheric conditions in the study area". However, from Figures 2b and 2c, it is clear that the model's wind speed simulation is quite poor. The scatter plots show that for wind speeds < 3 m/s, the model significantly overestimates, and > 3 m/s, it obviously underestimates. Regardless of whatever high and low wind speed, the model's performance is not good. When displaying daily data in 2c, the model consistently overestimates wind speed across different seasons. As mentioned by the authors, wind plays a critical role in CO2 transport, which indicates that there is a clear need for improvement or re-adjustment of the model parameters, rather than claiming it "effectively captured" the observed trends. In Line 187, the authors explain "the model's misrepresentation of land use." However, land use data can be optimized or modified before model runs, particularly since the authors likely ran WRF twice (as mentioned in the VPRM section). If the model's performance was poor, the proper course of action would be to optimize the model first before continuing with the analysis. Additionally, how did the authors handle model parameters for urban areas? Did they use the Urban Canopy Model, e.g., were the building heights adjusted based on local data?

Near-surface (10m) wind speed simulation in the Metropolitan Area of São Paulo has consistently been a challenge. Typically, simulated wind speeds tend to be overestimated compared to

observations. Various boundary layer and surface layer parameterizations have been tested in previous air quality studies (Vara-Vela et al, 2018; Albuquerque et al, 2019; Gavidia-Calderón et al, 2021; Ibarra-Espinosa et al, 2022; Benavente et al, 2023) and based on these findings, we selected the best available options for this research. However, it is important to highlight that most of these studies focused on short-term simulations, typically two to three weeks during ozone buildup events whereas our study covers a six-month period. This extended timescale introduces additional variability that may influence model performance.

Regarding land use data, we addressed this issue by replacing the default WRF land cover dataset with MAPBIOMAS (same land product used in the VPRM model) maps in the initial simulation. However, we acknowledge that further refinements, such as incorporating more detailed urban canopy information, could improve the representation of urban meteorology.

As for urban areas, we did not explicitly use the Urban Canopy Model (UCM) in this study. The inclusion of UCM could enhance wind field representation, particularly in highly built-up regions where building morphology significantly influences wind flow. Future work should explore the integration of UCM and updated land surface data to refine the model's performance in urban environments.

Section 3.3:

First, before analyzing the monthly and seasonal $CO_2$ variations, did the authors compare the hourly simulation results? I would suggest adding a supplementary figure to show this comparison. Without it, important details and the true performance of the model could be hidden by the averaging process.

Thank you for your valuable suggestion. In response to your comment and suggestion, we have added a supplementary figure (Figure B6 in Appendix B) that presents the hourly comparison between the simulated and observed $CO_2$ concentrations and the simulated planetary boundary layer (PBL) at both the IAG and PDJ sites.

The hourly comparison demonstrates that the model reasonably captures the diurnal variability of $CO_2$ at both sites. The simulated $CO_2$ levels generally follow the observed trends, with noticeable peaks and troughs closely aligning with the observations. However, some discrepancies emerge at specific periods, particularly during the early morning and late evening. These differences may be attributed to local emission sources and boundary layer dynamics that are not fully resolved by the model.

At the IAG site, the observed interquartile range is notably wide during nighttime, indicating significant variability in $CO_2$ accumulation. This variability is likely influenced by local atmospheric conditions, such as wind speed and boundary layer height, as illustrated in panel b) of Figure B6. For instance, under low PBL height conditions, mixing ratios exhibit a broad range, from 425 to 470 ppm.

In contrast, at the PDJ site located at a higher elevation and less affected by local anthropogenic emissions the observed interquartile range is narrower. Additionally, the overestimation of wind speed in the WRF-Chem model, as previously discussed, may contribute to discrepancies in simulated $CO_2$ concentrations, particularly during these transitional periods. These findings are consistent with results from Callewaert et al. (2022) in a tropical urban environment.

Indeed, the height of the PBL follows a characteristic diurnal cycle, reaching its maximum in the afternoon, around 14h local time, when surface temperature is highest. This corresponds to the lowest observed surface $CO_2$ concentrations (Fig. B6). Right after sunset, the PBL height drops to its minimum, persisting throughout the night until sunrise. During this period, $CO_2$ concentrations gradually increase as surface emissions accumulate within the stable, shallow nocturnal boundary layer (Fig. B4-h). The highest concentrations occur in the early morning, around 7h local time, when the PBL remains shallow but emissions are high. Once turbulent mixing intensifies later in the morning, concentrations drop sharply, shaping the characteristic diurnal cycle observed at both sites.

[Figure]

Figure B6. Diurnal cycle of in situ CO2 concentration and planetary boundary layer (PBL) height for the entire simulated period. The black line represents the median hourly concentrations from WRF-GHG, while the purple line corresponds to the observed values. The shaded areas indicate the interquartile ranges. Panel a) shows the observed and simulated surface CO2 concentration at the IAG site; b) the simulated PBL height at the IAG site; c) the observed and simulated surface CO2 concentration at the PDJ site; and d) the simulated PBL height at the PDJ site.

Line 255, Figure 7: it is clear that there are significant data gaps in the Picarro observational data. Given that the authors' analysis only covers 6 months, could the authors please specify the data availability for the observational data? Additionally, it seems that throughout the manuscript, the model and observation comparisons do not account for the absence of observational data in certain hours. It would be more reasonable to exclude simulated values during hours without observational data. Failing to do so may lead to significant discrepancies in the model-observation comparisons.

Comparisons were conducted only for time periods where observational data were available. Data gaps in the Picarro observational dataset were present primarily in February, while the other months in the study period had continuous records. To ensure a fair evaluation, simulated values were excluded for hours without corresponding observational data, preventing discrepancies caused by missing observations.

Line 236: the authors say the seasonal variation in CO2 levels is influenced by seasonal patterns of photosynthesis and vehicular traffic. What is (or where can we see) the temporal variation of the vehicular traffic in this manuscript?

We have added Figure B4-h) which shows the diurnal emissions profile by vehicular sources. Lower emissions are observed during the early morning hours, reflecting reduced traffic volumes, while peak emissions occur between 6h-9h local time, coinciding with periods of intense vehicular activity in the

morning. A second peak is observed in the evening, between 16h-18h local time, likely associated with increased traffic volumes as people commute home from work or other activities.

[Figure]

Figure B4 h). Hourly mean emissions profile by the VEIN model for each month at the IAG site.

In Line 233, at IAG station, the authors find that the seasonal variation peak in autumn, then winter, then summer. But in Line 237, same at IAG station, the authors say the monthly peak is in June, during a winter season. Same, the PDJ station peaks in summer in Line 241, and then peak in May a autumn season in Line 243. How could be a same station that reaches two conclusions? Therefore, the average method that the author used might not be reasonable. I think if the authors plot and show the time series of the 6 months of hourly CO2 data, they will figure out why.

We have found an error in the seasonal concentration, which was corrected in the text in lines 236-237, 240, 244-246, and Table 5.

Line 238-240, the authors say "During the summer months, …wind speed… typically lead to lower atmospheric stability". However, if we look at Figure 2, the wind speed looks quite similar and stable over the six months. Thus, is the author's conclusion based on prior knowledge or on the observational wind data from the stations shown in the figure?

This statement is based on prior studies conducted in the same region, as well as on the meteorological variables analyzed in our study. We considered data from four different locations where both simulated and observed meteorological variables were evaluated.
In the MASP, autumn and winter are generally associated with more stable atmospheric conditions due to the frequent influence of high-pressure systems from extratropical latitudes. These systems, often linked to cold front intrusions, contribute to a more stratified atmosphere, reducing vertical mixing. Additionally, thermal inversions are more common during these seasons, further enhancing atmospheric stability (Chiquetto et al., 2024).
During summer, in contrast, the combination of higher solar radiation, increased surface heating, and greater moisture availability leads to enhanced atmospheric instability. This results in more frequent convective activity, stronger vertical mixing, and higher turbulence, which reduce the persistence of stable atmospheric layers (Chiquetto et al., 2024). While Figure 2 shows relatively similar wind speeds throughout the six-month period, atmospheric stability is influenced by multiple factors beyond wind speed alone, such as temperature gradients, boundary layer dynamics, and moisture content (Wallace & Hobbs, 2006).
To improve clarity and accuracy, we updated the sentence:
*"During this month, the MASP experiences changes in synoptic circulation and atmospheric moisture that typically reduce atmospheric stability (Chiquetto et al., 2024)."* (lines 243–244).

Line 260-263: If the model's 3 km resolution is insufficient and the two sites are only 1 km apart, likely they are within the same grid cell, it is impossible to further draw conclusions about emission overestimation or underestimation near either site. This will make the subsequent emission estimates meaningless.

Due to the previously discussed error in the legend of Figure 1, there was a misunderstanding regarding the distance between the two sites (IAG and PDJ). The actual distance between the CO2 observation sites is approximately 13 km (Line 126), with a significant difference in altitude as well. Therefore, the two sites are not within the same model grid cell, and the model's resolution does not impose the limitation suggested in the comment.

Line 313-315: For Fig 7a at IAG site, the authors write "on February 22nd and 23rd, there was a peak in the CO2 concentration of the observed data". However, in the Fig 7a, there are no observation data on these two days!

As previously addressed in the section discussing the second concern, the error regarding the mention of February 22nd and 23rd was corrected in Figure 7. The correction was also reflected in the text, as explicitly mentioned in that section.

Line 317: the authors say "The model effectively captured peaks and profiles for this period". While there are several peaks and daily variations in Fig 7c that are not captured by the model.

As previously discussed in the section addressing the second concern, the correction made in Figure 7 for February impacted the results shown in Figure 7c. We acknowledge that some peaks and daily variations in Figure 7c are not fully captured by the model. However, the model does effectively reproduce the overall trend and major patterns of $CO_2$ variability during this period. The discrepancies may be attributed to unresolved fine-scale processes, local emission sources, or meteorological factors that are not fully represented in the model's resolution, as well as other uncertainties discussed by Lian et al. (2021). To clarify this, we have revised the statement to: "*The model captures the overall trend and major peaks of CO2 variability during this period...*" (lines 332-333).

Line 318: the authors say "the biogenic contributions at PDJ site emerging as more substantial (Figure 7d) compared to the IAG site". However, the biogenic concentrations in Figures 7b and 7d seem quite similar, don't they?

The correction made in Figure 7 for February impacted the results shown in Figure 7d, which influenced the biogenic contribution observed at the PDJ site. Previously, the biogenic contributions at PDJ and IAG appeared similar in Figures 7b and 7d, but after the correction, small differences can now be observed (Lines 333-334).

Line 328: Could the higher monthly average primarily be due to the single-day peak on August 14? Simply stating that the monthly observations are larger than the simulations might not be fully accurate, as Figure 7g shows that the observed values are smaller than the simulated ones for several days after August 24.

We acknowledge that the peak on August 14 influenced the monthly average. However, a more detailed analysis of the data reveals that from August 1 to 24, the observed mean CO2 concentration was 413 ppm, while the simulated mean was 410.5 ppm. On August 14 specifically, the observed concentration peaked at 423.2 ppm, whereas the model simulated 409 ppm. Additionally, from August 24 to 31, the observed mean decreased to 411.8 ppm, while the simulated mean was 417.0 ppm, indicating an overestimation by the model during this period. These results suggest that, although the peak on August 14 contributed to the higher monthly average, the discrepancy between observations and simulations varies throughout the month, particularly in the last week of August. We have updated the manuscript to clarify this point.

We updated the text to "*While the model slightly underestimated the monthly average, it generally captured the observed variability. The higher monthly mean in the observations was influenced by a peak in mid-August, but differences between observed and simulated values varied throughout the*

*month. Before late August, observed values tended to be higher than the simulations, whereas, in the final days of the month, the model overestimated CO2 concentrations. This highlights the role of both biogenic and meteorological processes in shaping CO2 variability at this site (Fig. 7h), emphasizing the importance of considering these dynamics in future simulations.*" (Lines 342-347).

Line 329 and Figure A4: Is this analysis based on daytime data (09:00–17:00)? Are these hourly data? If so, according to the bias and RMSE values, the model's performance in simulating CO2 does not appear to be satisfactory.

This analysis is based on hourly data from 09:00 to 17:00, as shown in Figure B7 (Appendix B), which was updated following the correction made for February. The updated data reflect the period mentioned and address previous discrepancies.
We acknowledge that the model's bias and RMSE values are not satisfactory. However, we would like to emphasize that these values are influenced by multiple factors, as previously discussed in earlier studies (Lauvaux et al, 2019,  Lian et al, 2022). Among these factors, uncertainties in boundary conditions are particularly notable, as global inversion models exhibit significant biases in tropical fluxes, especially in regions where observational data is lacking (Peiro et al., 2022). Additionally, uncertainties in emissions, due to the lack of a detailed local emissions inventory for point sources, contribute to these biases. To address this limitation, we used a global emissions inventory to represent industrial emissions, but this comes with temporal resolution constraints.
These uncertainties are well recognized and cannot be adequately validated due to the lack of observed updated data. However, when comparing the two sites (IAG and PDJ), we observe that for PDJ, which is farther from urban emissions sources, the RMSE and bias values improve considerably, as shown in Figure B7 of the Appendix B. This is consistent with the results from  Lian et al (2022), even with a more robust model testing different PBL schemes and two urban canopy schemes.
In areas farther from urban sources, anthropogenic emissions are lower, and the vertical CO2 concentration gradient generated by urban emissions is smoothed by convection and atmospheric diffusion processes. As a result, the uncertainty about vertical mixing efficiency is lower compared to urban areas, as also seen in the daily cycle shown in Figure B6 of the Appendix B.
Within the city, CO2 concentrations are highly sensitive not only to vertical atmospheric mixing near the surface but also to the temporal profile of anthropogenic emissions, as evidenced by the correlation between CO and CO2 that we analyzed.

Line 345 and Figure 8: For the CO data—or the data depicted in this figure—what time period does it cover? From Figure 1, it is difficult to immediately locate the Pinheiros site.

Thank you for pointing this out. We have clarified in the text that the data shown in Figure 8 correspond from February to August 2019, for all simulated periods in this study. To ensure this is clear, we have updated Line 358 and revised the captions for Figures 8 and 9 accordingly. Additionally, we have updated Figure 1 to provide a clearer zoom on the D01 domain, making it easier to identify the Pinheiros site.

Line 349-350: the "hourly correlation" here is R or R2? I do not think a R with a value of 0.25 shows a good correlation.

Thank you for your comment. We would like to clarify that Figure 8 has been updated to reflect the correction regarding the February period. In addition, we have revised the axis legend to clearly indicate whether the correlation refers to R or R². This should improve clarity regarding the analysis presented.

Line 351-353: The authors mention that the correlation is good before 10 AM and after 19 PM, while the correlation is poor at noon due to the effects of vegetation. However, vegetation also plays a role at night, doesn't it?

We acknowledge that vegetation influences CO2 levels both during the day and at night. However, the mechanisms differ: during daylight hours, photosynthesis actively removes CO2, leading to a decoupling between CO2 and CO emissions. At night, respiration dominates, leading to a net release of CO2. However, this process is more spatially homogeneous compared to anthropogenic sources, which could explain why the nighttime correlation remains relatively stronger. Additionally, Figure 8 has been updated, and we have refined the discussion in the text (Lines 369-370) to better clarify this point.

Line 359-362: According to the authors' analysis, before August 13, both CO2 and CO show peaks, with a large part of the CO2 concentrations at IAG coming from vehicular sources. However, the model struggles to capture this due to less accurate emission data, as emissions follow the same diurnal variation every day of the month. Given this, how should the authors explain the period between August 18 and 28, where only a CO peak is observed? What could account for the absence of a CO2 peak during this period, especially if vehicular emissions are still expected to contribute significantly to CO2 concentrations?

The period from August 18 to 21 was known as "the day that turned into night" due to the transport of smoke from Amazon region fires to São Paulo. The combination of this smoke, cloud cover, and a cold front resulted in an unusually dark daytime atmosphere.
Bencherif et al. (2020) analyzed the long-range transport of smoke plumes from these fires and identified two episodes of increased aerosol and CO concentrations over São Paulo, coinciding with elevated CO levels observed at the Pinheiros station between August 18–21 and August 27–28, as also shown in their study, bellow you can see the periods of this event.

[Figure]

(a) Manicoré      (b) São Paulo      (c) Santa Maria

Figure. CO time evolution during August 2019, over (a) Manicoré (5.8°S, 61.3°W), (b) São Paulo (23.5°S, 46.6°W), and (c) Santa Maria (29.4°S, 53.8°W) sites.
Source: Bencherif et al (2020).

However, this episode had little impact on CO2 concentrations due to the long transport distance. Biomass burning emits both CO and CO2, but the ratio varies with combustion efficiency, with CO being more prevalent in incomplete combustion processes. Additionally, smoke plume transport dynamics play a role: CO is more concentrated at altitudes favorable for long-range transport, whereas CO2 is more influenced by local emissions and atmospheric mixing.
These factors likely explain the observed increase in CO without a corresponding peak in CO2, highlighting the complexity of source attribution in urban air quality studies.

Similarly, Oliveira et al. (2023) investigated the impact of extreme wildfires in Brazilian forests and sugarcane burning in September 2020. Their findings indicated that while wildfires contributed to air pollution, the most significant enhancements related to CO2 concentrations were linked to nearby sources, such as sugarcane burning in the interior of São Paulo state.

We included this discussion in Lines 382-388 "*Additionally, a distinct increase in CO concentrations without a corresponding rise in CO2 was observed between August 18 and 21 and August 27 and 28, which coincided with the long-range transport of smoke plumes from Amazon forest fires to São Paulo (Bencherif et al., 2020). While biomass burning emits both CO and CO2, their atmospheric transport and dispersion differ significantly. CO is more prevalent in incomplete combustion and tends to be transported at altitudes that favor long-range dispersion, whereas CO2 concentrations are more influenced by local emissions and atmospheric mixing (Gatti et al., 2010). These transport dynamics, combined with the long distance of the event's origin, likely explain why the CO peak was detected at Pinheiros but not accompanied by a significant CO2 enhancement at the IAG site.*"

Line 380: The authors are comparing XCO2 in this section. If the surface wind is overestimated, it does not necessarily mean that the winds at higher altitudes are also overestimated, right?

We have revised the sentence to clarify that while surface wind speeds may be overestimated, this does not necessarily imply similar biases at higher altitudes. The revised sentence can be found in Line 398.

Line 381-382: The authors suddenly conclude, without any supporting figures or analysis, that there are "errors in the initial and boundary conditions of concentration provided by the Carbon Tracker." This conclusion is also highlighted in the abstract with the phrase "the large-scale contribution in global models." How did the authors arrive at this conclusion? Could they provide evidence to support this conclusion?

We acknowledge that attributing uncertainties to boundary conditions requires support from previous studies that have evaluated global inversion models. Several studies have demonstrated that global CO2 inversion models exhibit significant biases on monthly timescales, primarily due to large uncertainties in tropical net fluxes (Peiro et al., 2022). In addition, the uncertainties in CO2 lateral boundary conditions can lead to persistent differences in the background CO2 concentration, which may arise from biases in global inversion systems or systematic errors in the interpolation of global CO2 mole fractions to regional lateral boundaries (Chen et al., 2019; Lian et al., 2022; Zhao et al, 2022; Callewaert et al., 2023). Given these known issues in global models, we recognize that errors in initial and boundary conditions could significantly contribute to the discrepancies observed in our study. As suggested we included this references in Line 408-409: "*...errors in the initial and boundary conditions of concentration provided by the Carbon Tracker, which has also been seen in other studies (Chen et al., 2019; Lian et al., 2021; Peiro et al., 2022).*"

Section 4 (Conclusion) mainly introduce what was done in this study and some already well-known prior knowledge (e,g, wind is a pivotal factor; planetary boundary layer dynamics), rather than effectively highlighting the main conclusions of the paper. It lacks a clear synthesis of the key findings and insights that emerge from the analysis.

We have revised the conclusion and as was suggested we incorporate more findings and insights that emerge from this study.

Before in lines 392-403 "The spatial and temporal distribution of modeled CO2 concentrations, stemming from anthropogenic, biogenic, and background emission processes, underwent comprehensive analysis. Wind dynamics emerged as a pivotal factor, underscoring the importance of precise simulation of wind speed, wind direction, and planetary boundary layer dynamics. The

WRF-GHG model adeptly replicated meteorological variables such as temperature, however discrepancies in local wind speed and direction persisted. This can be attributed to the intricate topography and the limited model resolution (3 km), which impedes the capture of nuanced local dynamical processes. Surface $CO_2$ concentrations unveiled distinct diurnal cycles shaped by local anthropogenic emissions, boundary layer dynamics, and vegetation respiration. Importantly, the modeled $CO_2$ concentrations exhibited high sensitivity not only to atmospheric vertical mixing near the surface but also to the prescribed temporal profiles of anthropogenic and biogenic emissions, highlighting the underestimation of vehicular emissions. These sources of error, particularly pronounced in winter, present challenges in accurately quantifying city emissions. In suburban locations such as the PDJ site, distant from urban sources, anthropogenic emissions diminish, and the vertical gradient of $CO_2$ concentration generated by city emissions attenuates through atmospheric convection and diffusion processes."

We have rewritten what you can see in the article's revised in lines 424-435 "*The WRF-Chem model demonstrated skill in simulating meteorological variables, particularly temperature; however, discrepancies in local wind speed and direction persisted. These differences are attributed to the region's complex topography and the model's resolution (3 km), which limits its ability to capture fine-scale dynamical processes. Simulated $CO_2$ concentrations exhibited distinct diurnal cycles influenced by local emissions, boundary layer dynamics, and vegetation fluxes. The model's performance varied between monitoring stations, highlighting the interplay between urban and vegetative environments. At the IAG site, $CO_2$ concentrations were consistently underestimated, with negative biases of -9.17 ppm in February and -12.83 ppm in August. This underestimation was closely linked to the model's difficulty in capturing the impact of high vehicular emission densities, as indicated by the correlation with CO concentrations. Conversely, at the vegetated and elevated PDJ site, the model closely matched observational data, with minimal biases of 0.73 ppm in February and -0.61 ppm in August. In sub-urban locations such as the PDJ site, distant from urban sources, anthropogenic emissions diminish, and the vertical gradient of $CO_2$ concentration generated by city emissions attenuates through atmospheric convection and diffusion processes*."

**Minor comments**

Line 55-58: The manuscript mentions WRF-GHG, WRF-Chem, and WRF when referring to the model. WRF-GHG has existed for a long time and is an older version. However, it was incorporated into WRF-Chem and became a module in WRF-Chem shortly after WRF-GHG was made publicly available. Since the authors are using WRF-Chem V4.0, why are they still referring to WRF-GHG? Did Beck et al. (2011) make any specific modifications to the model? Lines 57–58 do not clarify this point, as the GHG module in WRF-Chem also does not include chemical processes. If the authors merely added a tracer to the model without other significant modifications, it would theoretically still be appropriate to refer to it as WRF-Chem. I suggest the authors standardize the terminology throughout the manuscript to avoid confusion for readers unfamiliar with the model.

As suggested, we have standardized the terminology from WRF-GHG to WRF-Chem throughout the manuscript. Additionally, we have updated Section 2.1 (Model Setup) in lines 56–57 to better clarify the use of WRF-Chem and the GHG tracer module:
"*The WRF-Chem was used to simulate the transport of the mole fraction of $CO_2$, and no chemical processes or reactions have been used (Beck et al., 2013)*"
We have replaced WRF-GHG with WRF-Chem in the following instances: lines 6, 44, 47, 53 (Section 2), the legend of Table 1, and lines 69, 78, 83, 87, 161, 165, 424, 441, and 442.

Table 3: the PARo value changes from 570 to 178615 for Atlantic Forest. Does the author have some explanation? Does this value have actual physical meaning? Is it reasonable? Or is it just a mathematical optimization?

This value is a mathematical optimization aimed at adjusting the parameter of the Atlantic Forest. However, we acknowledge the limitations of this optimization due to the scarcity of observational

data for our region. For this reason, we also included in the conclusion a recommendation for the deployment of additional flux towers and targeted measurement campaigns to better characterize the ecosystems in our study area (Lines 419-421).

Technical corrections

Line 125, Fig 4 caption: typo, it should be PFT as in Line 123, not PTF.

Done.

Line 132: Add S and W to the latitude and longitude.

Done.

Line 191: typo, "wind Direction" D should be small letter.

Done.

Line 196: typo, Figure 3?

Corrected.

Fig 5 caption, Table 4 caption, Line 275, Line 303, Figure 9 caption, etc: CO2 should be CO2

Done.

Line 355: why "both profiles (modeled and simulated CO2)"? modeled and simulated are the same?

Uptated to "*...both the modeled and observed CO2 profiles suggest…*" in line 375.

Line 301: 09-17h local is not only mid-afternoon but daytime.

Done.

Line 340 and 341: typo for Figure A2b and A2d, the authors refer to wrong figures.

Done.

Line 357: reference typo

Done.

Line 376: "positive RMSE"? RMSE is always positive…

Corrected to "*Higher RMSE value*s…" in line 402.

**Anonymous Referee #2**

Major issues: 1) Model setup
Since authors in this study attempted to perform WRF-Chem/GHG simulation in a small domain particularly in urban settings, they didn't configure 3km resolution from a nested domain. If you think configuring nested domains is computationally expensive, you might perform a shorter period (Feb and August only). Nested domain configuration will allow you to perform model evaluation in two different resolutions. In Line#259: How did you establish this statement that model resolution is actually causing the mismatch between simulated and observed concentration at the PDJ site? Chemical IC/BCs should be well representative. If you think chem Carbon Tracker based IC/BCs are good enough for your model setup it needs well justification, but as you mentioned in line# 381-382 why you didn't give a try to see what benefit you will achieve if you use other chem IC/BCs such as CAMS which has more finer resolution than CT? It needs more detailed information for the VEIN and EDGAR products while creating anthropogenic emissions. What is the spatial and temporal resolution and year of representation? Did you combine and sum both VEIN and EDGAR products to get total anthropogenic emissions? At this moment it's not clear how these two inventories are configured in the model. You can also provide a map showing the differences of VEIN and EDGAR products. Are

there any emission scaling factors being implemented in the inventory for diurnal cycle? It's not clear how diurnal variation is imposed in the model. You might need to impose a temporal anthropogenic emission variation following Nassar et al. (2013). Also look into the following paper where they imposed temporal emission factor Callewaert, S., Brioude, J., Langerock, B., Duflot, V., Fonteyn, D., Müller, J.-F., Metzger, J.-M., Hermans, C., Kumps, N., Ramonet, M., Lopez, M., Mahieu, E., and De Mazière, M.: Analysis of $CO_2$, $CH_4$, and CO surface and column concentrations observed at Réunion Island by assessing WRF-Chem simulations, Atmos. Chem. Phys., 22, 7763–7792, https://doi.org/10.5194/acp-22- 7763-2022, 2022. I saw you referring to Gourdji et al. (2022) in a couple of areas but I'm surprised why you didn't user their approach to optimized your VPRM model. In section 2.2.2 (CO2 fluxes data) you should provide more detailed information about the Eddy flux sites used in the VPRM optimization process (perhaps in the supplementary or appendix).

The same concern regarding model resolution and the use of nesting to improve simulations was also raised by Reviewer 1 and has already been discussed in the *Major Comments – Model Setup* section, as well as the issue of boundary conditions. Additionally, we tested the CAMS boundary conditions, but the results were less representative compared to those from Carbon Tracker. In the figure below, we present a sensitivity test conducted for the month of August, where we used the same configuration for both tests and the same emission files, changing only the boundary and initial conditions. The red curve (Simulation_egg4) represents the CAMS data (*CAMS global greenhouse gas reanalysis – EGG4*), which has a horizontal resolution of 0.75° × 0.75°, while the gray curve (Simulation_CT) refers to the Carbon Tracker data.

[Figure]

**Figure.** Sensitivity test comparing simulated CO2 concentrations using CAMS (BC/IC) and Carbon Tracker (BC/IC) boundary conditions for August 2019 at the IAG site, alongside observed concentrations.

Regarding anthropogenic emissions, we acknowledge that some important aspects need further clarification. We have added a detailed explanation regarding the spatial resolution of the VEIN model in the manuscript and clarified that the VEIN model produces spatially and temporally resolved emissions based on high-resolution inputs, such as traffic flow, urban morphology, and emission factors (Lines 78–84). To ensure consistency with the WRF-Chem model, VEIN outputs were aggregated to a resolution of 3 km, corresponding to the study domain. Additionally, we have included a new figure (Fig. B4) in Appendix B, illustrating the spatial distribution of vehicular emissions in the region. This figure supports discussions about $CO_2$ simulation performance and the influence of vehicle emissions.
We also specified in the manuscript that we used the EDGAR v6.0 GHG inventory (Crippa et al., 2021) for the year 2018, which provides a spatial resolution of 0.1° × 0.1°. We detailed the data processing steps, including conversion to NetCDF format, horizontal resizing to 3 km resolution, and

temporal adaptation for the simulation period. These details have been incorporated into the methodology section (Lines 82–84).

Additionally, while VEIN data includes hourly temporal profiles derived from local traffic data, EDGAR emissions are assumed to be temporally invariant due to the lack of hourly resolution in the original dataset. The revised manuscript acknowledges this limitation and now includes a spatial distribution of EDGAR emissions for the study region in Appendix B (Fig. B5). After processing, both VEIN and EDGAR emissions were combined and summed to generate the total anthropogenic emissions used as input for WRF-Chem.

Regarding VPRM optimization, we acknowledge that our approach is not ideal, and it would have been beneficial to follow the same methodology as Gourdji et al. (2022). However, due to the limited number of flux measurement sites (only three towers covering different ecosystems) and the need for a full seasonal cycle over a year to constrain the VPRM model, we were unable to set aside an independent subset for evaluation.

To address this, we followed the recommendations of previous studies (Mahadevan et al., 2008; Ahmadov et al., 2007; Xiao et al., 2014), adapting them to our specific context and constraints by using available observational datasets and evaluating model performance with non-independent measurements.

While this approach is not ideal, we optimized parameters over the entire year, using MODIS EVI as a proxy for photosynthesis—an important simplification for evergreen ecosystems like the Atlantic Forest—and incorporating soil moisture to represent water availability, which is a key limitation for tropical ecosystems. As shown in Figure 3, certain periods exhibit poor performance, indicating that despite optimization, the VPRM model does not always fully capture seasonal dynamics.

To clarify this point and balance our conclusions, we have added the following text in the paper to recommend additional deployment of flux towers and possibly dedicated measurement campaigns to improve the characterization of ecosystems in the region. This recommendation has been included in the conclusion section (Lines 409–421). The explanation about the eddy covariance sites is in Section 2.2.2, and the sites shown in Figure 4 (a) and (b) correspond to the flux towers used for VPRM optimization.

2) Analyzing model outputs
Meteorological output such as wind speed and direction should be presented with a wind rose figure. Wind rose figures will help to interpret the sources of $CO_2$ in a site. In many places you are talking about atmospheric stability but there is no analysis to support atmospheric stabilitybased conclusions (such as section 3.3.1). You can perform PBL height related $CO_2$ profiles at two sites (IAG and PDJ). I'm not convinced with the conclusion made in this paragraph (line#345 to 354 – source contribution), because: Not sure the hour format in Fig. 8. Is this UTC or local hour? If midday has less vehicles how do you expect late night to have high vehicle emissions and cause high correlation between $CO_2$ and CO? Again, it's important to provide some evidence of the diurnal cycle in the VEIN or EDGAR product. In line#236: Do you have any information on the seasonal variation of traffic emission? Seasonal wind rose plot will help you to establish the wind speed related statement written in this line. Abstract is not well written and there is only one conclusion in the abstract without any quantitative evidence.

Thank you for your comments and suggestion. Regarding Figure 8 has been revised to include local time information. Additionally, Figure B4 has been added to Appendix B, showing the spatial distribution of vehicle emissions and their temporal variation at the IAG site based on the VEIN model. The higher vehicle emissions occur between 07:00–09:00 and 16:00–18:00 local time. Furthermore, Figure B5 presents the spatial distribution of industrial emissions from EDGAR. We have added a supplementary figure (Figure B6 in Appendix B) that presents the hourly comparison between the simulated and observed $CO_2$ concentrations and the simulated planetary boundary layer (PBL) at both the IAG and PDJ sites.

Indeed, the height of the PBL follows a characteristic diurnal cycle, reaching its maximum in the afternoon, around 14h local time, when surface temperature is highest. This corresponds to the lowest observed surface $CO_2$ concentrations (Fig. B6). Right after sunset, the PBL height drops to its minimum, persisting throughout the night until sunrise. During this period, $CO_2$ concentrations

gradually increase as surface emissions accumulate within the stable, shallow nocturnal boundary layer (Fig. B4-h). The highest concentrations occur in the early morning, around 7h local time, when the PBL remains shallow but emissions are high. Once turbulent mixing intensifies later in the morning, concentrations drop sharply, shaping the characteristic diurnal cycle observed at both sites. Below, wind roses are provided for the IAG and PDJ sites, comparing February and August 2019.

February 2019

[Figure]

August 2019

[Figure]

The wind roses show the climatological patterns in the Metropolitan Area of São Paulo. Throughout the year, the prevailing wind is from the Southeast. In February, prevailing winds are both from the Northwest and Southeast, showing the sea breeze influence (Northwestern winds are pre-sea breeze front and Southeastern winds are after the sea breeze enters São Paulo). In August, the pre-sea breeze

front is not so evident, but the entrance of the sea-breeze is well defined. In August, Southeastern winds are also due to the Extratropical Anticyclones that follow cold front systems. In February, some transport of $CO_2$ from the city center to PDJ is visible. The IAG site is almost inside São Paulo city, surrounded by vehicular emissions, and thus, source attribution is not easily depicted in the wind roses.

3) Figures

Most of the figures in the current manuscript are not publication standard. Figure captions are not well described. For instance, four panels in Fig. 1 but not clear which one is for what; no panel number for WS and WD panels in Fig. 2. Please elaborate captions in all figures and make clear for all legends.

Thank you for your comment. In order to improve the figures in the manuscript, we have made the following revisions:

**Figure 1**: We have restructured the figure to clarify the information and provided a more detailed caption. The figure now includes data on terrain elevation (m) and land use for the domain (D01) used in this study. The legend specifies the classification of the stations through different symbols (triangle, circle, and cross) and colors (green, red, and black), each corresponding to the type of measurement performed (meteorological variables, $CO_2$, or CO).

**Figure 2**: The figure has been revised by adding axes and standardizing the scatter plots to better align with common practices for visualizing such data. The caption has been rewritten to provide a clearer explanation of what each panel represents: "*The panels in a) show the scatter plots of hourly measurements of 2 m air temperature (T2m) and b) show 10 m wind speed (WS) compared to observed data from the PDJ station. The figure illustrates the relationship between modeled and observed data. The panels in c) show the daily averages from February to August 2019 of 2 m air temperature (T2m), 10 m wind speed (WS), and wind direction (WD). Black line represents the observed data and red line represents the model simulation*". We also included panel numbers for the WS and WD.

**Figure 3**: We have changed the color of the VPRM_optimized curve (from red to green) and added axes for better clarity. The caption has also been updated to provide a more detailed explanation: "*Daily variability of NEE fluxes (µmol m−2 s−1) from the flux tower (black line), compared with NEE fluxes simulated by the VPRM model using default (red line) and optimized (green line) parameters for the Atlantic Forest, Cerrado/Savanna, and Sugarcane*". The corresponding sentence in the manuscript (line 211) has been rewritten to: "*(VPRM_optimized, shown as the green curve in Figure 3).*"

**Figure 5**: We split the figure into two plots, one for each site (IAG and PDJ), to better represent and highlight the differences between the model and the observations for each month, as well as their variations.

**Figure 6**: We adjusted the number of bins in the color bar to prevent any misinterpretation of the colors representing $CO_2$ concentrations.

**Figure 7**: The caption has been revised to explain the meaning of the abbreviations BCK, VPRM, and ANTH for the curves representing the simulations of these scenarios:

"*Daily mean $CO_2$ concentrations simulated and observed for the IAG site in February 2019 (a), for the PDJ site in February (c), for the IAG site in August (e), and for the PDJ site in August (g). Additionally, the daily simulated concentrations at the BCK (background), VPRM (biogenic), and ANTH (anthropogenic) scenarios for the IAG site during February (b), for the PDJ site in February (d), for the IAG site during August (f), and for the PDJ site in August (h).*"

**Figure 8**: We revised the x-axis label to "Hour (Local time)" and updated the figure legend to:

"Hourly correlation between $CO_2$ concentrations observed at the IAG site and CO concentrations observed at the Pinheiros site (blue bars), and between simulated $CO_2$ concentrations at the IAG site and observed CO concentrations at the Pinheiros site (orange bars) for the period from February to August 2019."

**Figure 9:**This figure has been redone to better conform to publication standards. The caption has been rewritten to enhance clarity and completeness of the information presented.

**Appendix B:**

Figures B1, B2, and B3: These figures have been revised following the same approach as Figure 2, but adapted for the other measurement locations.

Figures B4 and B5: These new figures have been added to provide complementary information that was previously missing from the manuscript, in response to concerns raised by Reviewers 1 and 2 regarding the spatial and temporal distributions of anthropogenic emissions.

Figure B6: This figure has been included to enhance the analysis of CO2 concentrations for both locations, as requested by Reviewers 1 and 2.

Figure B7: This figure has been revised solely to correct an error related to the simulated data outputs for February.

Minor comments:

I'm confused with line#86 and #139 because in #86 it is indicated meteorological variables will come from wrf outputs but in line#139 it is written VPRM model is driven by the meteorological measurements of the sites. Please make it clear.

In Section 2.1.2 (Biogenic Fluxes), the VPRM model is applied across the entire study domain (D01) using meteorological variables from WRF model outputs, such as T2m in the respiration equation (Equation 3) and downward shortwave radiation (PAR) in Equation 2. This section refers to VPRM as an input to WRF-Chem.

In Section 2.2.2 (CO2 Fluxes Data), we refer to the flux data used to optimize VPRM parameters, which are applied specifically to the three PFTs (Atlantic Forest, Cerrado, and Sugarcane). Here, VPRM is applied only at the flux tower locations, using measured meteorological variables such as temperature and shortwave radiation. These measurements come directly from the flux towers and are used for the available data period. Once the VPRM parameters are optimized for these PFTs, they are then applied to the entire D01 domain in the same manner as described in Section 2.1.2.

To further clarify this distinction, we have renamed Section 2.2.2 to "*CO2 Fluxes and VPRM Optimization*", explicitly indicating that this section refers to the VPRM parameter optimization for the three PFTs.

How did you decide 5 days would be good enough for spin up? Please give a justification.

We conducted test simulations and analyzed the first few days of output. As shown in the Figure below, CO2 concentrations stabilize after 3–4 days, suggesting that a 5-day spin-up period is sufficient.

[Figure]

Figure. CO2 concentration time series at the IAG site for February 2019. The model (purple) stabilizes within the first few days, justifying the chosen spin-up period. Observed data (black) is also shown for comparison.

Table 2, better to write full name of the variables (T2, Wd, Ws etc.) in the table footnote. I think it's more relevant to classify PDJ as Sub-urban park and IAG as University campus/urban park.

In response to your comment on Table 2, we have added the full names of the variables (T2, Wd, Ws) in the table 2 footnote to ensure clarity.

Regarding the classification of PDJ and IAG, we have adopted the classification of PDJ as a park and IAG as an urban park following the nomenclature used in previous studies conducted in the region, such as Benavente et al. (2023).

Section 2.2.2 (CO2 fluxes data) should be merged with section 2.1.2 or vice versa.

Thank you for your suggestion. However, we prefer to keep these sections separate because Section 2.1 refers specifically to the inputs used in the WRF-Chem model for the period from February to August 2019. To avoid any misunderstandings regarding the methodology, the variables used, and the time period considered, we believe it is clearer to keep them as distinct sections.

Nonetheless, to improve clarity, we have added a reference at the end of Section 2.1.2 linking to the VPRM optimization process in Section 2.2.2 (lines 104–105):

*"The VPRM parameters ($\lambda$, $PAR_0$, $\alpha$, $\beta$) were optimized against flux tower NEE for the main land cover type over the study domain, as described in Section 2.2.2."*

NEE should have a standard unit system (PgC/yr or TgC/yr) and be consistent throughout the manuscript. Currently, two different units are introduced for NEE in Fig. 3 and Fig. 4.

Since our study does not cover multiple years—where the suggested unit system (PgC/yr or TgC/yr) is more commonly used, as in Raju et al. (2023)—we find it more appropriate and conventional to use $\mu mol\ m^{-2}\ s^{-1}$. Our analysis spans February to August 2019, and even the $CO_2$ flux dataset used for VPRM optimization does not extend beyond a single year. Therefore, adopting $\mu mol\ m^{-2}\ s^{-1}$ aligns with prior studies such as Mahadevan et al. (2008), Ahmadov et al. (2007), Gourdj et al. (2022), and Dayalu et al. (2018).

Regarding the figures:

Figure 3 represents the daily variability of net CO2 flux over a time series for each PFT, which is conventionally expressed in $\mu mol\ m^{-2}\ s^{-1}$.

Figure 4:

Panels (a) and (b) display the monthly mean diurnal cycle of NEE for February and August, respectively.

Panel (c) shows the daily variability of NEE in these months for each PFT (Atlantic Forest, Cerrado/Savanna, and Sugarcane), which is also typically reported in $\mu mol\ m^{-2}\ s^{-1}$.

To ensure clarity and consistency, we have updated Figure 4 caption to explicitly include the units:

*The first panel shows the monthly mean diurnal cycle of net ecosystem exchange (NEE) (mol km$^{-2}$ h$^{-1}$) for February (a) and August (b) 2019. The second panel (c) presents the daily variability of NEE ($\mu mol\ m^{-2}\ s^{-1}$) for the same months (February and August) at three different plant functional types (PFTs): Atlantic Forest, Cerrado/Savanna, and Sugarcane.*

Line#221: Is this statement correct? I see barely green colored in the August map (Fig. 4b).

Yes, the statement is correct. In August, CO2 fluxes are predominantly lower than in February (panel a). The faint greenish tint in the August map (Fig. 4b) indicates NEE flux values closer to zero, particularly inland, where agriculture and grasses dominate (as shown in Figure 1). In contrast, the coastal region exhibits a stronger and more predominant reddish tint, highlighting areas with higher positive NEE fluxes.

Line:#223: only vegetation? Line#227 to line#230: Please use quantitative comparison between your study and Raju et al. (2023). Therefore, I suggested following the methodology of Gourdji et al. (2022) while using Eddy flux data for optimizing VPRM.

The discussion in line 233 specifically refers to the net fluxes from vegetation as calculated by the VPRM model. We recognize that it would be ideal to another approach to optimize the VPRM parameters as Gourdji et al (2022). However, due to the limited availability of flux measurement sites (only three towers covering different ecosystems) and the need for a full seasonal cycle over a year to

effectively constrain the VPRM model, we were unable to reserve an independent subset for evaluation.
We leveraged available observational datasets and assessed model performance using non-independent measurements. Although this approach is not ideal, we optimized parameters over the entire study period, utilizing MODIS EVI as a proxy for photosynthesis—a useful simplification for evergreen ecosystems like the Atlantic Forest—and incorporated soil moisture to account for water availability, a key constraint in tropical ecosystems.

Be consistent with Pico do Jaragua and PDJ.

We have changed from Pico do Jaraguá to PDJ to be more consistent in table 2 and 3, lines 115, 126 and 176.

Spatial map: Fig 6: Adding a third row with the differences between first and 2nd row would be great. Line 276: coastal region southwest of MASP -> southeast??

Regarding this comment, we have revised the paragraph related to this analysis to provide additional information and improve clarity.
*"Although the VPRM model did not explicitly calculate CO2 fluxes in urban areas due to limited vegetation coverage, the transport of biogenic signals from the surrounding vegetated regions into the urban area is evident. The southwest region of the domain, characterized by the Atlantic Forest, exhibits the highest CO2 concentrations in this scenario, ranging from 420 to 424 ppm. This dense vegetation region and higher ecosystem respiration contribute to elevated CO2 levels, underscoring the influence of biogenic sources on regional concentration patterns. This region has altitudes lower than 200 m and the CO2 released to the atmosphere by the vegetation is trapped due to the Serra do Mar, with altitudes higher than 500 m. The Atlantic Forest present on the northern coast, on the other hand, is concentrated on the plateau of Serra do Mar, and thus, the CO2 released is better dispersed to other areas."* (Lines 280-288).

Line 312: how do you confirm this is related to vehicular emission?

We acknowledge that, as previously presented, the connection between the observed patterns and vehicular emissions was not sufficiently substantiated for the reader. To address this, we have now included additional figures illustrating the spatial and temporal distribution of anthropogenic emissions used in this study. Specifically, we provide maps detailing the spatial and temporal distribution of vehicular emissions as well as the spatial distribution of industrial emissions. These additions help clarify the contribution of different emission sources and provide stronger support for our interpretation.

Line 317-318: what do you mean by profile? Did you perform any altitude related comparison? Also, I don't see the simulated concentrations are consistent with the observed data at Fig. 7c.
As mentioned earlier, all figures related to the February period have been revised. Additionally, to address Reviewer 1's comments, we have already modified this sentence to: *"At the PDJ site, the mean observed and simulated CO2 concentration for the study period was 414 ppm. The model captures the overall trend and major peaks of CO2 variability during this period, with biogenic contributions being more pronounced at PDJ compared to the IAG site (Figure 7d)."* (Lines 332–334).

Line# 149: Osterman et al. -> year??

It has been corrected in line 155.

Line# 151: I think satellite measurement time was also matched with the model data during interpolation. Please mention this information. But, if measurement time was not matched please do that.

The simulated data were selected to match the OCO-2 overpass time (13:30 LT) as closely as possible for our study region, ensuring consistency in the comparison. This information has now been explicitly mentioned in the manuscript in Lines 157-159: *"Additionally, to ensure consistency in the comparison, the simulated data were selected to correspond as closely as possible to the OCO-2 overpass time (13:30 Local Solar Time) over the study region"*.

Line#177:178: Acronym should be consistent throughout the manuscript. For instance, TM, WS and WD in these lines are different than Table 2.

We padronized the 2m air temperature as $T_{2m}$, wind speed as WS and wind direction as WD. It has changed in Table 2, line 91 and equation 3.

**Textual Updates**

**Line198: "The optimization results are shown in **Table 4**."**

**Line236: "Figure 5 and **Table 5** depict the monthly mean…"**

**Line185: "... (see figure **B1**a and **B2**a in Appendix)..."**

**Line187: "(Figure **B3**a in Appendix)..."**

**Line400: "... (depicted by red dotted lines in **Figure B8 in Appendix B**)..."**

**Structural Additions**

Appendix A: Metrics evaluation

Appendix B: Supplementary figures

**New References Included**

Lian, J., Bréon, F. M., Broquet, G., Lauvaux, T., Zheng, B., Ramonet, M., ... & Ciais, P. (2021). Sensitivity to the sources of uncertainties in the modeling of atmospheric CO 2 concentration within and in the vicinity of Paris. Atmospheric Chemistry and Physics, 21(13), 10707-10726.

Chen, H. W., Zhang, F., Lauvaux, T., Davis, K. J., Feng, S., Butler, M. P., & Alley, R. B. (2019). Characterization of regional‑scale CO2 transport uncertainties in an ensemble with flow‑dependent transport errors. Geophysical Research Letters, 46(7), 4049-4058.

Chiquetto, J. B., Machado, P. G., Mouette, D., & Ribeiro, F. N. D. (2024). Air quality improvements from a transport modal change in the São Paulo megacity. Science of The Total Environment, 945, 173968.

Peiro, H., Crowell, S., Schuh, A., Baker, D. F., O'Dell, C., Jacobson, A. R., ... & Baker, I. (2022). Four years of global carbon cycle observed from the Orbiting Carbon Observatory 2 (OCO-2) version 9 and in situ data and comparison to OCO-2 version 7. Atmospheric Chemistry and Physics, 22(2), 1097-1130.

**Acknowledgements Update**

"...the FAPESP (process number 2016/18438-0 and *2021/11762-5,.., the National Institute of Science and Technology – INCT Klimapolis , which is funded by the Brazilian Ministry of Science..*"

**References Used to Revise the Manuscript**

The following references were consulted during the revision process:

Ahmadov, R., Gerbig, C., Kretschmer, R., Koerner, S., Neininger, B., Dolman, A. J., & Sarrat, C. (2007). Mesoscale covariance of transport and $CO_2$ fluxes: Evidence from observations and simulations using the WRF‑VPRM coupled atmosphere‑biosphere model. *Journal of Geophysical Research: Atmospheres*, *112*(D22).

Bencherif, H., Bègue, N., Kirsch Pinheiro, D., Du Preez, D. J., Cadet, J. M., da Silva Lopes, F. J., ... & Clerbaux, C. (2020). Investigating the long-range transport of aerosol plumes following the Amazon fires (August 2019): a multi-instrumental approach from ground-based and satellite observations. *Remote Sensing*, *12*(22), 3846.

Benavente, Noelia Rojas, et al. "Air quality simulation with WRF-Chem over southeastern Brazil, part I: Model description and evaluation using ground-based and satellite data." *Urban Climate* 52 (2023): 101703.

Callewaert, S., Brioude, J., Langerock, B., Duflot, V., Fonteyn, D., Müller, J. F., ... & De Mazière, M. (2022). Analysis of $CO_2$, $CH_4$, and CO surface and column concentrations observed at Réunion Island by assessing WRF-Chem simulations. *Atmospheric Chemistry and Physics*, *22*(11), 7763-7792.

Callewaert, S., Zhou, M., Langerock, B., Wang, P., Wang, T., Mahieu, E., & De Mazière, M. (2023). A WRF-Chem study on the variability of $CO_2$, $CH_4$ and CO concentrations at Xianghe, China supported by ground-based observations and TROPOMI. *EGUsphere*, *2023*, 1-37.

Chen, H. W., Zhang, F., Lauvaux, T., Davis, K. J., Feng, S., Butler, M. P., & Alley, R. B. (2019). Characterization of regional‑scale $CO_2$ transport uncertainties in an ensemble with flow‑dependent transport errors. *Geophysical Research Letters*, *46*(7), 4049-4058.

Chiquetto, J. B., Machado, P. G., Mouette, D., & Ribeiro, F. N. D. (2024). Air quality improvements from a transport modal change in the São Paulo megacity. *Science of The Total Environment*, *945*, 173968.

de A. Albuquerque, T. T., West, J., de F. Andrade, M., Ynoue, R. Y., Andreão, W. L., Dos Santos, F. S., ... & Moreira, D. M. (2019). Analysis of $PM_{2.5}$ concentrations under pollutant emission control strategies in the metropolitan area of São Paulo, Brazil. *Environmental Science and Pollution Research*, *26*, 33216-33227.

Gatti, L. V., Miller, J. B., D'amelio, M. A., Martinewski, A., Basso, L. S., Gloor, M. E., ... & Tans, P. (2010). Vertical profiles of $CO_2$ above eastern Amazonia suggest a net carbon flux to the atmosphere and balanced biosphere between 2000 and 2009. *Tellus B: Chemical and Physical Meteorology*, *62*(5), 581-594.

Gavidia-Calderón, M. E., Ibarra-Espinosa, S., Kim, Y., Zhang, Y., & Andrade, M. D. F. (2020). Simulation of $O_3$ and $NO_x$ in Sao Paulo street urban canyons with VEIN (v0. 2.2) and MUNICH (v1. 0). *Geoscientific Model Development Discussions*, *2020*, 1-32.

Lauvaux, T., Díaz-Isaac, L. I., Bocquet, M., & Bousserez, N. (2019). Diagnosing spatial error structures in $CO_2$ mole fractions and $XCO_2$ column mole fractions from atmospheric transport. *Atmospheric Chemistry and Physics*, *19*(18), 12007-12024.

Lian, J., Bréon, F. M., Broquet, G., Lauvaux, T., Zheng, B., Ramonet, M., ... & Ciais, P. (2021). Sensitivity to the sources of uncertainties in the modeling of atmospheric $CO_2$ concentration within and in the vicinity of Paris. *Atmospheric Chemistry and Physics*, *21*(13), 10707-10726.

Mahadevan, P., Wofsy, S. C., Matross, D. M., Xiao, X., Dunn, A. L., Lin, J. C., ... & Gottlieb, E. W. (2008). A satellite‑based biosphere parameterization for net ecosystem $CO_2$ exchange: Vegetation Photosynthesis and Respiration Model (VPRM). *Global Biogeochemical Cycles*, *22*(2).

Monk, K., Guérette, E. A., Paton-Walsh, C., Silver, J. D., Emmerson, K. M., Utembe, S. R., ... & Cope, M. E. (2019). Evaluation of regional air quality models over Sydney and Australia: Part 1—Meteorological model comparison. *Atmosphere*, *10*(7), 374.

Peiro, H., Crowell, S., Schuh, A., Baker, D. F., O'Dell, C., Jacobson, A. R., ... & Baker, I. (2022). Four years of global carbon cycle observed from the Orbiting Carbon Observatory 2 (OCO-2) version 9 and in situ data and comparison to OCO-2 version 7. *Atmospheric Chemistry and Physics*, *22*(2), 1097-1130.

Souto-Oliveira, C. E., Marques, M. T., Nogueira, T., Lopes, F. J., Medeiros, J. A., Medeiros, I. M., ... & Andrade, M. D. F. (2023). Impact of extreme wildfires from the Brazilian Forests and sugarcane burning on the air quality of the biggest megacity on South America. *Science of the Total Environment*, *888*, 163439.

Vara‑Vela, A., de Fátima Andrade, M., Zhang, Y., Kumar, P., Ynoue, R. Y., Souto‑Oliveira, C. E., ... & Landulfo, E. (2018). Modeling of atmospheric aerosol properties in the São Paulo metropolitan area: impact of biomass burning. *Journal of Geophysical Research: Atmospheres*, *123*(17), 9935-9956.

Xiao, J., Ollinger, S. V., Frolking, S., Hurtt, G. C., Hollinger, D. Y., Davis, K. J., ... & Suyker, A. E. (2014). Data-driven diagnostics of terrestrial carbon dynamics over North America. *Agricultural and Forest Meteorology*, *197*, 142-157.

Zhao, X., Chen, J., Marschall, J., Gałkowski, M., Hachinger, S., Dietrich, F., ... & Gerbig, C. (2022). Understanding greenhouse gas (GHG) column concentrations in Munich using WRF. *Atmospheric Chemistry and Physics Discussions*, *2022*, 1-30.

---

## Editor Decision (ED1)

Thank you to the authors for the revised manuscript and their responses to the comments. Some of the comments have been resolved. However, after reviewing the revised manuscript, I find that it still contains several issues that need to be addressed. Some explanations in the results section are not really convincing, and there are noticeable discrepancies between the textual analysis and the figures, with several obvious errors. Additionally, many sentences are redundant or have syntax errors, and there are several typos throughout the text. The quality and color of the figures also need improvement. Therefore, I cannot recommend the manuscript for publication, and major revisions are still required. I strongly urge the authors to carefully proofread and revise the manuscript before the next submission, as it should not be the reviewers' responsibility to identify all obvious and elementary errors.

**Main:**

1. **Some explanations in the results section are not entirely correct or convincing.**
   - Line 219-220, Line 228: The description is inconsistent and could easily lead to misunderstanding. Line 219 mentions that summer is "less intense in the Atlantic Forest," while Line 228 states "negative net fluxes in February, particularly in the Atlantic Forest." These statements seem contradictory.
   - Figure 4b and 4c: Why does Figure 4b show the Atlantic Forest as red (positive), while in Figure 4c, the site clearly shows significant negative values during the day, which exceed the positive values at night? If the monthly average of the NEE from Figure 4c is calculated, would it still be positive as shown in Figure 4b?
   - Line 268-269: The small absolute value of the PDJ observation-simulation bias is due to the small $CO_2$ signal at this site. If the goal is to discuss or compare the simulation performance with the IAG site, a more reasonable approach would be to look at the relative error, e.g., signal-to-bias, rather than directly concluding that "model predictions are more accurate at PDJ."
   - Section 3.3.1: Line 271-277: As far as I know, in WRF-Chem, background, anthropogenic, and biogenic emissions are three separate variables (CO2_BCK, CO2_ANT, CO2_BIO) in the output netcdf files. The authors can simply display them individually or sum them to achieve the desired result, without the need to rerun simulations with different "emission scenarios" as stated. Did the authors modify some part of the model that requires re-simulation?
   - Line 346 and Figure 7h: The authors say, "This highlights the role of both biogenic and meteorological processes in shaping $CO_2$ variability at this site," but from Figure 7h, the increments for biogenic and anthropogenic emissions appear to be similar, and it seems that the increase is due to the rise in background concentrations. How does this align with the authors' statement?

- Section 3.3.2, in Line 361-363: Actually, I believe the analysis in Lines 349-363 is meaningless, especially the conclusions in Lines 361-363. When comparing observations and simulations, of course, the simulation values should consider anthropogenic, biogenic, and background emissions together, as this more closely reflects real-world conditions. Isn't this something that should be done? It's common sense. Why conduct so much analysis just to conclude that "simulating with only one factor leads to larger discrepancies with observations, while considering all three factors improves simulation performance"? If the authors' goal is to analyze the individual contributions of the three factors, then Figures 6 and 7 already serve that purpose.
- Line 371-372: Figure B4 shows that the evening rush hour for traffic occurs before 19:00, not after. How do the authors explain this?

2. **There are several discrepancies between the textual analysis and the figures, with several obvious errors.**

- Line 264: The simulated PDJ values are higher than the observed values. Why do the authors say "underestimated vehicular emissions in these areas"?
- Line 328 and Figure 7a: From Figure 7a, it appears that the IAG site has no observation data for February 21 and 22!
- Line 322 and Line 339: The IAG site has a model-to-observation difference of 8 ppm in February (Line 322) and 13 ppm in August (Line 339). The error in August is larger than that in February. Why do the authors say in Line 339 "only 13 ppm, i.e., a closer approximation compared to February"?
- Line 341-342: observation is 412 ppm, model is 412 ppm in the text. Why do the authors say that the observation is surpassing the simulation?
- Line 351-352: This is a clear mistake! It should be February, not August. Also, it's not Figure 7c, and it should be a positive bias, not a negative bias.
- Line 356-357: clear mistake! Figure B7 shows that the PDJ for only Anthropogenic emissions does not have the poorest RMSE.
- Line 358: clear mistake! Figure B7 shows that IAG with anthropogenic sources in August does not have the highest RMSE.

3. **Many sentences throughout the manuscript are redundant or have syntax errors. I recommend that the authors read through the entire text and remove redundant sentences. Please see some examples bellows:**
- Line 44-47: "Coupled VPRM" and "integrated VEIN" are parallel in structure, but the sentence is too long. The intended meaning is that the VEIN model is integrated into the VPRM model. I suggest rewriting the sentence for clarity.
- Line 67-70: You can directly state "$CO_2$ initial and boundary conditions" instead of repeating it twice "initial and boundary conditions" in the sentence.

Additionally, I recommend using "$CO_2$" instead of "chemical," also change it in Table 1.

- Line 83-84: "EDGAR lacks temporal variability" and "inventory does not provide hourly profile" are repetitive.
- Line 87: "as a flux input" and "as input data" are repetitive.
- Line 171-172: syntax errors, incomplete sentence.
- Line 251-252: The term "PDJ" appears twice in the same sentence, making it redundant.
- Line 267-268: The expression is redundant, as the parts before and after "and" convey the same meaning.

4. **The quality and color of the figures are not visually easy to get information.**

- The color scheme in Figure 1 is not visually appealing and makes it difficult to identify the stations. Additionally, I recommend using (a), (b), etc., instead of referring to "first panel" and "second panel" to improve clarity. Similarly, see Line 278 to avoid unclear descriptions.

- Why are the figure numbers in the supplementary material not assigned in the order in which they appear in the manuscript?

- Line 213 and Figure 4c: Figure 4c shows hourly data, not daily data, right?

- Figure 4a: Figure 4a shows the monthly mean NEE, not the "monthly mean diurnal cycle" as stated in the caption. How can the diurnal cycle be observed from Figure 4a?

- Figure 6: Why use a discrete colorbar instead of a continuous one? This makes it difficult to distinguish values like 6 ppm in Line 280 and 8 ppm in Line 295. Based on the current colorbar, one could also interpret the value as 4 ppm, right?

- Figure B4 and B5: Please include the latitude and longitude. Additionally, there is no need to display different months as it is difficult to discern any significant differences. It would be better to show just one figure of spatial distribution and use other types of charts to present the monthly emission totals. Moreover, the caption for Figure B5 is not accurate, e.g., "daily mean"? which sectors from EDGAR?

- Figure 9: redundant in caption "Daily mean concentrations of CO2 observed concentrations"

**Specific:**

- Line 48: What does "smoothed" XCO2 mean? How is the $XCO_2$ from WRF-Chem smoothed?

- Section 2.1.1: What is the total anthropogenic emission for the region? Why did the authors only consider emissions from vehicles, energy, and industry? What about other emission sources in the region? What are the proportions of emissions from different anthropogenic sources?

- Line 80: No need to use "In contrast".

- Line 83: What are "interpolation techniques" that were used from 0.1° to 3km? Based on Figure B5, it appears to be bilinear interpolation?

- Line 119: "surface model evaluation" is not accurate and could lead to ambiguity. Please rephrase the sentence.

- Line 138: typo "January to 2015", delete "to".

- Line 240: It is the location of the site that has an impact, not only the latitude of the observation site.

- Line 256-260, Line 321-322: The current description, such as "this figure was somewhat compromised", makes it difficult to understand how the observations and simulations are being compared. Were missing values removed during the comparison?

- Line 301-302: This sentence does not contribute to the analysis and explanation in the manuscript since the authors only used EDGAR's energy and industry emissions, without incorporating urban area emissions.

- Line 312: I mentioned last time that 09-17h local is not only mid-afternoon but daytime. The author replied and made changes, but did not.

- Line 332: suggest change "for the study period" to "in February" to improve clarity.

- Line 334: it should be "at PDJ", not "in PDJ".

- Section 3.3.3: Why does the $XCO_2$ data for February and March not included in the analysis?

- Line 341: add "observed" to "the monthly average concentration stood at 412 ppm" to improve clarity.

- Line 347-348: This sentence appears suddenly. When the authors say "Figure 4 illustrates more positive $CO_2$ fluxes". They compare the "more" to what?

- Line 401: it should be "Figure 10b and 10c", not "Figure 10b".

- Proper nouns should be written in full with their abbreviations in parentheses when they first appear. After that, only the abbreviation should be used throughout the rest of the manuscript. This issue appears multiple times in the manuscript. For example:

  a) "MASP" in Line 4, Line 32, and Line 72.

  b) "VPRM" in Line 7, Line 45, and Line 86.

  c) "WRF-Chem" in Line 4, Line 44, and Line 57.

d) "VEIN" should first appear in Line 46 instead of Line 73.

I recommend that the authors standardize the use of abbreviations accordingly.

- Line 5 and Line 59: Letter case for METROCLIMA or Metroclima.
- The letter "F" in "Atlantic Forest" is sometimes capitalized and sometimes lowercase in the manuscript. Please ensure consistent capitalization throughout the text.

---

## Author Response (AR2)

Author's Response to the editor's comments on **"Monitoring and modeling seasonally varying anthropogenic and biogenic *CO2* over a large tropical metropolitan area"**.

First, we would like to thank the editor for their valuable comments related to the manuscript "**Monitoring and modeling seasonally varying anthropogenic and biogenic *CO2* over a large tropical metropolitan area**" by Rafaela Cruz Alves Alberti et al. Your valuable feedback has helped us identify areas for improvement and refine the manuscript accordingly. The editor's comments are written in black, while our author's comments are in **blue**. Modifications from the manuscript are in cyan and *italic*.

**Main: 1. Some explanations in the results section are not entirely correct or convincing.**

• Line 219-220, Line 228: The description is inconsistent and could easily lead to misunderstanding. Line 219 mentions that summer is "less intense in the Atlantic Forest," while Line 228 states "negative net fluxes in February, particularly in the Atlantic Forest." These statements seem contradictory.

**Indeed, we agree that these lines could lead to a misunderstanding regarding the Atlantic Forest results. We have clarified the apparent inconsistency between lines 219 and 228 by explicitly distinguishing the spatially averaged behavior of the Atlantic Forest biome across the domain (panel a) from the point-based simulation results (panel b). The revised paragraph now clarifies that. While the Atlantic Forest as a whole shows heterogeneous behavior in February with both $CO_2$ sinks and sources, the flux tower site shown in panel b is located in the northeastern portion of the ecosystem, where net $CO_2$ uptake was observed. These changes were made to ensure that readers can clearly understand the results at both the ecosystem and site-specific levels.**

"*During the summer, ecosystem productivity is expected to peak across all land cover classes, typically resulting in negative NEE. This behavior was clearly observed in February (Figure 4a) for Cerrado, sugarcane, and pasture areas. In contrast, the Atlantic Forest in the southwestern portion of the domain exhibited positive NEE values, an unexpected pattern for a summer month. This may be linked to a combination of structural and anthropogenic factors, as well as limitations of the model itself. The Atlantic Forest is marked by structural heterogeneity, extreme biodiversity, and high fragmentation, which can lead to significant local variation in $CO_2$ fluxes. In addition, the SEEG (2021) report highlights a progressive decline in the biome's carbon sink function. Model limitations also likely contribute to these discrepancies, particularly simplifications in VPRM's equations of respiration and phenology, which may not fully capture the complex dynamics of ecosystems like the Atlantic Forest (Rezende et al., 2018; Segura-Barrero et al., 2025).*"

• Figure 4b and 4c: Why does Figure 4b show the Atlantic Forest as red (positive), while in Figure 4c, the site clearly shows significant negative values during the day, which exceed the positive values at night? If the monthly average of the NEE from Figure 4c is calculated, would it still be positive as shown in Figure 4b?

**We appreciate the opportunity to clarify this point by providing the monthly mean NEE values for each flux tower site represented in Figure 4c. For August, the monthly average NEE values were the Atlantic Forest is 1.55 μmol m-2 s-1, the cerrado 0.55 μmol m-2 s-1, and the sugarcane 3.00 μmol m-2 s-1. Although Figure 4c shows pronounced negative fluxes during the daytime for the Atlantic Forest site, the nighttime positive fluxes are more consistent and frequent throughout the month. As a result, the monthly NEE is positive, which is consistent with the light red shading shown in Figure 4b. This same behavior is observed to varying degrees in the other ecosystems as well.**

• Line 268-269: The small absolute value of the PDJ observation-simulation bias is due to the small $CO_2$ signal at this site. If the goal is to discuss or compare the simulation performance with the IAG

site, a more reasonable approach would be to look at the relative error, e.g., signal-to-bias, rather than directly concluding that "model predictions are more accurate at PDJ."

**In response, we have revised the paragraph to avoid making direct claims about model performance at PDJ based solely on the magnitude of the absolute bias. Instead, we emphasize that the PDJ site exhibited low biases and lower variability between model and observationsand, which attribute this to PDJ site characteristics, its higher elevation, dense vegetation cover, and reduced influence from urban emissions. These factors contribute to a lower CO2 signal and a more straightforward representation of seasonal trends, which likely explains the smaller bias. Now the paragraph reads:**

*"The PDJ station exhibited low positive biases and smaller standard deviations between the model and observations. Its higher elevation and dense vegetation cover simplify the representation of seasonal trends, reducing the influence of urban emissions and resulting in lower CO2 concentrations at this site (see Figure B7 in Appendix B)."*

• Section 3.3.1: Line 271-277: As far as I know, in WRF-Chem, background, anthropogenic, and biogenic emissions are three separate variables (CO2_BCK, CO2_ANT, CO2_BIO) in the output netcdf files. The authors can simply display them individually or sum them to achieve the desired result, without the need to rerun simulations with different "emission scenarios" as stated. Did the authors modify some part of the model that requires resimulation?

**While it is true that WRF-Chem (WRF-GHG configurations) can provide separate CO2 variables such as CO2_BCK, CO2_ANT, and CO2_BIO, in our study, the VPRM model was used in offline mode, not coupled with WRF-Chem. As a result, the separation of flux components was not available within a single simulation, since the biogenic and anthropogenic fluxes were already combined (summed) in a single input file. To analyze the contribution of each component separately, we performed additional simulations using the same model configuration but different emission input files: one containing both anthropogenic and biogenic fluxes (as originally used), and others including only one component at a time. No changes were made to the model configuration; only the input files were varied between scenarios.**

• Line 346 and Figure 7h: The authors say, "This highlights the role of both biogenic and meteorological processes in shaping $CO_2$ variability at this site," but from Figure 7h, the increments for biogenic and anthropogenic emissions appear to be similar, and it seems that the increase is due to the rise in background concentrations. How does this align with the authors' statement?

**Indeed, we agree with your observation that the increase in CO2 concentrations at the PDJ site in late August is primarily associated with rising background concentrations, as shown in Figure 7h. This pattern was also observed at the IAG site (Figure 7f), indicating that the background signal played an important role at both locations during this period.**
**Additionally, we acknowledge that the original sentence "*This highlights the role of both biogenic and meteorological processes in shaping CO2 variability at this site*" may have led to some confusion, as it seemed to refer specifically to the end of the month. In fact, our intention was to highlight the overall behavior at the PDJ site throughout the entire period, not just the final days of August. To address this, we have revised the sentence to clarify this:**

*"... Regarding the source contributions, the model simulation aligned with the observed temporal profile, displaying a more pronounced biogenic signal than at the IAG site, which further emphasizes the significant role of vegetation as a source of CO2 emissions at this location.(Figure 7h). Before late August, observed values tended to be higher than the simulations, whereas in the final days of the month, the model overestimated CO2 concentrations. This overestimation is associated with an increase in background concentrations, a pattern also observed at the IAG site during the same period."*

• Section 3.3.2, in Line 361-363: Actually, I believe the analysis in Lines 349-363 is meaningless, especially the conclusions in Lines 361-363. When comparing observations and simulations, of course, the simulation values should consider anthropogenic, biogenic, and background emissions together, as this more closely reflects real-world conditions. Isn't this something that should be done? It's common sense. Why conduct so much analysis just to conclude that "simulating with only one factor leads to larger discrepancies with observations, while considering all three factors improves simulation performance"? If the authors' goal is to analyze the individual contributions of the three factors, then Figures 6 and 7 already serve that purpose.

**We agree that, from a modeling standpoint, it is indeed expected that simulations incorporating anthropogenic, biogenic, and background contributions would yield the best agreement with observed CO2 concentrations. However, our intention in that section was not to restate an obvious conclusion, but rather to provide a quantitative assessment of how each emission component, used in isolation or in combination, affects model performance in our specific study region. To clarify this and avoid general or redundant conclusions, we revised the paragraph to focus solely on presenting the bias and RMSE values for each scenario, allowing for a more objective comparison of model performance.**

**We rewrote the paragraph to:** *"The bias and RMSE for each simulation at the IAG and PDJ sites for February and August 2019 are illustrated (see Figure B7 in Appendix B). At IAG, the average bias ranged from -14.31 to -9.17 ppm, while at PDJ it ranged from -3.54 to -0.96 ppm. RMSE values were consistently higher at IAG, exceeding 20 ppm in most scenarios, while PDJ showed lower errors, generally below 12 ppm."*

• Line 371-372: Figure B4 shows that the evening rush hour for traffic occurs before 19:00, not after. How do the authors explain this?

**Indeed, this can not attribute a evening rush hour traffic. To correct this, we removed this affirmation in the sentence. Now it is read:**

*In Figure 8, both bar graphs of the hourly correlation between CO2 and CO concentrations show values above 0.5 for observed CO2 and above 0.25 for simulated CO2 during the early hours of the day (until 10h) and again in the evening (after 19h).*

**2. There are several discrepancies between the textual analysis and the figures, with several obvious errors.**

• Line 264: The simulated PDJ values are higher than the observed values. Why do the authors say "underestimated vehicular emissions in these areas"?

**It seems there has been a mistake in writing. Instead, it should be written:**

*"... likely stems from model limitations, including grid resolution and insufficient representation of localized characteristics at different sites."*

• Line 328 and Figure 7a: From Figure 7a, it appears that the IAG site has no observation data for February 21 and 22!

**We carefully reviewed the dataset and the corresponding figure. There is, in fact, observational data available for both February 21 and 22. However, due to temporal gaps, particularly on February 21st, this data was partially excluded during postprocessing. Specifically, our filtering step considered only data between 09:00 and 17:00 local time to ensure consistency in daily comparisons. On February 21st, data was only available during nighttime hours, and thus it was excluded from the analysis. In contrast, on February 22nd, there was sufficient daytime data**

**and which is therefore represented in Figure 7a. To clarify this, you can see in the figure below only the observed CO2 data used in Figure 7a. This provides a clearer view of the data gaps and supports our explanation.**

[Figure]

**Additionally, we revised the sentence in the manuscript to avoid misinterpretation and to correct the statement regarding February 21. The updated sentence now reads:**

"...Furthermore, on February 2nd and 22nd, observed CO2 peaks were captured by the model with similar magnitude only when both anthropogenic and biogenic emissions were included."

• Line 322 and Line 339: The IAG site has a model-to-observation difference of 8 ppm in February (Line 322) and 13 ppm in August (Line 339). The error in August is larger than that in February. Why do the authors say in Line 339 "only 13 ppm, i.e., a closer approximation compared to February"?

**Indeed. This is a wrong statement. This sentence is now:**

"...resulting in a discrepancy of 13 ppm, i.e., a higher difference compared to February."

• Line 341-342: observation is 412 ppm, model is 412 ppm in the text. Why do the authors say that the observation is surpassing the simulation?

**The original statement referred to a small decimal difference between the observed and simulated monthly averages (observed: 412.75 ppm; simulated: 412.15 ppm). However, since these decimal values were not presented in the text, the wording has led to confusion. To avoid misinterpretation, we have revised the sentence in the manuscript to:**

"...the monthly average concentration observed and simulated were 412 ppm. While the model slightly underestimated some days in the month and overestimated others, it generally captured the observed variability."

• Line 351-352: This is a clear mistake! It should be February, not August. Also, it's not Figure 7c, and it should be a positive bias, not a negative bias.

**Indeed. This paragraph was rewritten due to the previous review comment.**

"The bias and RMSE for each simulation at the IAG and PDJ sites for February and August 2019 are illustrated (see Figure B7 in Appendix B). At IAG, the average bias ranged from -14.31 to -9.17 ppm, while at PDJ it ranged from -3.54 to -0.96 ppm. RMSE values were consistently higher at IAG, exceeding 20 ppm in most scenarios, while PDJ showed lower errors, generally below 12 ppm."

• Line 356-357: clear mistake! Figure B7 shows that the PDJ for only Anthropogenic emissions does not have the poorest RMSE.

**Same as the previous comment. This paragraph was rewritten due to the previous review comment.**

*"The bias and RMSE for each simulation at the IAG and PDJ sites for February and August 2019 are illustrated (see Figure B7 in Appendix B). At IAG, the average bias ranged from -14.31 to -9.17 ppm, while at PDJ it ranged from -3.54 to -0.96 ppm. RMSE values were consistently higher at IAG, exceeding 20 ppm in most scenarios, while PDJ showed lower errors, generally below 12 ppm."*

• Line 358: clear mistake! Figure B7 shows that IAG with anthropogenic sources in August does not have the highest RMSE.

**Same as the previous comment. This paragraph was rewritten due to the previous review comment.**

*"The bias and RMSE for each simulation at the IAG and PDJ sites for February and August 2019 are illustrated (see Figure B7 in Appendix B). At IAG, the average bias ranged from -14.31 to -9.17 ppm, while at PDJ it ranged from -3.54 to -0.96 ppm. RMSE values were consistently higher at IAG, exceeding 20 ppm in most scenarios, while PDJ showed lower errors, generally below 12 ppm."*

**3. Many sentences throughout the manuscript are redundant or have syntax errors. I recommend that the authors read through the entire text and remove redundant sentences. Please see some examples bellows:**

• Line 44-47: "Coupled VPRM" and "integrated VEIN" are parallel in structure, but the sentence is too long. The intended meaning is that the VEIN model is integrated into the VPRM model. I suggest rewriting the sentence for clarity.

**We have rewritten the paragraph to improve clarity and have also included additional information regarding the specific sectors considered from the EDGAR inventory, which was not previously mentioned in the manuscript.**

*"This study aims to address these gaps by conducting a comprehensive analysis of anthropogenic and biospheric CO2 dynamics near the MASP. To achieve this, we employed the Weather Research and Forecasting model with Chemistry (WRF-Chem), offline coupled with the Vegetation Photosynthesis and Respiration Model (VPRM) (Mahadevan et al., 2008). Vehicular emissions were incorporated using the VEIN model (Ibarra-Espinosa et al., 2018), while emissions from the industrial, energy, residential, and refinery sectors were derived from the EDGAR inventory. This integrated modeling framework enables a detailed assessment of the main drivers of CO2 variability in the region."*

• Line 67-70: You can directly state "$CO_2$ initial and boundary conditions" instead of repeating it twice "initial and boundary conditions" in the sentence. General Additionally, I recommend using "$CO_2$" instead of "chemical," also change it in Table 1.

**We have revised the paragraph to avoid the repetition of "initial and boundary conditions" and to improve readability by breaking down long sentences. Additionally, we replaced the term "chemical" with "CO2" in Table 1, as suggested.**

*"The meteorological conditions used to drive the simulations were obtained from the European Centre for Medium-Range Weather Forecasts (ECMWF) ERA5 reanalysis dataset, with a horizontal resolution of 0.25° × 0.25° and 6-hourly intervals (Hersbach, 2016). For $CO_2$, initial and boundary conditions were provided by Carbon Tracker, which offers data at a horizontal resolution of 3° in longitude and 2° in latitude, with 25 vertical layers (http://carbontracker.noaa.gov)."*

• Line 83-84: "EDGAR lacks temporal variability" and "inventory does not provide hourly profile" are repetitive.

**We revised the sentence to:**

*"However, as the EDGAR inventory does not provide hourly emission profiles, its emissions were assumed constant throughout the day (Fig. B2 in Appendix B)."*

• Line 87: "as a flux input" and "as input data" are repetitive.

**We have revised the sentence to eliminate the redundancy. The updated sentence now reads:**

*"Biogenic CO2 fluxes were simulated offline using the VPRM model (Mahadevan et al., 2008) and incorporated as flux input data in the WRF-Chem simulations."*

• Line 171-172: syntax errors, incomplete sentence.

**We have corrected the grammatical inconsistency. The revised version reads:**

*"To evaluate the model performance, we calculated the bias, RMSE, and R², with the corresponding equations provided in Appendix A."*

• Line 251-252: The term "PDJ" appears twice in the same sentence, making it redundant.

**We have revised the sentence to remove the redundancy.**

*"Additionally, lower CO2 concentrations were expected at PDJ during the summer due to the stronger vegetation signal compared to the IAG site."*

• Line 267-268: The expression is redundant, as the parts before and after "and" convey the same meaning.

**We have written the sentence to remove the redundancy and enhance its clarity. The revised version is:**

*"The PDJ station exhibited low positive biases, indicating better agreement and lower errors between the model and observations across all periods. The higher elevation and vegetation cover at PDJ simplify seasonal trend modeling, reducing the impact of urban factors and enhancing model performance (see Figure B7 in Appendix B)."*

**4. The quality and color of the figures are not visually easy to get information:**

• The color scheme in Figure 1 is not visually appealing and makes it difficult to identify the stations. Additionally, I recommend using (a), (b), etc., instead of referring to "first panel" and "second panel" to improve clarity. Similarly, see Line 278 to avoid unclear descriptions.

**We revised Figure 1 by updating the color scheme to improve visual contrast among land-use categories and to make the station locations more distinguishable. Additionally, we labeled the panels as (a) and (b) and updated both the figure caption to replace vague descriptions like "first panel" or "second panel," as suggested.**

[Figure]

*"Figure 1. Panel (a) shows the terrain height and urban boundaries of the MASP region within the WRF-Chem model domain (D01). Station classifications are indicated using different symbols: Urban (★), Urban Park (✚), and Park (▲). Panel (b) presents the land use category map for the same domain (D01), which was used by the VPRM model to calculate CO2 fluxes. The colors of the station markers represent the type of measurements conducted at each location: red indicates stations measuring both meteorological variables (Met) and CO2 concentrations; green indicates stations measuring only Met; dark yellow denotes stations measuring both Met and CO concentrations; and black indicates stations measuring only CO2 concentrations. The IAG station is marked as (✚), the PDJ station is (▲), Pinheiros station is (★), Guarulhos and Parque D.Pedro II are (★)."*

• Why are the figure numbers in the supplementary material not assigned in the order in which they appear in the manuscript?

**Thank you for pointing this out. We have reordered the supplementary figures to match the order in which they are cited in the manuscript, ensuring better clarity and consistency for the reader, and also updated in the text. Below you can see the new ordered in appendix B:**

*Figure B1. Vehicular CO2 emissions over the study domain (D01). (a) Spatial distribution of average daily CO2 emissions from vehicles for August 2019, as estimated by the VEIN model. (b) Total monthly vehicular CO2 emissions from February to August 2019 over domain D01. (c) Diurnal profile of vehicular CO2 emissions at the IAG site during August 2019.*

*Figure B2. CO2 emissions from energy, residential, refineries and industry sectors by the EDGAR inventory over the study domain (D01). (a) Spatial distribution of average daily CO2 emissions in August 2019. (b) Monthly total CO2 emissions from February to August 2019 over domain D01.*

*Figure B3. Average daily anthropogenic CO2 emissions (in tons) for August 2019 within the simulated domain, disaggregated by sector. Bars represent the mean daily emissions per sector; while percentages indicate each sector's relative contribution to total anthropogenic emissions.*

*Figure B4. The panels in a) show the scatter plots of hourly measurements of 2 m air temperature (T2m) and b) show 10 m wind speed (WS) compared to observed data from the Parque D.Pedro II station. The figure illustrates the relationship between modeled and observed data. The panels in c) show the daily averages from February to August 2019 of 2 m air temperature (T2m), 10 m wind speed (WS), and wind direction (WD). Black line represents the observed data and red line represents the model simulation.*

*Figure B5. The same as B4 for Guarulhos.*

*Figure B6. The same as B4 for Pinheiros.*

*Figure B7. Diurnal cycle of in situ CO2 concentration and planetary boundary layer (PBL) height for the entire simulated period. The black line represents the median hourly concentrations from WRF-Chem, while the purple line corresponds to the observed values. The shaded areas indicate the interquartile ranges. Panel a) shows the observed and simulated surface CO2 concentration at the IAG site; b) the simulated PBL height at the IAG site; c) the observed and simulated surface CO2 concentration at the PDJ site; and d) the simulated PBL height at the PDJ site.*

*Figure B8. Bias (ppm) and RMSE (ppm) for each simulation at the surface CO2 observation sites. Panels (a) and (b) represent the simulations for February, while panels (c) and (d) represent the simulations for August (ALL_\*: black, ANTH_\*: red, VPRM_\*: green) \*Represents the observation sites, e.g. IAG and PDJ.*

*Figure B9. Time series of smoothed column concentrations observed (black) and modeled (red) for the period from 1 April 2019 to 31 August 2019.*

• Line 213 and Figure 4c: Figure 4c shows hourly data, not daily data, right?

**Indeed, Figure 4c presents hourly variation, not daily data. We have corrected this in the text and also revised the figure caption to reflect the content accurately.**

*"...The first panel in Figure 4 shows the monthly net CO2 flux simulated by the VPRM model for 2019.February represents a summer month, while August represents a winter month. The second panel shows the monthly hourly net CO2 flux simulated at the three flux tower sites used to optimize the VPRM model parameters"*

*Figure 4. The first panel shows the monthly mean of net ecosystem exchange (NEE) $(mol\,km^{-2}\,h^{-1})$ for February (a) and August (b) 2019. The second panel (c) presents the hourly variability of NEE $(\mu mol\,m^{-2}\,s^{-1})$ for the same months (February and August) at three different PFTs: Atlantic Forest, Cerrado/Savanna, and Sugarcane.*

• Figure 4a: Figure 4a shows the monthly mean NEE, not the "monthly mean diurnal cycle" as stated in the caption. How can the diurnal cycle be observed from Figure 4a?

**Thank you for this correction. Figure 4a shows the monthly mean NEE, not the monthly mean diurnal cycle. This was a typo in the figure caption, while the description in the main text was already correct. We have corrected the caption in Figure 4 to reflect this.**

*Figure 4. The first panel shows the monthly mean of net ecosystem exchange (NEE) $(mol\,km^{-2}\,h^{-1})$ for February (a) and August (b) 2019. The second panel (c) presents the hourly variability of NEE $(\mu mol\,m^{-2}\,s^{-1})$ for the same months (February and August) at three different PFTs: Atlantic Forest, Cerrado/Savanna, and Sugarcane.*

• Figure 6: Why use a discrete colorbar instead of a continuous one? This makes it difficult to distinguish values like 6 ppm in Line 280 and 8 ppm in Line 295. Based on the current colorbar, one could also interpret the value as 4 ppm, right?

**In response to this suggestion, we revised Figure 6 by replacing the discrete colorbar with a continuous one. Additionally, we improved the font sizes for better readability and included latitude and longitude gridlines to enhance spatial reference. These changes help clarify the CO2 concentration gradients across the domain and make the increases more visually evident.**

[Figure]

February 2019

August 2019

• Figure B4 and B5: Please include the latitude and longitude. Additionally, there is no need to display different months as it is difficult to discern any significant differences. It would be better to show just one figure of spatial distribution and use other types of charts to present the monthly emission totals. Moreover, the caption for Figure B5 is not accurate, e.g., "daily mean"? which sectors from EDGAR?

**We have revised Figures B4 and B5, which are now renumbered as Figures B1 and B2 due to reordering. As suggested by the reviewer, we removed the multiple monthly spatial plots and now present only one representative spatial distribution of average daily $CO_2$ emissions for August 2019. Latitude and longitude ticks have been added. Additionally, monthly total emissions from February to August 2019 are now presented using bar charts. In Figure B1 (VEIN emissions), we included a third panel (c) showing the diurnal profile of vehicular $CO_2$ emissions at the IAG site during August 2019. In Figure B2 (EDGAR emissions), the panels display the spatial distribution of average daily $CO_2$ emissions for August 2019 and the corresponding monthly emissions totals from February to August 2019. We have also clarified in the figure caption that these emissions include combined contributions from the energy, industry, residential, and refinery sectors.**

[Figure]

*Figure B1. Vehicular CO2 emissions as estimated by the VEIN model over the study domain (D01). The panel (a) represents the spatial distribution of average daily CO2 emissions for August 2019 over D01. Panel (b) represents the total monthly CO2 emissions from February to August 2019 over the D01. Panel (c) shows the diurnal profile of CO2 emissions at the IAG site during August 2019.*

[Figure]

*Figure B2. CO2 emissions from energy, residential, refineries, and industry sectors by the EDGAR inventory over the study domain (D01). Panel (a) shows the spatial distribution of average daily CO2 emissions for August 2019 over D01. The panel (b) represents the monthly total CO2 emissions from February to August 2019 over the domain.*

• Figure 9: redundant in caption "Daily mean concentrations of CO2 observed concentrations".

**We have revised the caption for Figure 9:**

*"Daily mean concentrations of CO2, both observed (black dashed line) and simulated (purple line), at the IAG site, along with observed CO concentrations (red dotted line) at the Pinheiros site during August 2019."*

***Additional improvements have been made to Figures 2, 4, 5, and 7, which were revised to enhance axis labels and captions. Similarly, Figures B4, B5, and B6 in Appendix B have been updated to improve the readability of axes, titles, and legends.***

**Specific:**

• Line 48: What does "smoothed" XCO2 mean? How is the $XCO_2$ from WRF-Chem smoothed?

**In the sentence on line 48, "smoothed XCO2" refers to WRF-Chem-simulated column-averaged $CO_2$ concentrations that were post-processed using the satellite averaging kernels and a priori profiles, as described in Section 2.2.3 of the Methods. This smoothing step ensures that the modeled CO2 is vertically weighted in a manner consistent with the sensitivity of the OCO-2 instrument, allowing for a valid comparison between model output and satellite retrievals. To avoid misunderstanding, we revised the sentence in the manuscript to clarify this.**

*In addition, we utilized data from the OCO-2 satellite to cover the study domain, comparing WRF-Chem-simulated XCO2 concentrations (considering biogenic and anthropogenic emissions) post-processed using OCO-2 averaging kernels (i.e., smoothed XCO2).*

• Section 2.1.1: What is the total anthropogenic emission for the region? Why did the authors only consider emissions from vehicles, energy, and industry? What about other emission sources in the region? What are the proportions of emissions from different anthropogenic sources?

**We have revised the paragraph in Section 2.1.1 to enhance clarity and precision. In response to your question regarding the scope and total of anthropogenic emissions, we have updated the section to explicitly state that, in addition to vehicular sources (from VEIN), our analysis**

**includes emissions from the energy, industry, residential, and refinery sectors based on the EDGAR inventory.**

**To further address the proportions of these emissions, we have added a figure (Figure B3 in Appendix B) that illustrates the average daily anthropogenic CO2 emissions in August 2019 for the MASP. According to this analysis, vehicular emissions represent 76.1% of the total anthropogenic CO2, followed by industry (10.0%), refineries (7.6%), residential (3.8%), and energy (2.5%) sectors.**

[Figure]

*Figure B3. Average daily anthropogenic CO2 emissions (in tons) for August 2019 within the simulated domain, disaggregated by sector. Bars represent the mean daily emissions per sector, while percentages indicate each sector's relative contribution to total anthropogenic emissions.*

• Line 80: No need to use "In contrast".

**The expression "In contrast" was removed, as it was unnecessary in that context.**

• Line 83: What are "interpolation techniques" that were used from 0.1° to 3km? Based on Figure B5, it appears to be bilinear interpolation?

**We also clarified that bilinear interpolation was used to regrid the EDGAR emissions from the original 0.1° × 0.1° resolution to the 3 km spatial resolution required by WRF-Chem. Thank you for these observations. In response, we revised the paragraph in Section 2.1.1 to improve clarity and address the concerns raised.**

*"...EDGAR provides global annual emissions at 0.1° × 0.1° spatial resolution, which we regridded to 3 km using bilinear interpolation to match the WRF-Chem model domain. Because EDGAR does not provide hourly temporal profiles, these emissions were assumed constant over the day (Fig. B2 in Appendix B). To evaluate the relative contribution of each sector to total emissions in the region, Figure B3 (Appendix B) presents the mean daily CO2 emissions in August 2019. Transport emissions (VEIN) represented the dominant share, accounting for 76.1%, followed by industry (10.0%), refineries (7.6%), residential (3.8%), and energy (2.5%) sectors."*

• Line 119: "surface model evaluation" is not accurate and could lead to ambiguity. Please rephrase the sentence.

**We agree that the original expression "surface model evaluation" could lead to ambiguity. To improve clarity, we revised the sentence to:**

*"We assessed near-surface model performance using CO2 observations from the METROCLIMA network in São Paulo (see Table 3 and Figure 1), the first conventional in situ greenhouse gas measurement network established in South America (www.metroclima.iag.usp.br)."*

• Line 138: typo "January to 2015", delete "to".

**Done.**

• Line 240: It is the location of the site that has an impact, not only the latitude of the observation site.

**Thank you for your comment. We agree that the latitude was not the correct word to use in this context. To improve clarity and accuracy, we revised the sentence to emphasize the influence of the site's geographical location.**

*"This variation in CO2 levels is primarily influenced by the geographical location of the observation site, as well as meteorological conditions such as wind speed and atmospheric stability, and seasonal patterns of photosynthesis and vehicular traffic (see Fig. B4 in Appendix B)."*

• Line 256-260, Line 321-322: The current description, such as "this figure was somewhat compromised", makes it difficult to understand how the observations and simulations are being compared. Were missing values removed during the comparison?

**We agree that the phrasing was unclear and revised the text to better explain the comparison between observed and simulated data. Specifically, we clarified that missing values in the observational dataset were removed before calculating the monthly mean. The revised text avoids vague expressions and more clearly states how the comparison was made.**

*"In Figure 7a, which represents only one summer month with available observational data (February 2019), the model generally underestimated CO2 concentrations. The observed average was 424.0 ppm, while the simulated average was 416.0 ppm, an underestimation of approximately 8 ppm. This difference may be partially attributed to the presence of data gaps in the observational data for this site, as only available values were considered when calculating the monthly mean."*

• Line 301-302: This sentence does not contribute to the analysis and explanation in the manuscript since the authors only used EDGAR's energy and industry emissions, without incorporating urban area emissions.

**Thank you for the comment. However, we would like to clarify that, in addition to the energy and industry sectors, we also incorporated emissions from the residential and refinery sectors from EDGAR inventory, all of which include sources within urban areas.**

• Line 312: I mentioned last time that 09-17h local is not only mid-afternoon but daytime. The author replied and made changes, but did not.

**We corrected them in section 3.3.2.**

• Line 332: suggest change "for the study period" to "in February" to improve clarity.

**Thank you for the suggestion, and we incorporated it.**

• Line 334: it should be "at PDJ", not "in PDJ".

**Done.**

• Section 3.3.3: Why does the $XCO_2$ data for February and March not included in the analysis?

**We did not include XCO2 data for February and March in the analysis because there were no valid OCO-2 observations available over the study domain for those months.**

• Line 341: add "observed" to "the monthly average concentration stood at 412 ppm" to improve clarity.

**This sentence had already been revised in response to a previous comment to address clarity regarding the observed and simulated values. The updated sentence now reads:** *"In contrast, for PDJ (Figure 7g), both the observed and simulated monthly average concentrations were 412 ppm."* **Which explicitly states that the 412 ppm refers to the observed concentration, as well as the simulated one.**

• Line 347-348: This sentence appears suddenly. When the authors say "Figure 4 illustrates more positive $CO_2$ fluxes". They compare the "more" to what?

**We decided to remove it, as it did not contribute meaningfully to the analysis or improve the interpretation of the results.**

• Line 401: it should be "Figure 10b and 10c", not "Figure 10b".

**Done.**

• Proper nouns should be written in full with their abbreviations in parentheses when they first appear. After that, only the abbreviation should be used throughout the rest of the manuscript. This issue appears multiple times in the manuscript. For example: a) "MASP" in Line 4, Line 32, and Line 72. b) "VPRM" in Line 7, Line 45, and Line 86. c) "WRF-Chem" in Line 4, Line 44, and Line 57. General d) "VEIN" should first appear in Line 46 instead of Line 73.

I recommend that the authors standardize the use of abbreviations accordingly.

**We have reviewed the manuscript and corrected the errors pointed out by the reviewer.**

• Line 5 and Line 59: Letter case for METROCLIMA or Metroclima.

**Done.**

• The letter "F" in "Atlantic Forest" is sometimes capitalized and sometimes lowercase in the manuscript. Please ensure consistent capitalization throughout the text.

**Done.**